# A Lazy Hessian Evaluation Framework for Accelerating Stochastic Bilevel Optimization

## Abstract

Bilevel optimization has recently gained popularity because of its applicability in many machine learning applications. Hypergradient-based algorithms have been widely used for solving bilevel optimization problems because of their strong theoretical and empirical performance in many applications. However, computing these hypergradients requires the evaluation of Hessians (or Hessian-vector products) of the lower-level objective, which presents a major computational bottleneck. To address this challenge, in this paper, we propose LazyBLO (Lazy Hessian Evaluation in Bilevel Optimization), an algorithmic framework that allows infrequent Hessian computation during the execution of the algorithm for solving stochastic bilevel problems. This allows the algorithm to execute faster compared to the state-of-the-art (SOTA) algorithms that evaluate either a single or multiple Hessians in each iteration. We theoretically establish the performance of vanilla SGD-based LazyBLO and show that, despite the additional errors incurred by the infrequent Hessian evaluations, LazyBLO surprisingly matches the computation complexity of the existing SGD-based bilevel algorithms. Extensive experiments further demonstrate that LazyBLO enjoys significant gains in numerical performance compared to the SOTA approaches. To our knowledge, this is the first work to theoretically establish that multiple Hessian computations are not necessary within each iteration to guarantee the convergence of stochastic bilevel algorithms.

## 1 Introduction

Bilevel optimization refers to the class of problems with two levels of hierarchy, wherein the solution of the upper-level problem depends on the minimizer of the lower-level problem. Formally, a bilevel problem is stated as:

$$\min_{\mathbf{x} \in \mathbb{R}^u} \left\{ \ell(\mathbf{x}) \triangleq f\left(\mathbf{x}, \mathbf{y}^*(\mathbf{x})\right) \triangleq \mathbb{E}_{\xi \sim \pi_f}\left[ f\left(\mathbf{x}, \mathbf{y}^*(\mathbf{x}); \xi\right) \right] \right\}$$

$$\text{s.t. } \mathbf{y}^*(\mathbf{x}) = \arg\min_{\mathbf{y} \in \mathbb{R}^l} \left\{ g(\mathbf{x}, \mathbf{y}) \triangleq \mathbb{E}_{\zeta \sim \pi_g}[g(\mathbf{x}, \mathbf{y}; \zeta)] \right\}, \tag{1}$$

where $f(\mathbf{x}, \mathbf{y}) : \mathbb{R}^u \times \mathbb{R}^l \to \mathbb{R}$ and $g(\mathbf{x}, \mathbf{y}) : \mathbb{R}^u \times \mathbb{R}^l \to \mathbb{R}$ are upper (UL) and lower-level (LL) objectives, respectively. Both the UL and LL objectives are assumed to be smooth while the LL objective is strongly convex with respect to $\mathbf{y}$. Moreover, $\xi \sim \pi_f$ (resp. $\zeta \sim \pi_g$) represents a sample of the UL (resp. LL) objective from distribution $\pi_f$ (resp. $\pi_g$).

Stochastic bilevel problems in (1) have recently gained prominence as many popular machine learning problems can be modeled in this form. A few typical examples include hyperparameter optimization (Franceschi et al., 2018; Shaban et al., 2019; Bao et al., 2021), meta-learning (Franceschi et al., 2018; Rajeswaran et al., 2019; Ji et al., 2020), adversarial training (Li et al., 2019; Tian et al., 2021; Zhang et al., 2022), reinforcement learning (Konda & Tsitsiklis, 1999; Hong et al., 2020), neural architecture search (Liu et al., 2018; Hu et al., 2020; Lian et al., 2019), data hyper-cleaning (Franceschi et al., 2018; Shaban et al., 2019), dictionary learning (Mairal et al., 2011; Lecouat et al., 2020a;b), and more recently, the pretraining-finetuning pipeline (Li et al., 2024; Wu et al., 2024) and data reweighting (Pan et al., 2024) in large language models (LLMs). Consequently, a major research effort has been focused on developing efficient algorithms for solving stochastic bilevel optimization problems.

Among all existing methods for stochastic bilevel optimization (see Section 2 for detailed discussions), a state-of-the-art (SOTA) approach is the approximate implicit differentiation (AID) method, which relies on directly computing the approximate implicit gradient of the objective $\ell(\cdot)$ using the implicit function theorem (Ghadimi & Wang, 2018). Because of its ease of implementation, AID is usually the algorithm of choice for many machine learning applications. A typical AID algorithm updates the LL variable using standard stochastic gradient descent (SGD) while the UL variable is updated in each iteration using: $\mathbf{x}^+ = \mathbf{x} - \alpha h^f$, where the descent direction $h^f$ (also often referred to as hypergradient) is an approximation of the implicit gradient, i.e.,

$$h^f \approx \nabla\ell(\mathbf{x}) = \nabla_{\mathbf{x}} f(\mathbf{x}, \mathbf{y}^*(\mathbf{x})) - \nabla^2_{\mathbf{xy}} g(\mathbf{x}, \mathbf{y}^*(\mathbf{x})) \big[\nabla^2_{\mathbf{yy}} g(\mathbf{x}, \mathbf{y}^*(\mathbf{x}))\big]^{-1} \nabla_{\mathbf{y}} f(\mathbf{x}, \mathbf{y}^*(\mathbf{x})). \quad (2)$$

Although AID has been widely adopted for stochastic bilevel optimization in the literature, the computation of the hypergradient $h^f$ in AID faces two major challenges:

① The hypergradient in Eq. (2) requires **multiple** Hessian-vector product (HVP) evaluations for approximating the Hessian inverse in each iteration. This creates a major computational bottleneck for solving the problem in Eq. (1) since the explicit Hessian evaluations are computationally expensive. For example, the Hessian contains one million elements even for a moderately sized problem of dimension $d = 1000$. What is worse is that inverting such a Hessian typically has a computation complexity of $\mathcal{O}(d^3)$, which is time-consuming even for a moderately sized problem. Some modern automatic differentiation tools (e.g., Pearlmutter trick (Pearlmutter, 1994) and Jax (Bradbury et al., 2018)) have been proposed to accelerate the Hessian computation, and HVP computation may not be a major computational bottleneck in some situations where extremely computationally powerful GPUs are available. However, for many resource-constrained and computation-constrained settings (e.g., using small or edge-based devices without GPUs), HVP computation is still a computational bottleneck. For example, each HVP computation could be at least two to six times more expensive than gradient computation using Jax when performed on CPUs, which is still non-trivial, and the cost due to HVP remains not negligible in such systems. Moreover, we note that one Hessian inverse estimation needs multiple HVP computations (Ghadimi & Wang, 2018; Hong et al., 2020). As a result, the total cost of the HVP computation depends on the Hessian-inverse estimation accuracy. This would make the computational cost even higher.

② The hypergradient in Eq. (2) depends on the optimal solution of the LL problem $\mathbf{y}^*(\mathbf{x})$. However, solving the LL problem often requires an iterative method. Thus, solving the LL problem to optimality to obtain an exact value of $\mathbf{y}^*(\mathbf{x})$ may be expensive or even infeasible in practice.

We note that, although Challenge ② has been intensively studied in the literature and addressed to some extent (e.g., the hypergradient is approximated with $\mathbf{y}^*(\mathbf{x})$ being replaced by $\mathbf{y}^+ \approx \mathbf{y}^*(\mathbf{x})$), Challenge ① remains under-explored. So far, a foundational open problem in the theory of stochastic bilevel optimization naturally arises:

> **(Q)**: Can we design algorithms that require fewer Hessian evaluations compared to SOTA, and is it feasible to guarantee any theoretical performance for such algorithms?

In this paper, we answer the above question by developing a new algorithmic framework called LazyBLO (Lazy Hessian Evaluation in Bilevel Optimization), which allows infrequent Hessian (Hessian-vector product) evaluations in solving stochastic bilevel problems. Thus, LazyBLO alleviates the computational bottleneck in stochastic bilevel optimization. Specifically, in our LazyBLO approach, a stale version of Hessian is used for multiple iterations while new gradients are computed at each step, thus leading to computational savings. The intuition behind LazyBLO is that, for iterations that are not separated too far from each other, the parameter values usually do not vary significantly. This implies that the Hessians evaluated at these points are highly correlated. Thus, a stale Hessian can still be used to approximate a new one.

However, due to the additional errors accumulated because of the use of these stale Hessians, approximate Hessian (HVP) evaluations, and the coupling hierarchical structure of the bilevel problems, it is unclear whether LazyBLO will converge or not. Somewhat surprisingly, we prove that, despite the previously mentioned accumulated errors, LazyBLO not only converges but also achieves the *same* convergence rate as those of the SOTA non-lazy bilevel algorithms. To our knowledge, this is the first work that uses infrequent Hessian computations for computational savings but still can achieve convergence guarantee in solving stochastic bilevel problems.

Our major contributions in this work are summarized as follows:

- We develop a new algorithmic framework LazyBLO that allows the stochastic bilevel algorithms to compute HVPs infrequently. Specifically, the proposed framework updates the HVPs only over a subset of training iterations, while using stale Hessian information in the rest of the iterations.

- We theoretically establish the performance of LazyBLO when the UL and LL updates are performed using vanilla SGD-type updates. We show that the proposed lazy approach, which is supposed to perform worse due to stale Hessian information, can actually *match* the convergence performance of the SOTA bilevel algorithms. Specifically, we show that to achieve an $\epsilon$-stationary point, LazyBLO requires $\mathcal{O}(\epsilon^{-2})$ partial gradient and HVP evaluations. Moreover, thanks to the less frequent Hessian evaluations, the *wall-clock time* of LazyBLO is significantly reduced compared to the SOTA approaches.

- We corroborate our theoretical findings via numerical experiments on data hyper-cleaning and deep hyper-representation tasks with real-world datasets. Our numerical results verify that the infrequent evaluations of HVP lead to considerable computational savings.

## 2 RELATED WORK

In this section, we provide a brief overview of several areas of the most related work: ① AID-based bilevel optimization, ② Hessian-free bilevel optimization, and ③ other uses of infrequent Hessian evaluations, thus putting our work into comparative perspective to highlight our novelty. Due to space limitation, we give a summary of other related bilevel optimization methods in Appendix A.

① **AID-Based Bilevel Optimization:** In Table 1, we compare existing AID-based stochastic bilevel algorithms. BSA (Ghadimi & Wang, 2018) provided the first finite-time convergence guarantees for bilevel optimization. The stochastic bilevel algorithms (e.g., BSA in (Ghadimi & Wang, 2018)), stocBiO in (Ji et al., 2021), AmIGO in (Arbel & Mairal, 2022)) that use vanilla-SGD updates require $\mathcal{O}\left(\epsilon^{-2}\right)$ for both partial gradient evaluations and HVP evaluations to reach an $\epsilon$-stationary point. Meanwhile, several works (e.g., SUSTAIN in (Khanduri et al., 2021b), SVRB in (Guo et al., 2021), MRBO and VRBO in (Yang et al., 2021)) utilize momentum-based approaches and/or variance reduction approaches to accelerate the convergence of vanilla SGD-based algorithms, achieving $\mathcal{O}\left(\epsilon^{-1.5}\right)$ for both partial gradient evaluations and HVP evaluations. Although these works guarantee finite-time convergence, the practical numerical performance of these bilevel algorithms is slow since they require multiple Hessian (or HVP) evaluations of the LL objective in each iteration to approximate the Hessian inverse. In this work, we show that the Hessian computations can be *skipped* and *stale* Hessian information computed from the previous rounds can be used without hurting the convergence performance while allowing the algorithms to execute much faster.

Table 1: Comparison of stochastic bilevel algorithms (TTSA (Hong et al., 2020), BSA (Ghadimi & Wang, 2018), stocBiO (Ji et al., 2021), SOBA (Dagréou et al., 2022), ALSET (Chen et al., 2021), AmIGO (Arbel & Mairal, 2022), MSTSA (Khanduri et al., 2021a), SUSTAIN (Khanduri et al., 2021b), MRBO (Yang et al., 2021), SEMA (Yang et al., 2021), SVRB (Guo et al., 2021), MA-SOBA (Chen et al., 2024), VRBO (Yang et al., 2021), FSLA (Li et al., 2022)).

|  | # of PG | # of HVP | Update |
|---|---|---|---|
| TTSA | $\mathcal{O}\left(\epsilon^{-2.5}\right)$ | $\mathcal{O}\left(\epsilon^{-2.5}\right)$ | SGD |
| BSA | $\mathcal{O}\left(\epsilon^{-2}\right)$ | $\tilde{\mathcal{O}}\left(\epsilon^{-2}\right)$ | SGD |
| stocBiO | $\mathcal{O}\left(\epsilon^{-2}\right)$ | $\tilde{\mathcal{O}}\left(\epsilon^{-2}\right)$ | SGD |
| SOBA | $\mathcal{O}\left(\epsilon^{-2}\right)$ | $\mathcal{O}\left(\epsilon^{-2}\right)$ | SGD |
| ALSET | $\mathcal{O}\left(\epsilon^{-2}\right)$ | $\mathcal{O}\left(\epsilon^{-2}\right)$ | SGD |
| AmIGO | $\mathcal{O}\left(\epsilon^{-2}\right)$ | $\mathcal{O}\left(\epsilon^{-2}\right)$ | SGD |
| **LazyBLO** | $\mathcal{O}\left(\epsilon^{-2}\right)$ | $\mathcal{O}\left(\epsilon^{-2}\right)$ | **SGD** |
| MSTSA | $\mathcal{O}\left(\epsilon^{-2}\right)$ | $\tilde{\mathcal{O}}\left(\epsilon^{-2}\right)$ | Momentum |
| SUSTAIN | $\tilde{\mathcal{O}}\left(\epsilon^{-1.5}\right)$ | $\tilde{\mathcal{O}}\left(\epsilon^{-1.5}\right)$ | Momentum |
| MRBO | $\mathcal{O}\left(\epsilon^{-1.5}\right)$ | $\tilde{\mathcal{O}}\left(\epsilon^{-1.5}\right)$ | Momentum |
| SEMA | $\tilde{\mathcal{O}}\left(\epsilon^{-2}\right)$ | $\tilde{\mathcal{O}}\left(\epsilon^{-2}\right)$ | Momentum |
| SVRB | $\mathcal{O}\left(\epsilon^{-1.5}\right)$ | $\mathcal{O}\left(\epsilon^{-1.5}\right)$ | Momentum |
| MA-SOBA | $\mathcal{O}\left(\epsilon^{-2}\right)$ | $\mathcal{O}\left(\epsilon^{-2}\right)$ | Momentum |
| VRBO | $\tilde{\mathcal{O}}\left(\epsilon^{-1.5}\right)$ | $\tilde{\mathcal{O}}\left(\epsilon^{-1.5}\right)$ | VR |
| FSLA | $\mathcal{O}\left(\epsilon^{-2}\right)$ | $\mathcal{O}\left(\epsilon^{-2}\right)$ | VR |

PG: Partial gradient evaluation
VR: Variance Reduction

② **Hessian-Free Bilevel Optimization:** To avoid the expensive Hessian evaluations, several Hessian-free bilevel algorithms have been proposed. For example, FO-MAML (Finn et al., 2017; Nichol et al., 2018) ignores the Hessian computation but does not offer any performance guarantee (Antoniou et al., 2018; Fallah et al., 2020). Several approaches have also been proposed to replace the LL problem with optimality-based constraints (Chen et al., 2023b; Liu et al., 2022a; Shen & Chen, 2023). However, these methods mostly focus on deterministic settings rather than stochastic ones. Several zeroth-order

methods have been proposed to approximate the hypergraident (e.g., ES-MAML (Song et al., 2019), HOZOG (Gu et al., 2021), and PZOBO (Sow et al., 2022)). However, ES-MAML and HOZOG do not provide any theoretical convergence guarantee, while PZOBO achieves $\mathcal{O}\left(d^2\epsilon^{-2}\right)$ to reach an $\epsilon$-stationary point, where $d$ is the problem dimension. Recently, F$^2$SA and F$^3$SA (momentum-based version of F$^2$SA) (Kwon et al., 2023) have been proposed, which are two first-order methods based on the value-function-based lower-level problem reformulation. To reach an $\epsilon$-stationary point, F$^2$SA and F$^3$SA require $\mathcal{O}\left(\epsilon^{-3.5}\right)$ and $\mathcal{O}\left(\epsilon^{-2.5}\right)$ iterations, respectively. The work in (Chen et al., 2023a) improves the convergence rate for F$^2$SA, resulting in a rate of $\mathcal{O}\left(\epsilon^{-2}\log(1/\epsilon)\right)$. However, this improved rate is still slower than that of our proposed LazyBLO approach by a logarithmic factor. Compared to (Kwon et al., 2023), our proposed LazyBLO algorithm strikes a good balance in terms of the use of Hessian information: On one hand, we leverage Hessian information to maintain good convergence performance; on the other hand, we infrequently use Hessian information to signficantly reduce the wall-clock time.

③ **Other Uses of Infrequent Hessian Evaluations:** We note that infrequent Hessian evaluations have also been used for speeding up second-order methods for single-level optimization problems (Shamanskii, 1967; Adler et al., 2020; Doikov et al., 2022; Lampariello & Sciandrone, 2001; Wang et al., 2006; Fan, 2013). However, in bilevel optimization, the Hessian information *necessarily* emerges due to the hypergradient computation, rather than as a "second-order" option in single-level optimization. Also, due to the complex problem structure, analyzing the use of infrequent Hessian in bilevel optimization is far more challenging than in a single-level setting.

## 3 PRELIMINARIES

In this section, we provide some preliminaries for solving Problem (1). We first state a set of assumptions that are needed to establish the convergence of LazyBLO:

**Assumption 3.1** (UL Objective). $f(\mathbf{x}, \mathbf{y})$ satisfies:

1) For any $(\mathbf{x}, \mathbf{y}) \in \mathbb{R}^u \times \mathbb{R}^l$, $\nabla_{\mathbf{x}} f(\mathbf{x}, \mathbf{y})$ is Lipschitz continuous (w.r.t. $\mathbf{y}$) with constant $L_{f_x} \geq 0$, and $\nabla_{\mathbf{y}} f(\mathbf{x}, \mathbf{y})$ is Lipschitz continuous (w.r.t. both $\mathbf{x}$ and $\mathbf{y}$) with constant $L_{f_y} \geq 0$.

2) For any $(\mathbf{x}, \mathbf{y}) \in \mathbb{R}^u \times \mathbb{R}^l$, we have $\|\nabla_{\mathbf{y}} f(\mathbf{x}, \mathbf{y})\| \leq B_{f_y}$ for some constant $B_{f_y} \geq 0$.

**Assumption 3.2** (LL Objective). $g(\mathbf{x}, \mathbf{y})$ satisfies:

1) For any $\mathbf{x} \in \mathbb{R}^u$, $g(\mathbf{x}, \cdot)$ is $\mu_g$-strongly convex with respect to $\mathbf{y}$ for some $\mu_g > 0$.

2) For any $(\mathbf{x}, \mathbf{y}) \in \mathbb{R}^u \times \mathbb{R}^l$, $\nabla_{\mathbf{y}} g(\mathbf{x}, \mathbf{y})$ is Lipschitz continuous (w.r.t. $\mathbf{y}$) with constant $L_g \geq 0$, and $\nabla^2_{\mathbf{xy}} g(\mathbf{x}, \mathbf{y})$ and $\nabla^2_{\mathbf{yy}} g(\mathbf{x}, \mathbf{y})$ are Lipschitz continuous (w.r.t. both $\mathbf{x}$ and $\mathbf{y}$) with constants $L_{g_{xy}} \geq 0$ and $L_{g_{yy}} \geq 0$, respectively.

3) For any $(\mathbf{x}, \mathbf{y}) \in \mathbb{R}^u \times \mathbb{R}^l$, we have $\left\|\nabla^2_{\mathbf{xy}} g(\mathbf{x}, \mathbf{y})\right\| \leq B_{g_{xy}}$ for some constant $B_{g_{xy}} > 0$.

Note that all the above assumptions are standard in the analysis of bilevel optimization problems (e.g., Ghadimi & Wang (2018); Hong et al. (2020); Khanduri et al. (2021b); Liu et al. (2022b); Qiu et al. (2022)). With the above assumptions and using implicit function theorem (Rudin et al., 1976), the hypergradient of $\ell(\cdot)$ can be computed as $\nabla\ell(\mathbf{x}) = \nabla_{\mathbf{x}} f(\mathbf{x}, \mathbf{y}^*(\mathbf{x})) - \nabla^2_{\mathbf{xy}} g(\mathbf{x}, \mathbf{y}^*(\mathbf{x})) \left[\nabla^2_{\mathbf{yy}} g(\mathbf{x}, \mathbf{y}^*(\mathbf{x}))\right]^{-1} \nabla_{\mathbf{y}} f(\mathbf{x}, \mathbf{y}^*(\mathbf{x}))$.

Instead of computing the Hessian inverse explicitly, there are different ways to approximate the Hessian inverse or HVPs in bilevel optimization, such as conjugate gradient (CG) (Pedregosa, 2016) and Neumann series (Ghadimi & Wang, 2018) methods. For example, stocBiO (Ji et al., 2021) uses Neumann series, while AID-BiO (Ji et al., 2021), AID-CG (Grazzi et al., 2020) and AmIGO (Arbel & Mairal, 2022) implement CG. In this paper, we use CG to efficiently estimate the HVPs ($\left[\nabla^2_{\mathbf{yy}} g(\mathbf{x}, \mathbf{y}^*(\mathbf{x}))\right]^{-1} \nabla_{\mathbf{y}} f(\mathbf{x}, \mathbf{y}^*(\mathbf{x}))$), which finds the minimizer of a quadratic function by solving a linear system derived from the hypergradient. The quadratic optimization problem is formulated as follows:

$$\min_{\mathbf{z} \in \mathbb{R}^l} q(\mathbf{x}, \mathbf{y}, \mathbf{z}) \triangleq \frac{1}{2}\mathbf{z}^\top \nabla^2_{\mathbf{yy}} g(\mathbf{x}, \mathbf{y})\mathbf{z} + \mathbf{z}^\top \nabla_{\mathbf{y}} f(\mathbf{x}, \mathbf{y}). \tag{3}$$

For the function $q(\cdot, \cdot, \cdot)$ defined in Eq. (3), the following lemma together with Assumption 3.2 implies that $q(\mathbf{x}, \mathbf{y}, \mathbf{z})$ is $\mu_g$-strongly convex and $L_q$-Lipschitz smooth.

**Lemma 3.3** (Quadratic Problem). *For any* $(\mathbf{x}, \mathbf{y}, \mathbf{z})$, *the quadratic problem* $q(\mathbf{x}, \mathbf{y}, \mathbf{z})$ *with respect to* $\mathbf{z}$ *is Lipschitz-smooth with constant* $L_q \geq 0$.

The admitted unique minimizer $\mathbf{z}^*(\mathbf{x}, \mathbf{y})$ of Eq. (3) can then be utilized to compute the hypergradient estimate as $\nabla \ell(\mathbf{x}) = \nabla_{\mathbf{x}} f(\mathbf{x}, \mathbf{y}^*(\mathbf{x})) + \nabla_{\mathbf{xy}}^2 g(\mathbf{x}, \mathbf{y}^*(\mathbf{x})) \mathbf{z}^*(\mathbf{x}, \mathbf{y}^*(\mathbf{x}))$. Since it is challenging to obtain $\mathbf{y}^*(\mathbf{x})$ and $\mathbf{z}^*(\mathbf{x}, \mathbf{y})$ in closed form, it is natural to consider their approximations. Specifically, let $\bar{\mathbf{y}}$ and $\bar{\mathbf{z}}$ be some approximations of $\mathbf{y}^*(\mathbf{x})$ and $\mathbf{z}^*(\mathbf{x}, \mathbf{y})$, respectively. Then, we have the approximation for $\nabla \ell(\mathbf{x})$ defined as follows:

$$\nabla f(\mathbf{x}, \bar{\mathbf{y}}, \bar{\mathbf{z}}) = \nabla_{\mathbf{x}} f(\mathbf{x}, \bar{\mathbf{y}}) + \nabla_{\mathbf{xy}}^2 g(\mathbf{x}, \bar{\mathbf{y}}) \bar{\mathbf{z}}. \tag{4}$$

Since Problem (1) can potentially be a large-scale stochastic optimization problem, computing a full gradient approximation in Eq. (4) can be computationally expensive. To address this challenge, a common approach for evaluating Eq. (4) is to build a stochastic gradient estimator. Define stochastic approximations as $f(\mathbf{x}, \mathbf{y}; \mathcal{D}^f) \triangleq \frac{1}{|\mathcal{D}^f|} \sum_{\xi \in \mathcal{D}^f} f(\mathbf{x}, \mathbf{y}; \xi)$ and $g(\mathbf{x}, \mathbf{y}; \mathcal{D}^g) \triangleq \frac{1}{|\mathcal{D}^g|} \sum_{\zeta \in \mathcal{D}^g} g(\mathbf{x}, \mathbf{y}; \zeta)$, where $\mathcal{D}^f$ and $\mathcal{D}^g$ are the batches of independent and identically distributed samples with sizes $|\mathcal{D}^f| \geq 1$ and $|\mathcal{D}^g| \geq 1$, respectively. Then, a stochastic estimator of Eq. (4) can be computed as:

$$\nabla f(\mathbf{x}, \mathbf{y}, \mathbf{z}; \mathcal{D}^{f_x}, \mathcal{D}^{g_{xy}}) = \nabla_{\mathbf{x}} f(\mathbf{x}, \mathbf{y}; \mathcal{D}^{f_x}) + \nabla_{\mathbf{xy}}^2 g(\mathbf{x}, \mathbf{y}; \mathcal{D}^{g_{xy}}) \mathbf{z}.$$

Here, for simplicity, we slightly abuse the notations $\bar{\mathbf{y}}$ and $\bar{\mathbf{z}}$ as $\mathbf{y}$ and $\mathbf{z}$ in the above equation and the rest of the paper as long as there is no confusion from the context. For $\nabla f(\mathbf{x}, \mathbf{y}, \mathbf{z}; \mathcal{D}^{f_x}, \mathcal{D}^{g_{xy}})$ and $\nabla_{\mathbf{y}} g(\mathbf{x}, \mathbf{y}; \mathcal{D}^{g_y})$, we make the following typical assumption in stochastic optimization analysis.

**Assumption 3.4** (Stochastic Gradients). For any $(\mathbf{x}, \mathbf{y}) \in \mathbb{R}^u \times \mathbb{R}^l$ and data batch $\mathcal{D}^{f_x}$, $\mathcal{D}^{f_y}$, $\mathcal{D}^{g_y}$, $\mathcal{D}^{g_{xy}}$ and $\mathcal{D}^{g_{yy}}$, define $\sigma_{f_x}^2 \triangleq \tilde{\sigma}_{f_x}^2 |\mathcal{D}^{f_x}|^{-1}$, $\sigma_{f_y}^2 \triangleq \tilde{\sigma}_{f_y}^2 |\mathcal{D}^{f_y}|^{-1}$, $\sigma_{g_y}^2 \triangleq \tilde{\sigma}_{g_y}^2 |\mathcal{D}^{g_y}|^{-1}$, $\sigma_{g_{xy}}^2 \triangleq \tilde{\sigma}_{g_{xy}}^2 |\mathcal{D}^{g_{xy}}|^{-1}$, and $\sigma_{g_{yy}}^2 \triangleq \tilde{\sigma}_{g_{yy}}^2 |\mathcal{D}^{g_{yy}}|^{-1}$, where $\tilde{\sigma}_{f_x}^2, \tilde{\sigma}_{f_y}^2, \tilde{\sigma}_{g_y}^2, \tilde{\sigma}_{g_{xy}}^2$ and $\tilde{\sigma}_{g_{yy}}^2$ represent the variance of a single sample of the corresponding functions. The gradient estimates $\nabla_{\mathbf{x}} f(\mathbf{x}, \mathbf{y}; \mathcal{D}^{f_x})$, $\nabla_{\mathbf{y}} f(\mathbf{x}, \mathbf{y}; \mathcal{D}^{f_y})$, $\nabla_{\mathbf{y}} g(\mathbf{x}, \mathbf{y}; \mathcal{D}^{g_y})$, $\nabla_{\mathbf{xy}}^2 g(\mathbf{x}, \mathbf{y}; \mathcal{D}^{g_{xy}})$ and $\nabla_{\mathbf{yy}}^2 g(\mathbf{x}, \mathbf{y}; \mathcal{D}^{g_{yy}})$ are unbiased and have bounded variances:

$$\mathbb{E}[\|\nabla_{\mathbf{x}} f(\mathbf{x}, \mathbf{y}; \mathcal{D}^{f_x}) - \nabla_{\mathbf{y}} f(\mathbf{x}, \mathbf{y})\|^2] \leq \sigma_{f_x}^2, \qquad \mathbb{E}[\|\nabla_{\mathbf{y}} f(\mathbf{x}, \mathbf{y}; \mathcal{D}^{f_y}) - \nabla_{\mathbf{y}} f(\mathbf{x}, \mathbf{y})\|^2] \leq \sigma_{f_y}^2,$$

$$\mathbb{E}[\|\nabla_{\mathbf{y}} g(\mathbf{x}, \mathbf{y}; \mathcal{D}^{g_y}) - \nabla_{\mathbf{y}} g(\mathbf{x}, \mathbf{y})\|^2] \leq \sigma_{g_y}^2, \qquad \mathbb{E}[\|\nabla_{\mathbf{xy}}^2 g(\mathbf{x}, \mathbf{y}; \mathcal{D}^{g_{xy}}) - \nabla_{\mathbf{xy}}^2 g(\mathbf{x}, \mathbf{y})\|^2] \leq \sigma_{g_{xy}}^2,$$

$$\mathbb{E}[\|\nabla_{\mathbf{yy}}^2 g(\mathbf{x}, \mathbf{y}; \mathcal{D}^{g_{yy}}) - \nabla_{\mathbf{yy}}^2 g(\mathbf{x}, \mathbf{y})\|^2] \leq \sigma_{g_{yy}}^2.$$

Lastly, we define the following performance metrics for solving the Problem (1).

**Definition 3.5** ($\epsilon$-Stationary Point). Point $\mathbf{x}$ is an $\epsilon$-stationary point if $\mathbb{E}\left[\|\nabla \ell(\mathbf{x})\|^2\right] \leq \epsilon$, where $\mathbf{x}$ is the output of a stochastic algorithm, and the expectation is taken over all randomness of the algorithm.

**Definition 3.6** ($\epsilon$-Optimal Point). Point $\mathbf{x}$ is an $\epsilon$-optimal point if $\mathbb{E}\left[\ell(\mathbf{x}) - \ell^*\right] \leq \epsilon$, where $\ell^* \triangleq \min_{\mathbf{x} \in \mathbb{R}^u} \ell(\mathbf{x})$, and $\mathbf{x}$ is the output of a stochastic algorithm. The expectation is taken over all randomness of the algorithm.

# 4 THE LazyBLO ALGORITHM

In this section, we propose a new algorithmic framework called LazyBLO to solve the bilevel optimization problem in Eq. (1). Our goal is to reduce the computation of the HVPs, and our key idea is to update the HVP periodically on a subset of the entire training iterations while using stale Hessian information in the remaining iterations.

The most basic algorithm in the LazyBLO framework incorporates SGD-style updates, which is illustrated in Algorithm 1. We note that more sophisticated algorithms in the LazyBLO framework can include advanced algorithmic techniques, such as momentum and/or variance reduction to accelerate the convergence and enhance other performances. As shown in Algorithm 1, the LazyBLO framework uses a double-loop structure and constructs iterates $\mathbf{x}_t^n$, $\mathbf{y}_t^n$ and $\mathbf{z}_t$, where the inner iteration counter $n$ goes from 0 to $N - 1$ and the outer iteration counter $t$ runs from 0 to $T - 1$, so that $\mathbf{x}_t^n$ approaches a stationary point of $\ell(\cdot)$, and $\mathbf{y}_t^n$ and $\mathbf{z}_t$ keep track of the quantities $\mathbf{y}^*(\mathbf{x}_t^n)$ and $\mathbf{z}^*(\mathbf{x}_t^N, \mathbf{y}_t^N)$. In

---

**Algorithm 1** The LazyBLO Algorithmic Framwork with Basic SGD-type Updates.

---

**Input:** Initial parameters $\mathbf{x}_0^0, \mathbf{y}_0^0, \mathbf{z}_0$, and stepsize $\{\alpha_t\}_{t=0}^{T-1}, \{\beta_t\}_{t=0}^{T-1}, \{\gamma_t\}_{t=0}^{T-1}$
**for** $t = 0$ **to** $T - 1$ **do**
    **for** $n = 0$ **to** $N - 1$ **do**
        Initialize $\mathbf{x}_t^0 = \mathbf{x}_{t-1}^N$ and $\mathbf{y}_t^0 = \mathbf{y}_{t-1}^N$
        Sample data batches $\mathcal{D}_{t,n}^g, \mathcal{D}_{t,n}^{f_x}$, and $\mathcal{D}_{t,n}^{g_{xy}}$
        Compute the gradient estimate $h_{t,n}^g$ using (6) and update $\mathbf{y}_t^{n+1} = \mathbf{y}_t^n - \beta_t h_{t,n}^g$
        Compute the gradient estimate $h_{t,n}^f$ using (5) and update $\mathbf{x}_t^{n+1} = \mathbf{x}_t^n - \alpha_t h_{t,n}^f$
    **end for**
    Sample data batches $\mathcal{D}_t^{g_{yy}}$ and $\mathcal{D}_t^{f_y}$
    Compute the gradient estimate $h_{t,n}^q$ using (7) and update $\mathbf{z}_{t+1} = \mathbf{z}_t - \gamma_t h_t^q$
**end for**

---

the inner loop, the algorithm updates $\mathbf{x}_t^n$ and $\mathbf{y}_t^n$ using the stochastic gradient estimators $h_{t,n}^f$ and $h_{t,n}^g$ defined as:

$$h_{t,n}^f = \nabla_{\mathbf{x}} f\left(\mathbf{x}_t^n, \mathbf{y}_t^n; \mathcal{D}_{t,n}^{f_x}\right) + \nabla_{\mathbf{xy}}^2 g\left(\mathbf{x}_t^n, \mathbf{y}_t^n; \mathcal{D}_{t,n}^{g_{xy}}\right)\mathbf{z}_t, \tag{5}$$

$$h_{t,n}^g = \nabla_{\mathbf{y}} g\left(\mathbf{x}_t^n, \mathbf{y}_t^n; \mathcal{D}_{t,n}^g\right). \tag{6}$$

The variable $\mathbf{z}_t$ in Eq. (5) is updated in the outer loop using a stochastic gradient estimator $h_t^q$ as:

$$h_t^q = \nabla_{\mathbf{yy}}^2 g(\mathbf{x}_t^N, \mathbf{y}_t^N; \mathcal{D}_t^{g_{yy}})\mathbf{z}_t + \nabla_{\mathbf{y}} f(\mathbf{x}_t^N, \mathbf{y}_t^N; \mathcal{D}_t^{f_y}). \tag{7}$$

Note that, compared to $h_{t,n}^f$ and $h_{t,n}^g$, only $h_t^q$ contains the HVP, and is computed *infrequently* after every $N$ inner loop iterations. In addition, $N$ needs to be chosen with a tolerable approximation error of the HVP. If $N$ gets too large, the error of the HVP approximation would also increase, thus inevitably degrading the performance of LazyBLO. With less frequent Hessian computations, LazyBLO executes faster per iteration in terms of *wall-clock time* compared to standard bilevel algorithms that require multiple Hessian/vector evaluations in each round of updates (Ghadimi & Wang, 2018; Arbel & Mairal, 2022; Ji et al., 2021; Chen et al., 2021), resulting in a significant reduction in computational cost and savings in implementation time.

Another insightful remark on the Jacobian-vector product in (5) is also in order. To date, most of the existing bilevel algorithms compute only one single Jacobian-vector product (JVP) in each iteration, whereas HVPs are computed multiple times in each iteration even in some single-loop bilevel algorithms (e.g., SUSTAIN (Khanduri et al., 2021b), TTSA (Hong et al., 2020), BSA (Ghadimi & Wang, 2018), and ALSET (Chen et al., 2021)). Due to this difference between JVP and HVP in bilevel optimization algorithms, reducing the number of HVP computations is far more important than reducing the computations of JVPs. Therefore, we only focus on reducing the HVP in this paper. We further note that reducing the computation of JVPs can be done in a similar manner as the HVPs established in our work.

## 5 THEORETICAL PERFORMANCE ANALYSIS

In this section, we conduct the theoretical convergence analysis for the LazyBLO framework for solving the bilevel optimization problem in Eq. (1). Note that, although LazyBLO executes faster per iteration, we have a noisier hypergradient due to the use of stale Hessian information. As a result, it remains unclear whether LazyBLO can converge and, if yes, what theoretical convergence rate (i.e., iteration complexity) it can achieve. Intuitively, due to the lazy Hessian information updates, one can expect that the theoretical convergence rate of LazyBLO cannot outperform its non-lazy counterpart. Surprisingly, in this paper, we show that LazyBLO achieves the same convergence rate as their non-lazy counterpart. This, together with the much lower per-iteration wall-clock time, implies that LazyBLO will enjoy a much faster speed in terms of wall-clock time. This will also be verified by our experiments in Section 6.

The convergence analysis for LazyBLO is highly non-trivial due to the following technical challenges: *i)* The use of lazy Hessian evaluation increases the error of stochastic gradient estimator $h_{t,n}^f$ for the upper-level function; *ii)* Due to the hierarchical and coupled structure of bilevel optimization problems, the error resulting from the stochastic gradient estimator $h_{t,n}^f$ with stale Hessian information

further propagates to and increases the approximation error of $\mathbf{y}^*(\mathbf{x})$ and the approximation error of $\mathbf{z}^*(\mathbf{x}, \mathbf{y})$. What is even worse is that $\mathbf{z}^*(\mathbf{x}, \mathbf{y})$ is also associated with $\mathbf{y}^*(\mathbf{x})$. All the above complex couplings of *laziness-induced errors* and the complications associated with these approximation errors are *unseen* in bilevel optimization algorithm analysis, which significantly increases the difficulty of analyzing the convergence of LazyBLO and necessitate *new* proof techniques.

## 5.1 SUPPORTING LEMMAS

Toward this end, we first state two basic lemmas needed for the convergence analysis of LazyBLO.

**Lemma 5.1** (Lemma 2.2 in (Ghadimi & Wang, 2018), Proposition 6 in (Arbel & Mairal, 2022))**.** *Under Assumptions 3.1 and 3.2, we have*

$$\|\nabla f(\mathbf{x}, \bar{\mathbf{y}}, \bar{\mathbf{z}}) - \nabla \ell(\mathbf{x})\| \le L_f \left( \|\bar{\mathbf{y}} - \mathbf{y}^*(\mathbf{x})\| + \|\bar{\mathbf{z}} - \mathbf{z}^*(\mathbf{x}, \bar{\mathbf{y}})\| \right),$$
$$\|\mathbf{y}^*(\mathbf{x}_1) - \mathbf{y}^*(\mathbf{x}_2)\| \le L_y \|\mathbf{x}_1 - \mathbf{x}_2\|, \qquad \|\nabla \ell(\mathbf{x}_1) - \nabla \ell(\mathbf{x}_2)\| \le L_l \|\mathbf{x}_1 - \mathbf{x}_2\|,$$

*for all $\mathbf{x}, \mathbf{x}_1, \mathbf{x}_2 \in \mathbb{R}^u$, and $\bar{\mathbf{y}}, \bar{\mathbf{z}} \in \mathbb{R}^l$, where the Lipschitz constants above are defined as:*

$$L_f = \max\left\{ L_{f_x} + \left( L_{g_{xy}} B_{f_y} / \mu_g \right) + B_{g_{xy}} L_z, B_{g_{xy}} \right\}, \quad L_l = L_f' + \left( L_f' B_{g_{xy}}^2 / \mu_g \right), \quad L_y = B_{g_{xy}}^2 / \mu_g,$$

*and where $L_f' = L_{f_x} + \left( L_{f_y} B_{g_{xy}}^2 / \mu_g \right) + B_{f_y} \left[ \left( L_{g_{xy}} / \mu_g \right) + \left( L_{g_{yy}} B_{g_{xy}}^2 / \mu_g^2 \right) \right].$*

We note that Lemma 5.1 plays a key role in the analysis of AID-based bilevel algorithms. First of all, it characterizes the bias of the implicit gradient as a function of approximation error in $\bar{\mathbf{y}}$ and $\bar{\mathbf{z}}$ (see Eq. (4)). It also ensures the Lipschitzness of the mapping $\mathbf{y}^*(\mathbf{x})$ in characterizing the behavior of the LL problem's iterates. Most importantly, Lemma 5.1 establishes the Lipschitz-smoothness of the implicit function $\ell(\cdot)$, which allows the development of SGD-type algorithms for solving stochastic bilevel problems. To complement Lemma 5.1, next result states the properties of the optimal solution $\mathbf{z}^*(\mathbf{x}, \mathbf{y})$ of the quadratic problem in Eq. (3).

**Lemma 5.2** (Proposition 6 in(Arbel & Mairal, 2022))**.** *Under Assumptions 3.1 and 3.2, $\forall \, \mathbf{x}, \mathbf{x}_1, \mathbf{x}_2 \in \mathbb{R}^u$ and $\mathbf{y}, \mathbf{y}_1, \mathbf{y}_2 \in \mathbb{R}^l$, we have*

$$\|\mathbf{z}^*(\mathbf{x}_1, \mathbf{y}_1) - \mathbf{z}^*(\mathbf{x}_2, \mathbf{y}_2)\| \le L_z \left( \|\mathbf{x}_1 - \mathbf{x}_2\| + \|\mathbf{y}_1 - \mathbf{y}_2\| \right), \qquad \|\mathbf{z}^*(\mathbf{x}, \mathbf{y})\| \le B_{f_y} / \mu_g,$$

*where $L_z = \left( L_{g_{yy}} B_{f_y} / \mu_g^2 \right) + L_{f_y} / \mu_g.$*

Lemma 5.2 also plays a key role in the analysis of LazyBLO as it is utilized to bound the drift in the Hessain vector product estimates (see Eq. (3)). Next, we present the main results of the paper.

## 5.2 MAIN RESULTS

① **The Non-convex $\ell(\mathbf{x})$ Setting:** By leveraging Lemmas 5.1 and 5.2, we establish the main convergence result of the proposed LazyBLO for non-convex $\ell(\mathbf{x})$ in Theorem 5.3.

**Theorem 5.3** (Non-Convex $\ell(\mathbf{x})$)**.** *Under Assumptions 3.1–3.4, with step-sizes $\alpha_t = \alpha = \mathcal{O}\left(\frac{1}{N^2}\right)$, $\beta_t \triangleq c_\beta \alpha = \mathcal{O}\left(\frac{1}{N^2}\right)$, and $\gamma_t \triangleq c_\gamma \alpha = \mathcal{O}(1)$ for all $t \in \{0, 1, \ldots, T-1\}$, where $c_\beta$ and $c_\gamma$ are defined in Eq. 22 in Appendix C. Then, the iterates generated by LazyBLO satisfy:*

$$\frac{1}{TN} \sum_{t=0}^{T-1} \sum_{n=0}^{N-1} \mathbb{E}\left[ \|\nabla \ell(\mathbf{x}_t^n)\|^2 \right] = \mathcal{O}\left( \frac{N\Delta_0}{T} \right) + \mathcal{O}\left( \sigma_{g_y}^2 + \sigma_{g_{xy}}^2 + \sigma_{f_x}^2 + \sigma_{g_{yy}}^2 + \sigma_{f_y}^2 \right),$$

*where $\Delta_0 = (\ell(\mathbf{x}_0^0) - \ell^*) + \|\mathbf{y}_0^0 - \mathbf{y}^*(\mathbf{x}_0^0)\|^2 + \|\mathbf{z}_0 - \mathbf{z}^*(\mathbf{x}_0^0, \mathbf{y}_0^0)\|^2.$*

The proof of Theorem 5.3 can be found in Appendix C. Theorem 5.3 establishes the convergence of LazyBLO under the most general setting, where the implicit function $\ell(\cdot)$ can be non-convex. The result characterizes the effect of different parameters on the convergence of LazyBLO. Specifically, as $N$ increases, the performance of LazyBLO degrades. This is unsurprising since more stale Hessian information is expected to slow the convergence. Hence, $N$ should be chosen below a certain threshold to maintain the accuracy of the hypergradient estimations. On the other hand, to enjoy the benefits of the LazyBLO approach, $N$ is supposed to be strictly larger than 1. We can potentially choose $N = 1$, and our algorithm, which becomes fully single-loop, recovers standard results for

bilevel algorithms under the same assumptions as ours (e.g., the guarantees achieved in (Arbel & Mairal, 2022)). Interestingly, under an appropriate $N$-value, the $N$-dependent slowdown effect in LazyBLO can be offset by Hessian computations skippings, allowing LazyBLO to run even faster than non-lazy approaches in terms of wall-clock time.

Our next result characterizes the computation complexity of LazyBLO.

**Corollary 5.4** (Computation Complexity). *Under the setting of Theorem 5.3, choose* $\left|\mathcal{D}^{f_x}\right|, \left|\mathcal{D}^{f_y}\right|, \left|\mathcal{D}^{g_y}\right|, \left|\mathcal{D}^{g_{xy}}\right|, \left|\mathcal{D}^{g_{yy}}\right| = \Theta\left(\epsilon^{-1}\right)$. *Then,* LazyBLO *requires* $\mathcal{O}(\epsilon^{-2})$ *partial gradient and HVP evaluations to reach an $\epsilon$-stationary point.*

We note that, when $\epsilon$ is small, the batch sizes in Corollary 5.4 could be large. However, it is worth noting that the use of large batch sizes is not a consequence of the proposed LazyBLO algorithm design; rather, these batch size choices are common in the literature, as the above guarantees are the same as those achieved in standard SGD-based bilevel algorithms (e.g., (Arbel & Mairal, 2022; Ji et al., 2021; Huang et al., 2022)) that require the computation of (multiple) Hessian/HVPs in each iteration. It is also worth noting that the large batch sizes are required only for theoretical analysis and can be eliminated by using a third-order Lipschitz assumption, as done by SOBA (Dagréou et al., 2022). In our experiments, we use a small batch size instead, and our algorithm still outperforms the baseline algorithms.

Given that LazyBLO can converge and even matches the performance of SOTA non-lazy bilevel methods, another question also arises: under which settings could LazyBLO theoretically outperform current bilevel approaches? The next result shows that if the LL problem is deterministic, we can, in fact, improve upon the current approaches and reduce the HVP evaluations from $\mathcal{O}(\epsilon^{-2})$ to $\mathcal{O}(\epsilon^{-1})$.

**Corollary 5.5** (Computation Complexity for Deterministic LL Problems). *Suppose the lower-level problem is deterministic. Under the condition of Theorem 5.3,* LazyBLO *requires* $\mathcal{O}(\epsilon^{-1})$ *for HVP evaluations to achieve an $\epsilon$-stationary point.*

Corollary 5.5 suggests that LazyBLO significantly reduces the HVP evaluations. In contrast, for standard bilevel optimization algorithms, the HVPs stay the same as the total number of rounds required by an algorithm to achieve the $\epsilon$-stationary solution. For example, under the same deterministic setting, the baseline methods BSA (Ghadimi & Wang, 2018), stocBiO (Ji et al., 2021) and AmIGO (Arbel & Mairal, 2022) require $\mathcal{O}(\epsilon^{-2})$ gradient computations and TTSA (Hong et al., 2020) requires $\mathcal{O}(\epsilon^{-2.5})$ gradient computations, which is equivalent to the number of outer function's gradients evaluated during the execution of the algorithm.

So far, our results characterize the performance of LazyBLO in the non-convex settings. However, for some problems (e.g., quadratic UL and LL problems), the implicit function may have additional structures that might lead to better convergence of LazyBLO. Next, we characterize the performance of LazyBLO when the implicit function is strongly convex, which is often of interest for applications in robust and inverse optimization, optimal control in robotics and aerospace with quadratic cost, etc.

② **The Strongly Convex $\ell\left(\mathbf{x}\right)$ Setting:** Under the setting where $\ell\left(\cdot\right)$ is $\mu_f$-strongly convex, we provide a stronger performance guarantee for the convergence of LazyBLO, which is stated as follows:

**Theorem 5.6** (Strongly Convex $\ell\left(\mathbf{x}\right)$). *Suppose the upper-level function $\ell\left(\mathbf{x}\right)$ is $\mu_f$-strongly-convex. Under Assumptions 3.1–3.4, choose the step-sizes $\alpha_t = \alpha = \mathcal{O}\left(\frac{1}{N}\right)$, $\beta_t \triangleq \hat{c}_\beta\alpha = \mathcal{O}\left(\frac{1}{N}\right)$ and $\gamma_t \triangleq \hat{c}_\gamma\alpha = \mathcal{O}\left(\frac{1}{N^2}\right)$ for all $t \in \{0, 1, \dots, T-1\}$, where $\hat{c}_\beta$ and $\hat{c}_\gamma$ are defined in Eq. 34 in Appendix D. Then, the iterates generated by* LazyBLO *satisfy:*

$$\frac{1}{N}\sum_{n=0}^{N-1}\mathbb{E}\left[\ell\left(\mathbf{x}_t^n\right) - \ell^*\right] \leq (1 - \mu_f\alpha)^t\,\hat{\Delta}_0 + \mathcal{O}\left(\sigma_{g_{xy}}^2 + \sigma_{f_x}^2 + \frac{1}{N^4}\sigma_{g_{yy}}^2 + \frac{1}{N^4}\sigma_{f_y}^2 + \frac{1}{N}\sigma_{g_y}^2\right),$$

*for any $t \geq 1$, where $\hat{\Delta}_0 = \frac{1}{N}\sum_{n=0}^{N-1}\left(\ell\left(\mathbf{x}_0^n\right) - \ell^*\right) + \frac{1}{N}\sum_{n=0}^{N-1}\|\mathbf{y}_0^n - \mathbf{y}^*\left(\mathbf{x}_0^n\right)\|^2 + \frac{1}{N}\|\mathbf{z}_0 - \mathbf{z}_0^*\|^2$.*

The detailed proof of Theorem 5.6 is provided in Appendix D due to space limitations. Theorem 5.6 demonstrates that, under the strongly convex setting, LazyBLO achieves a much faster linear convergence rate. Theorem 5.6 also immediately implies the following computation complexity:

**Corollary 5.7** (Computation Complexity). *Under the setting of Theorem 5.6, choosing $\left|\mathcal{D}^{f_x}\right| = \left|\mathcal{D}^{g_{xy}}\right| = \Theta\left(\epsilon^{-1}\right)$, $\left|\mathcal{D}^{g_y}\right| = \Theta\left(N^{-1}\epsilon^{-1}\right)$, and $\left|\mathcal{D}^{f_y}\right| = \left|\mathcal{D}^{g_{yy}}\right| = \Theta\left(N^{-4}\epsilon^{-1}\right)$,* LazyBLO *requires*

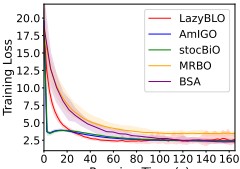 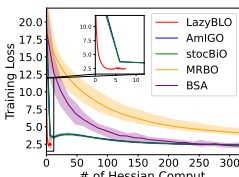 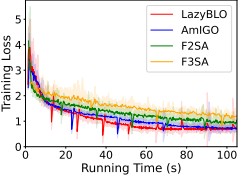 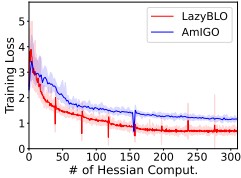

(a) Wall-clock time.   (b) # of Hessian comput.        (a) Wall-clock time.   (b) # of Hessian comput.

Figure 1: Comparison for data hyper-cleaning on MNIST (corruption rate $p = 0.1$, 10 repetitions).

Figure 2: Training loss for deep hyper-representation on CIFAR-10 (10 repetitions).

*$\mathcal{O}(\epsilon^{-1}\log\epsilon^{-1})$ partial gradient evaluations and $\mathcal{O}(N^{-4}\epsilon^{-1}\log\epsilon^{-1})$ HVP evaluations to reach an $\epsilon$-optimal point.*

Corollary 5.7 shows that LazyBLO significantly reduces the number of HVP evaluations. Again, note that the complexity of partial gradient evaluations in Corollary 5.7 matches the same guarantee achieved in (Arbel & Mairal, 2022), which is obtained by multiple Hessian evaluations per iteration.

## 6 NUMERICAL RESULTS

In this section, we verify the theoretical performance of LazyBLO on different optimization tasks and with two different datasets: 1) data hyper-cleaning on the MNIST dataset, and 2) deep hyper-representation with the ResNet network on the CIFAR-10 dataset. Due to space limitations, additional experimental details and results are included in Appendix B.

**Task 1) Data Hyper-Cleaning on the MNIST Dataset:** We conduct experiments on a data hyper-cleaning task with MNIST dataset (LeCun et al., 1998). Data hyper-cleaning aims to train a classifier on a corrupted dataset. We compare LazyBLO with stochastic bilevel algorithms AmIGO (Arbel & Mairal, 2022), stocBiO (Ji et al., 2021), BSA (Ghadimi & Wang, 2018), and MRBO (Yang et al., 2021) as baselines. We also perform data hyper-cleaning with fully single-loop bilevel algorithms TTSA (Hong et al., 2020), SOBA (Dagréou et al., 2022), and MA-SOBA (Chen et al., 2024).

Table 2 shows that TTSA, SOBA, and MA-SOBA all need an exceedingly long time to converge. Specifically, the convergence of TTSA, SOBA, and MA-SOBA are 74, 73, and 82× slower, respectively, than LazyBLO. In addition, TTSA, SOBA, and MA-SOBA require 390, 126, and 150× more Hessian computations, respectively, compared to LazyBLO. Given the significantly slow convergence of these fully single-loop bilevel algorithms, we exclude them from the following comparison.

From Fig. 1a, we can see that AmIGO and stocBiO have similar convergence performance. LazyBLO outperforms all baseline methods in terms of wall-clock time, which shows the

Table 2: Convergence performance of TTSA, SOBA, and MA-SOBA compared with our LazyBLO on data hyper-cleaning on MNIST (corruption rate $p = 0.1$, average over 10 repetitions).

| ALGORITHM | WALL-CLOCK TIME | # OF HESSIAN | TRAINING LOSS |
|---|---|---|---|
| TTSA | 4290 s | 1950 | 3.95 |
| SOBA | 4210 s | 630 | 3.28 |
| MA-SOBA | 4740 s | 750 | 3.05 |
| **LazyBLO** | **58 s** | **5** | **2.35** |

advantages of LazyBLO. Specifically, it only takes LazyBLO approximately 60 seconds to converge, while AmIGO and stocBiO converge in around 100 seconds. This much-improved wall-clock time is due to the fact that LazyBLO uses stale Hessian information and saves a lot of Hessian computation time. It is worth pointing out that the comparison with MRBO is not entirely fair since MRBO is equipped with more sophisticated momentum techniques to accelerate convergence, while LazyBLO *only* uses vanilla-SGD updates. LazyBLO can also be equipped with momentum-based SGD updates to further accelerate the convergence. Furthermore, the training loss of LazyBLO is similar to those of AmIGO, stocBiO, and BSA, which use up-to-date Hessian information during the training. This result is surprising because LazyBLO with stale Hessian information can still *match* the methods with non-lazy Hessian updates. This implies that the Hessian information evolves gradually during the training, and one may use stale Hessians to construct good approximations of the hypergradient in bilevel optimization.

It can be seen in Fig. 1b that the convergence speed with respect to the number of Hessian evaluations for LazyBLO is much faster compared with all the baseline algorithms (see the zoomed-in area in

Fig. 1b). Table 3 also demonstrates that, to achieve the same convergence training loss, AmIGO, stocBiO and BSA all need 252 Hessian computations, while LazyBLO only needs 5 Hessian computations (i.e., $50\times$ faster). Note that we do not include MRBO in this table since it has a higher error floor compared to other algorithms. As a consequence, it can not reach the same training loss as the other algorithms.

Fig. 3 captures the effect of $N$ on the performance of LazyBLO. Specifically, we observe that as we increase the value of $N$, the execution of the algorithm becomes faster and faster. However, we note that increasing $N$ beyond a certain threshold may not yield additional benefits and could even lead to performance degradation. This is because, as $N$ increases, the difference between stale and fresh Hessian information becomes larger, potentially causing the hypergradient $h_{t,n}^f$ to become less accurate and adversely affecting the training loss of LazyBLO.

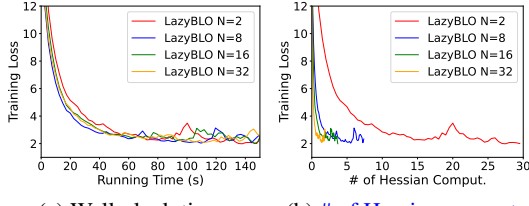

(a) Wall-clock time.  (b) # of Hessian comput.

Figure 3: Comparison of LazyBLO on data hyper-cleaning on MNIST at a different # of x-updates ($N$).

**Task 2) Deep Hyper-Representation with ResNet-20 on the CIFAR-10 Dataset:** To demonstrate the effectiveness of LazyBLO in training neural networks, we conduct experiments on a deep hyper-representation task (Yang et al., 2023; Sow et al., 2022) with the ResNet-20 model (He et al., 2016) on CIFAR-10 dataset (Krizhevsky et al., 2009), which aims to classify CIFAR-10 images. We compare LazyBLO with a standard stochastic bilevel algorithm AmIGO (Arbel & Mairal, 2022), and two fully first-order (Hessian/Jacobian-free) stochastic bilevel algorithms F$^2$SA (Kwon et al., 2023) and F$^3$SA (Kwon et al., 2023) as baselines. We do not compare LazyBLO with other baselines from the previous data hyper-cleaning experiments since stocBiO performs almost identically to AmIGO, and they both outperform MRBO in terms of training loss and BSA in terms of wall-clock time.

As shown in Fig. 2a, LazyBLO converges faster in terms of wall-clock time compared to AmIGO, F$^2$SA and F$^3$SA. In addition, Fig. 2a indicates that the training loss of LazyBLO is smaller than those of F$^2$SA and F$^3$SA. The superior performance of LazyBLO in comparison to F$^2$SA and F$^3$SA establishes the necessity of Hessian/Jacobian evaluations in stochastic bilevel optimization. Without them, both the convergence speed and the training loss would degrade as demonstrated by the experiments. Fig. 2b illustrates the convergence performance of

Table 3: The number of hypergradient computations and Hessian computations required by various algorithms to achieve the same training loss in data hyper-cleaning experiments (Task 1) and hyper-representation experiments (Task 2) (average over 10 repetitions).

|  | Algorithm | # of HGC | # of Hessian |
|---|---|---|---|
| Task 1 | AmIGO | 42 | 252 |
|  | stocBiO | 42 | 252 |
|  | BSA | 21 | 252 |
|  | **LazyBLO** | **40** | **5** |
| Task 2 | AmIGO | 361 | 722 |
|  | **LazyBLO** | **640** | **320** |

HGC: hypergradient computation

LazyBLO compared to AmIGO in terms of the number of Hessian computations. Note that we do not include F$^2$SA and F$^3$SA in this figure since they are Hessian-free. Fig. 2b demonstrates that with the same number of Hessian evaluations, LazyBLO has a lower training loss compared to AmIGO. Furthermore, as shown in Table 3, to reach the same training loss, LazyBLO uses 320 Hessian computations, while AmIGO uses 722 Hessian computations. This significantly reduces computational costs, especially for large-scale problems.

## 7 CONCLUSION

In this paper, we proposed the LazyBLO algorithmic framework for solving bilevel optimization problems. Compared to existing works, LazyBLO reduces the Hessian-vector product (HVP) evaluations by updating them periodically and less frequently. Although LazyBLO uses stale HVP evaluations that introduce additional errors, our theoretical analysis demonstrated that LazyBLO not only surprisingly enjoys the same convergence rate guarantee, but also achieves a much faster wall-clock time performance. Specifically, to reach an $\epsilon$-stationary point, LazyBLO requires $\mathcal{O}(\epsilon^{-2})$ for both partial gradient evaluations and HVP evaluations, which matches the SOTA non-lazy methods. We conducted experiments on multi-hyperparameter optimization tasks to verify our theoretical findings.

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

## A    ADDITIONAL RELATED WORK

**Bilevel Optimization:** The history of bilevel optimization dates back to 1973 (Bracken & McGill, 1973). Some early attempts for solving bilevel problems include: value function (Liu et al., 2021; Sinha et al., 2018; Zemkoho & Zhou, 2021), Karush–Kuhn–Tucker conditions based reformulations (Allende & Still, 2013; Sinha et al., 2019; Zemkoho & Zhou, 2021), penalty function (White & Anandalingam, 1993; Anandalingam & White, 1990; Wan et al., 2014), approximate descent (Falk & Liu, 1995; Vicente et al., 1994), and trust region methods (Dempe & Bard, 2001; El-Sobky & Abo-Elnaga, 2018). Among these approaches, approximate descent methods have gained prominence recently because of their ease of implementation as well as strong theoretical and empirical performance in many machine learning applications. Two standard descent-based approaches to tackle problems of form (1) are iterative differentiation (ITD) (Domke, 2012; Maclaurin et al., 2015; Franceschi et al., 2017; 2018; Shaban et al., 2019; Grazzi et al., 2020; MacKay et al., 2019) and approximate implicit differentiation (AID) (Domke, 2012; Pedregosa, 2016; Liao et al., 2018; Ghadimi & Wang, 2018; Grazzi et al., 2020; Lorraine et al., 2020; Gould et al., 2016; Ji & Liang, 2021; MacKay et al., 2019; Khanduri et al., 2021a; Hong et al., 2020). The basic idea of ITD is to obtain an approximate hypergradient of the loss function $\ell(\mathbf{x})$ in Eq. (1) by differentiating the unrolled iterates of the LL problem. Consequently, ITD-based approaches need to store all the LL iterates in the memory (Shaban et al., 2019). On the other hand, AID relies on the implicit function theorem to compute the implicit gradient of $\ell(\mathbf{x})$ without the need to maintain the sequence of LL iterates. Instead of differentiating the iterates of the LL problem, AID computes the implicit gradient by approximately solving a linear system of equations using HVPs. In this work, we focus on AID-based approaches for solving stochastic bilevel problems.

## B    ADDITIONAL EXPERIMENTAL DETAILS AND RESULTS

### B.1    SPECIFICATIONS OF THE BASELINE ALGORITHMS IN SECTION 6

In this section, we provide more description of the baseline algorithms used in our experiments, as follows:

- AmIGO (Arbel & Mairal, 2022): a double-loop stochastic AID-based bilevel algorithm that uses conjugate gradient to estimate the Hessian inverse.

- stocBiO (Ji et al., 2021): a two timescale stochastic AID-based bilevel approach that uses Neumann Series to estimate the Hessian inverse. The repository of stocBiO is available at https://github.com/JunjieYang97/StocBio.

- BSA (Ghadimi & Wang, 2018): an AID-based bilevel method that uses single-sample sampling.

- MRBO (Yang et al., 2021): a single-loop AID-based stochastic bilevel algorithm that uses momentum-based SGD to accelerate convergence. The implementation of MRBO is available at https://github.com/JunjieYang97/MRVRBO.

- F$^2$SA (Kwon et al., 2023): a fully first-order (Hessian/Jacobian-free) stochastic bilevel method, which is doulbe-loop.

- F$^3$SA (Kwon et al., 2023): a fully first-order stochastic bilevel approach that uses momentum-based SGD to accelerate convergence and is single timescale.

### B.2    EXPERIMENTAL DETAILS FOR DATA HYPER-CLEANING

In this section, we describe the details of the experiments on data hyper-cleaning. The goal of data hyper-cleaning is to train a classifier on a potentially corrupt dataset. To make fair comparison, we follow the same implementation as in (Ji et al., 2021; Yang et al., 2021) and apply it to other algorithms. The objective function can be written as follows:

$$\min_{\lambda} \mathcal{L}_{\mathcal{D}_{val}}\left(\lambda, w^*\right) = \frac{1}{|\mathcal{D}_{val}|} \sum_{(\mathbf{x}_i, \mathbf{y}_i) \in \mathcal{D}_{val}} \mathcal{L}\left(w^* \mathbf{x}_i, \mathbf{y}_i\right)$$

$$\text{s.t.} \quad w^* = \arg\min_{w}\left(\frac{1}{|\mathcal{D}_{tr}|} \sum_{(\mathbf{x}_i, \mathbf{y}_i) \in \mathcal{D}_{tr}} \sigma\left(\lambda_i\right) \mathcal{L}\left(w \mathbf{x}_i, \mathbf{y}_i\right) + C_r \|w\|^2\right),$$

where $(\mathbf{x}_i, \mathbf{y}_i)$ represents the data samples, $\mathcal{D}_{val}$ and $\mathcal{D}_{tr}$ correspond to the validation data and the training data, $\mathcal{L}$ denotes the cross-entropy loss, $\sigma$ represents the sigmoid function, and $C_r$ is the regularization parameter. Note that the training loss corresponds to the upper-level loss. We choose $C_r = 0.001$ in our experiments, which is the same as (Shaban et al., 2019; Ji et al., 2021). We conduct experiments on the MNIST dataset (LeCun et al., 1998), which is corrupted by replacing the training data label with a uniformly random one. Such replacement has a probability $p$, referred to as the corruption rate. We run the experiments with corruption rates of $p = \{0.1,\ 0.15,\ 0.2,\ 0.25,\ 0.3\}$.

We compare the performance of LazyBLO with AmIGO (Arbel & Mairal, 2022), stocBiO (Ji et al., 2021), BSA (Ghadimi & Wang, 2018), and MRBO (Yang et al., 2021). For all algorithms, we tune the parameters using grid search to achieve the best convergence performance based on the training loss as the metric. As a result, we set the batch size to 1000 for AmIGO, stocBiO, MRBO and LazyBLO . We set both the outer stepsize $\alpha$ and the inner stepsize $\beta$ as 0.1, and the Hessian update stepsize $\gamma$ as 0.5 for AmIGO, stocBiO and MRBO. We choose both the outer stepsize $\alpha$ and the inner stepsize $\beta$ to be 0.01, and the Hessian update stepsize $\gamma$ to be 0.1 for BSA. For LazyBLO, We set 0.5 as the inner stepsize $\beta$, and 0.1 as both the outer stepsize $\alpha$ and the Hessian update stepsize $\gamma$. We set the number of inner-loop iterations for y-update to 64 for AmIGO, stocBiO and BSA. We choose the number of iterations for Hessian inverse evaluations to be 6 for AmIGO and stocBiO, and 12 for MRBO and BSA. For LazyBLO, we set 8 as the inner-loop iteration number $N$ for x- and y-update. We conduct 10 repetitions for the experiments using different random seeds. The solid line shows the average training loss, and the shaded area represents the variance containing the maximum and the minimum values. We run the data hyper-cleaning experiments using NVIDIA GeForce RTX 3060 GPU.

## B.3 EXPERIMENTAL DETAILS FOR DEEP HYPER-REPRESENTATION

In this section, we show the details of the experiments on deep hyper-representation, which aims to classify the images. The objective function is given by:

$$\min_{\lambda} \mathcal{L}_{\mathcal{D}_{val}}(\lambda, w^*) = \frac{1}{|\mathcal{D}_{val}|} \sum_{(\mathbf{x}_i, \mathbf{y}_i) \in \mathcal{D}_{val}} \mathcal{L}\left(w^* f\left(\lambda; \mathbf{x}_i\right), \mathbf{y}_i\right)$$

$$\text{s.t.} \quad w^* = \arg\min_{w} \frac{1}{|\mathcal{D}_{tr}|} \sum_{(\mathbf{x}_i, \mathbf{y}_i) \in \mathcal{D}_{tr}} \mathcal{L}\left(w f\left(\lambda, \mathbf{x}_i\right), \mathbf{y}_i\right),$$

where $(\mathbf{x}_i, \mathbf{y}_i)$ denotes the data samples, $\mathcal{D}_{val}$ and $\mathcal{D}_{tr}$ are the validation data and the training data, $\mathcal{L}$ corresponds to the cross-entropy loss, $f\left(\lambda; \mathbf{x}_i\right)$ represents the features extracted from the data sample. We run the experiments with ResNet-20 network (He et al., 2016) on CIFAT-10 dataset (Krizhevsky et al., 2009) using a batch size of 128. We treat the last two layers in ResNet-20 as the LL parameters $w$ with a dimension of $5, 130$, and all remaining layers as the UL parameters $\lambda$ with a dimension of $11, 168, 832$.

We compare LazyBLO with AmIGO (Arbel & Mairal, 2022), F$^2$SA (Kwon et al., 2023) and F$^3$SA (Kwon et al., 2023). To ensure the best performance of all the algorithms, we fine tune the parameters using grid search with the goal of finding the lowest training loss. Consequently, for AmIGO, we set all the stepsize for updating $\mathbf{x}$, $\mathbf{y}$ and $\mathbf{z}$ to 0.01. We choose the number of y-update iterations to be 8 and the number of z-update iterations to be 2. For LazyBLO, we choose the stepsize $\alpha$ and $\gamma$ to be 0.01, and $\beta$ to be 0.05. We set 2 as the inner-loop iteration number $N$. Following the same notations as in (Kwon et al., 2023), for F$^2$SA, we choose the stepsize $\alpha$ as 0.1 and $\gamma$ as 0.01. We set both the step-size ratio $\xi$ and the Lagrangian multiplier $\lambda$ to 0.5. We choose the number of inner-loop iterations to be 1. For F$^3$SA, we set 0.05 as $\alpha$, 0.01 as $\gamma$, 0.1 as $\xi$, 0.5 as $\lambda$, and 0.9 as momentum-weight $\eta$. We repeat the experiments 10 times with different random seeds, where the solid line represents the average training loss or test accuracy, and the shaded area shows the variance containing the maximum and the minimum values. We run the deep hyper-representation experiments using NVIDIA Tesla V100 GPU.

## B.4 ADDITIONAL EXPERIMENT RESULTS

### B.4.1 DATA HYPER-CLEANING

We can see in Table 4 that the test accuracy of LazyBLO is *comparable* to the SOTA baseline algorithms although LazyBLO uses stale Hessian information. In addition, the number of Hessian

Table 4: Convergence performance of different bilevel algorithms on data hyper-cleaning on MNIST dataset (corruption rate $p = 0.1$, average over 10 repetitions).

|  | LAZYBLO | AMIGO | STOCBIO | MRBO | BSA |
|---|---|---|---|---|---|
| TEST ACCURACY (%) | 72.31 | 72.12 | 72.75 | 69.46 | 72.92 |
| # OF HESSIAN | 6 | 60 | 60 | 1440 | 720 |

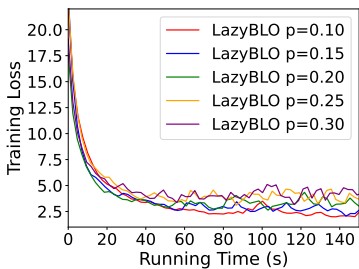

Figure 4: Comparison of LazyBLO on data hyper-cleaning on MNIST dataset with different corruption rates ($p$).

computations required for LazyBLO to converge is significantly reduced, which is ten times fewer than AmIGO and stocBiO, 240 times fewer than MRBO, and 120 times fewer than BSA.

Figure 4 illustrates the robustness of LazyBLO against corrupted datasets. We can see from Figure 4 that when the corruption rate $p$ (the probability that a training data label is replaced by a uniformly random one) is larger, the training loss becomes higher, which is natural since with larger corruption rate the classification problem becomes challenging. However, the convergence speed of LazyBLO is similar regardless of the corruption rate $p$.

### B.4.2 DEEP HYPER-REPRESENTATION

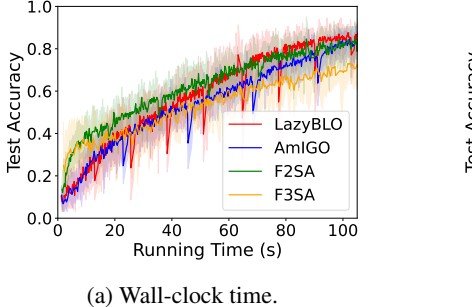

(a) Wall-clock time.

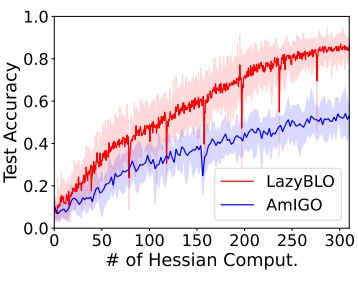

(b) # of Hessian comput.

Figure 5: Test accuracy of different bilevel algorithms on deep hyper-representation on CIFAR-10 dataset (10 repetitions).

Figure 5a illustrates the test accuracy of LazyBLO compared with the baseline algorithms, and it demonstrates that LazyBLO converges faster in terms of wall-clock time compared to both F$^2$SA and F$^3$SA. Figure 5b shows the test accuracy of LazyBLO compared to AmIGO in terms of the number of Hessian computations, and it indicates that with the same number of Hessian evaluations, LazyBLO has a higher test accuracy compared to AmIGO.

## C   PROOF OF THEOREM 5.3: NON-CONVEX $\ell(\mathbf{x})$

### C.1   PROOF SKETCHES

Here, we provide a detailed proof sketch of Theorem 5.3. The detailed proof is provided in Appendix C.2. The proof is organized into five key steps:

*Step 1) Descent in the upper-level objective function:* First, we show the bound for the per-iterate descent of the UL problem as follows:

**Lemma C.1.** *Under Assumptions 3.1–3.4, the following inequality holds for successive iterations of Algorithm 1:*

$$
\mathbb{E}\left[\ell\left(\mathbf{x}_t^{n+1}\right) - \ell\left(\mathbf{x}_t^n\right)\right] \leq -\frac{\alpha_t}{2}\mathbb{E}\left[\|\nabla\ell\left(\mathbf{x}_t^n\right)\|^2\right] - \left(\frac{\alpha_t}{2} - \frac{\alpha_t^2 L_l}{2}\right)\mathbb{E}\left[\left\|h_{t,n}^f\right\|^2\right] + 4\tilde{\sigma}_{g_{xy}}^2\frac{B_{f_y}^2}{\mu_g^2}\alpha_t
$$

$$
+ 8L_f^2 L_z^2\alpha_t^3 N\sum_{i=0}^{N-1}\mathbb{E}\left[\left\|h_{t,i}^f\right\|^2\right] + 2\alpha_t L_f^2\mathbb{E}\left[\|\mathbf{y}_t^n - \mathbf{y}^*\left(\mathbf{x}_t^n\right)\|^2\right] + 2\sigma_{f_x}^2\alpha_t + 16L_f^2 L_z^2\beta_t^2 N^2\sigma_{g_y}^2\alpha_t
$$

$$
+ \left(4\sigma_{g_{xy}}^2\alpha_t + 4L_f^2\alpha_t\right)\mathbb{E}\left[\|\mathbf{z}_t - \mathbf{z}_t^*\|^2\right] + 16L_f^2 L_z^2\beta_t^2 N\alpha_t\sum_{i=0}^{N-1}\mathbb{E}\left[\|\nabla_{\mathbf{y}}g\left(\mathbf{x}_t^i, \mathbf{y}_t^i\right)\|^2\right],
$$

*for all $t \in \{0, 1, \ldots, T-1\}$ and $n \in \{0, 1, \ldots, N-1\}$, where the expectation is taken over the stochasticity of the algorithm.*

Lemma C.1 indicates that the descent in the upper-level objective function depends on i) the stochastic gradient estimator $\mathbb{E}[\|h_{t,n}^f\|^2]$, ii) the full gradient $\mathbb{E}[\|\nabla_{\mathbf{y}}g\left(\mathbf{x}_t^n, \mathbf{y}_t^n\right)\|^2]$, iii) the approximation error of $\mathbf{y}^*\left(\mathbf{x}\right)$, which is $\mathbb{E}[\|\mathbf{y}_t^n - \mathbf{y}^*\left(\mathbf{x}_t^n\right)\|^2]$ and will be bounded in Step 2), and iv) the approximation gap of $\mathbf{z}^*\left(\mathbf{x}, \mathbf{y}\right)$, which is $\mathbb{E}[\|\mathbf{z}_t - \mathbf{z}_t^*\|^2]$ and will be bounded in Step 3).

*Step 2) Descent in the error of $\mathbf{y}^*\left(\mathbf{x}\right)$:* We bound the approximation error of $\mathbf{y}^*\left(\mathbf{x}\right)$ as follows:

**Lemma C.2.** *Under Assumptions 3.2–3.4, the approximation error of $\mathbf{y}^*\left(\mathbf{x}\right)$ for Algorithm 1 satisfies the following inequality:*

$$
\mathbb{E}\left[\left\|\mathbf{y}_t^{n+1} - \mathbf{y}^*\left(\mathbf{x}_t^{n+1}\right)\right\|^2\right] \leq (1+c_1)(1+c_2)\left(1 - \frac{2\beta_t\mu_g L_g}{\mu_g + L_g}\right)\mathbb{E}\left[\|\mathbf{y}_t^n - \mathbf{y}^*\left(\mathbf{x}_t^n\right)\|^2\right]
$$

$$
+ \left(1 + \frac{1}{c_1}\right)L_y^2\alpha_t^2\mathbb{E}\left[\left\|h_{t,n}^f\right\|^2\right] + (1+c_1)(1+c_2)\left(\beta_t^2 - \frac{2\beta_t}{\mu_g + L_g}\right)\mathbb{E}\left[\|\nabla_{\mathbf{y}}g\left(\mathbf{x}_t^n, \mathbf{y}_t^n\right)\|^2\right]
$$

$$
+ (1+c_1)\left(1 + \frac{1}{c_2}\right)\beta_t^2\sigma_{g_y}^2,
$$

*for all $t \in \{0, 1, \ldots, T-1\}$ and $n \in \{0, 1, \ldots, N-1\}$ with some constants $c_1, c_2 > 0$, where the expectation is taken over the randomness of the algorithm.*

Lemma C.2 shows that the approximation error of $\mathbf{y}^*\left(\mathbf{x}\right)$ is affected by the full gradient $\mathbb{E}[\|\nabla_{\mathbf{y}}g\left(\mathbf{x}_t^n, \mathbf{y}_t^n\right)\|^2]$, and the stochastic gradient estimator $\mathbb{E}[\|h_{t,n}^f\|^2]$, which is due to the coupled structure of the bilevel optimization problem.

*Step 3) Descent in the error of $\mathbf{z}^*\left(\mathbf{x}, \mathbf{y}\right)$:* Next, we demonstrate that the approximation error of $\mathbf{z}^*\left(\mathbf{x}, \mathbf{y}\right)$ can be bounded as follows:

**Lemma C.3.** *Under Assumptions 3.1–3.4, the following inequality of the approximation error of $\mathbf{z}^*\left(\mathbf{x}, \mathbf{y}\right)$ holds for Algorithm 1:*

$$
\mathbb{E}\left[\left\|\mathbf{z}_{t+1} - \mathbf{z}_{t+1}^*\right\|^2\right] \leq (1+c_3)(1+c_4)\left(\gamma_t^2 - \frac{2\gamma_t}{\mu_g + L_q}\right)\mathbb{E}\left[\left\|\nabla_{\mathbf{z}}q\left(\mathbf{x}_t^N, \mathbf{y}_t^N, \mathbf{z}_t\right)\right\|^2\right]
$$

$$
+ \left((1+c_3)(1+c_4)\left(1 - \frac{2\gamma_t\mu_g L_q}{\mu_g + L_q}\right) + 4\sigma_{g_{yy}}^2\gamma_t^2(1+c_3)\left(1 + \frac{1}{c_4}\right)\right)\mathbb{E}\left[\|\mathbf{z}_t - \mathbf{z}_t^*\|^2\right]
$$

$$
+ 2\left(1 + \frac{1}{c_3}\right)L_z^2\alpha_t^2 N\sum_{n=0}^{N-1}\mathbb{E}\left[\left\|h_{t,n}^f\right\|^2\right] + 4\left(1 + \frac{1}{c_3}\right)L_z^2\beta_t^2 N\sum_{n=0}^{N-1}\mathbb{E}\left[\|\nabla_{\mathbf{y}}g\left(\mathbf{x}_t^n, \mathbf{y}_t^n\right)\|^2\right]
$$

$$
+ 2\sigma_{f_y}^2(1+c_3)\left(1 + \frac{1}{c_4}\right)\gamma_t^2 + 4\sigma_{g_{yy}}^2\frac{B_{f_y}^2}{\mu_g^2}(1+c_3)\left(1 + \frac{1}{c_4}\right)\gamma_t^2 + 4\left(1 + \frac{1}{c_3}\right)L_z^2\beta_t^2 N^2\sigma_{g_y}^2,
$$

*for all $t \in \{0, 1, \ldots, T-1\}$ and $n \in \{0, 1, \ldots, N-1\}$ with some constants $c_3, c_4 > 0$, where $\mathbf{z}_t = \mathbf{z}\left(\mathbf{x}_t^0, \mathbf{y}_t^0\right)$ and $\mathbf{z}_t^* = \mathbf{z}^*\left(\mathbf{x}_t^0, \mathbf{y}_t^0\right)$. The expectation is taken over the stochasticity of the algorithm.*

Lemma C.3 shows that the approximation error of $\mathbf{z}^*(\mathbf{x}, \mathbf{y})$ is influenced by the full gradients $\mathbb{E}[\|\nabla_{\mathbf{y}} g(\mathbf{x}_t^n, \mathbf{y}_t^n)\|^2]$ and $\mathbb{E}[\|\nabla_{\mathbf{z}} q(\mathbf{x}_t^N, \mathbf{y}_t^N, \mathbf{z}_t)\|^2]$, and the stochastic gradient estimator $\mathbb{E}[\|h_{t,n}^f\|^2]$, which is due to the coupled structure of the quadratic problem in (3).

*Step 4) Descent in the potential function:* We define the potential function $W_t$ as follows:

$$
W_t = \ell\left(\mathbf{x}_t^0\right) + K_y \left\|\mathbf{y}_t^0 - \mathbf{y}^*\left(\mathbf{x}_t^0\right)\right\|^2 + K_z \left\|\mathbf{z}_t\left(\mathbf{x}_t^0, \mathbf{y}_t^0\right) - \mathbf{z}^*\left(\mathbf{x}_t^0, \mathbf{y}_t^0\right)\right\|^2 .
$$

To demonstrate the descent in the potential function, we prove the following lemma.

**Lemma C.4.** *Set $c_1 = \frac{\beta_t L_{\mu_g}}{2(1-\beta_t L_{\mu_g})}$, $c_2 = \frac{\beta_t L_{\mu_g}}{1-2\beta_t L_{\mu_g}}$, $c_3 = \frac{\gamma_t L_{\mu_q}}{2(1-\gamma_t L_{\mu_q})}$, and $c_4 = \frac{\gamma_t L_{\mu_q}}{1-2\gamma_t L_{\mu_q}}$. Under the same conditions as described in Theorem 5.3 and using Lemmas C.1-C.3, the iterates generated by Algorithm 1 satisfies: for all $t \in \{0, 1, \ldots, T-1\}$,*

$$
\mathbb{E}\left[W_{t+1} - W_t\right] \leq -\frac{\alpha_t}{2} \sum_{n=0}^{N-1} \mathbb{E}\left[\|\nabla\ell\left(\mathbf{x}_t^n\right)\|^2\right] + \sigma_{g_{xy}}^2 \alpha_t C_{g_{xy}} N + \sigma_{f_x}^2 \alpha_t C_{f_x} N + \sigma_{g_{yy}}^2 \alpha_t C_{g_{yy}} N
$$

$$
+ \sigma_{f_y}^2 \alpha_t C_{f_y} N + \sigma_{g_y}^2 \alpha_t \left(C_{g_1} N + C_{g_2} \frac{1}{N}\right)
$$

*where the constant values $C_{g_{xy}}$, $C_{f_x}$, $C_{g_1}$, $C_{g_2}$, $C_{g_{yy}}$ and $C_{f_y}$, which are independent of N, are defined in (18) of Appendix C.*

With the proper parameter choices, the coefficients of $\mathbb{E}[\|\mathbf{y}_t^n - \mathbf{y}^*(\mathbf{x}_t^n)\|^2]$, $\mathbb{E}[\|\mathbf{z}_t - \mathbf{z}_t^*\|^2]$, $\mathbb{E}[\|\nabla_{\mathbf{y}} g(\mathbf{x}_t^n, \mathbf{y}_t^n)\|^2]$, $\mathbb{E}[\|h_{t,n}^f\|^2]$ and $\mathbb{E}[\|\nabla_{\mathbf{z}} q(\mathbf{x}_t^N, \mathbf{y}_t^N, \mathbf{z}_t)\|^2]$ are made to be non-positive within the ranges of $\alpha_t$, $\beta_t$ and $\gamma_t$.

*Step 5) Proof of Theorem 5.3:* Choose a constant step-size $\alpha_t = \alpha$. Under the same conditions as described in Theorem 5.3, telescoping the result in Lemma C.4 from 0 to $T-1$ yields:

$$
\frac{1}{TN} \sum_{t=0}^{T-1} \sum_{n=0}^{N-1} \mathbb{E}\left[\|\nabla\ell\left(\mathbf{x}_t^n\right)\|^2\right] \leq \frac{2\left(W_0 - \ell^*\right)}{\alpha NT} + 2\left(\sigma_{f_x}^2 C_{f_x} + \sigma_{f_y}^2 C_{f_y} + \sigma_{g_y}^2 \left(C_{g_1} + C_{g_2} \frac{1}{N^2}\right)\right.
$$

$$
\left. + \sigma_{g_{yy}}^2 C_{g_{yy}} + \sigma_{g_{xy}}^2 C_{g_{xy}}\right),
$$

where $W_0 = \ell\left(\mathbf{x}_0^0\right) + K_y \left\|\mathbf{y}_0^0 - \mathbf{y}^*\left(x_0^0\right)\right\|^2 + K_z \left\|\mathbf{z}_0 - \mathbf{z}^*\left(\mathbf{x}_0^0, \mathbf{y}_0^0\right)\right\|^2$. The proof of Theorem 5.3 is completed.

## C.2 DETAILED PROOF

### C.2.1 DESCENT IN THE UPPER-LEVEL OBJECTIVE FUNCTION

**Lemma C.5.** *Under Assumptions 3.1–3.4, the following inequality holds for successive iterations of Algorithm 1:*

$$
\mathbb{E}\left[\ell\left(\mathbf{x}_t^{n+1}\right) - \ell\left(\mathbf{x}_t^n\right)\right]
$$

$$
\leq -\frac{\alpha_t}{2} \mathbb{E}\left[\|\nabla\ell\left(\mathbf{x}_t^n\right)\|^2\right] - \left(\frac{\alpha_t}{2} - \frac{\alpha_t^2 L_l}{2}\right) \mathbb{E}\left[\left\|h_{t,n}^f\right\|^2\right] + 8L_f^2 L_z^2 \alpha_t^3 N \sum_{i=0}^{N-1} \mathbb{E}\left[\left\|h_{t,i}^f\right\|^2\right]
$$

$$
+ 2\alpha_t L_f^2 \mathbb{E}\left[\|\mathbf{y}_t^n - \mathbf{y}^*\left(\mathbf{x}_t^n\right)\|^2\right] + \left(4\sigma_{g_{xy}}^2 \alpha_t + 4L_f^2 \alpha_t\right) \mathbb{E}\left[\|\mathbf{z}_t - \mathbf{z}_t^*\|^2\right] + 2\sigma_{f_x}^2 \alpha_t
$$

$$
+ 16L_f^2 L_z^2 \beta_t^2 N \alpha_t \sum_{i=0}^{N-1} \mathbb{E}\left[\|\nabla_{\mathbf{y}} g(\mathbf{x}_t^i, \mathbf{y}_t^i)\|^2\right] + 16L_f^2 L_z^2 \beta_t^2 N^2 \sigma_{g_y}^2 \alpha_t + 4\sigma_{g_{xy}}^2 \frac{B_{f_y}^2}{\mu_g^2} \alpha_t,
$$

*for all $t \in \{0, 1, \ldots, T-1\}$ and $n \in \{0, 1, \ldots, N-1\}$, where the expectation is taken over the stochasticity of the algorithm.*

*Proof.* We have

$$\mathbb{E}\left[\ell\left(\mathbf{x}_t^{n+1}\right) - \ell\left(\mathbf{x}_t^n\right)\right]$$

$$\stackrel{(a)}{\leq} \mathbb{E}\left[\left\langle\nabla\ell\left(\mathbf{x}_t^n\right), \mathbf{x}_t^{n+1} - \mathbf{x}_t^n\right\rangle + \frac{L_l}{2}\left\|\mathbf{x}_t^{n+1} - \mathbf{x}_t^n\right\|^2\right]$$

$$\stackrel{(b)}{=} \mathbb{E}\left[-\alpha_t\left\langle\nabla\ell\left(\mathbf{x}_t^n\right), h_{t,n}^f\right\rangle + \frac{\alpha_t^2 L_l}{2}\left\|h_{t,n}^f\right\|^2\right]$$

$$\stackrel{(c)}{=} \mathbb{E}\left[-\frac{\alpha_t}{2}\left\|\nabla\ell\left(\mathbf{x}_t^n\right)\right\|^2 - \frac{\alpha_t}{2}\left\|h_{t,n}^f\right\|^2 + \frac{\alpha_t}{2}\left\|\nabla\ell\left(\mathbf{x}_t^n\right) - h_{t,n}^f\right\|^2 + \frac{\alpha_t^2 L_l}{2}\left\|h_{t,n}^f\right\|^2\right], \quad (8)$$

where (a) uses the Lipschitz continuous gradients of $\ell$ (see Lemma 5.1). (b) follows from the update rule of Algorithm 1. (c) is because of $\langle x, y\rangle = \frac{1}{2}\|x\|^2 + \frac{1}{2}\|y\|^2 - \frac{1}{2}\|x - y\|^2$.

Next, we bound the third term on the right in (8) above. Before that, we bound $\left\|\mathbf{x}_t^n - \mathbf{x}_t^0\right\|^2$ and $\left\|\mathbf{y}_t^n - \mathbf{y}_t^0\right\|^2$.

$$\left\|\mathbf{x}_t^n - \mathbf{x}_t^0\right\|^2 \stackrel{(a)}{=} \alpha_t^2\left\|\sum_{i=0}^{n-1} h_{t,i}^f\right\|^2 \stackrel{(b)}{\leq} \alpha_t^2 n\sum_{i=0}^{n-1}\left\|h_{t,i}^f\right\|^2 \leq \alpha_t^2 N\sum_{i=0}^{N-1}\left\|h_{t,i}^f\right\|^2, \quad (9)$$

where (a) is because of the update rule of Algorithm 1. (b) is due to $\|z_1 + \cdots + z_k\|^2 \leq k\|z_1\|^2 + \cdots + k\|z_k\|^2$.

Similarly,

$$\left\|\mathbf{y}_t^n - \mathbf{y}_t^0\right\|^2 \leq \beta_t^2 N\sum_{i=0}^{N-1}\left\|h_{t,i}^g\right\|^2. \quad (10)$$

Considering $\mathbb{E}\left[\left\|\nabla\ell\left(\mathbf{x}_t^n\right) - h_{t,n}^f\right\|^2\right]$, we have

$$\mathbb{E}\left[\left\|\nabla\ell\left(\mathbf{x}_t^n\right) - h_{t,n}^f\right\|^2\right] = \mathbb{E}\left[\left\|\nabla\ell\left(\mathbf{x}_t^n\right) - \nabla f\left(\mathbf{x}_t^n, \mathbf{y}_t^n, \mathbf{z}_t\right) + \nabla f\left(\mathbf{x}_t^n, \mathbf{y}_t^n, \mathbf{z}_t\right) - h_{t,n}^f\right\|^2\right]$$

$$\stackrel{(a)}{\leq} \mathbb{E}\left[2\left\|h_{t,n}^f - \nabla f\left(\mathbf{x}_t^n, \mathbf{y}_t^n, \mathbf{z}_t\right)\right\|^2 + 2\left\|\nabla f\left(\mathbf{x}_t^n, \mathbf{y}_t^n, \mathbf{z}_t\right) - \nabla\ell\left(\mathbf{x}_t^n\right)\right\|^2\right]$$

$$\stackrel{(b)}{\leq} \mathbb{E}\left[2\left\|h_{t,n}^f - \nabla f\left(\mathbf{x}_t^n, \mathbf{y}_t^n, \mathbf{z}_t\right)\right\|^2 + 2L_f^2\left(\left\|\mathbf{y}_t^n - \mathbf{y}^*\left(\mathbf{x}_t^n\right)\right\| + \left\|\mathbf{z}_t - \mathbf{z}^*\left(\mathbf{x}_t^n, \mathbf{y}_t^n\right)\right\|\right)^2\right]$$

$$\leq \mathbb{E}\left[2\left\|h_{t,n}^f - \nabla f\left(\mathbf{x}_t^n, \mathbf{y}_t^n, \mathbf{z}_t\right)\right\|^2 + 4L_f^2\left\|\mathbf{y}_t^n - \mathbf{y}^*\left(\mathbf{x}_t^n\right)\right\|^2 + 4L_f^2\left\|\mathbf{z}_t - \mathbf{z}^*\left(\mathbf{x}_t^n, \mathbf{y}_t^n\right)\right\|^2\right]$$

$$\stackrel{(c)}{\leq} \mathbb{E}\left[2\left\|h_{t,n}^f - \nabla f\left(\mathbf{x}_t^n, \mathbf{y}_t^n, \mathbf{z}_t\right)\right\|^2 + 4L_f^2\left\|\mathbf{y}_t^n - \mathbf{y}^*\left(\mathbf{x}_t^n\right)\right\|^2 + 8L_f^2\left\|\mathbf{z}_t - \mathbf{z}_t^*\right\|^2\right.$$

$$+8L_f^2\left\|\mathbf{z}^*\left(\mathbf{x}_t^0, \mathbf{y}_t^0\right) - \mathbf{z}^*\left(\mathbf{x}_t^n, \mathbf{y}_t^n\right)\right\|^2\right]$$

$$\stackrel{(d)}{\leq} \mathbb{E}\left[2\left\|h_{t,n}^f - \nabla f\left(\mathbf{x}_t^n, \mathbf{y}_t^n, \mathbf{z}_t\right)\right\|^2 + 4L_f^2\left\|\mathbf{y}_t^n - \mathbf{y}^*\left(\mathbf{x}_t^n\right)\right\|^2 + 8L_f^2\left\|\mathbf{z}_t - \mathbf{z}_t^*\right\|^2\right.$$

$$+8L_f^2 L_z^2\left(\left\|\mathbf{x}_t^0 - \mathbf{x}_t^n\right\| + \left\|\mathbf{y}_t^0 - \mathbf{y}_t^n\right\|\right)^2\right]$$

$$\leq \mathbb{E}\left[2\left\|h_{t,n}^f - \nabla f\left(\mathbf{x}_t^n, \mathbf{y}_t^n, \mathbf{z}_t\right)\right\|^2 + 4L_f^2\left\|\mathbf{y}_t^n - \mathbf{y}^*\left(\mathbf{x}_t^n\right)\right\|^2 + 8L_f^2\left\|\mathbf{z}_t - \mathbf{z}_t^*\right\|^2 + 16L_f^2 L_z^2\left\|\mathbf{x}_t^n - \mathbf{x}_t^0\right\|^2\right.$$

$$+16L_f^2 L_z^2\left\|\mathbf{y}_t^n - \mathbf{y}_t^0\right\|^2\right]$$

$$\stackrel{(e)}{\leq} \mathbb{E}\left[2\left\|h_{t,n}^f - \nabla f\left(\mathbf{x}_t^n, \mathbf{y}_t^n, \mathbf{z}_t\right)\right\|^2 + 4L_f^2\left\|\mathbf{y}_t^n - \mathbf{y}^*\left(\mathbf{x}_t^n\right)\right\|^2 + 8L_f^2\left\|\mathbf{z}_t - \mathbf{z}_t^*\right\|^2\right.$$

$$+ 16 L_f^2 L_z^2 \alpha_t^2 N \sum_{i=0}^{N-1} \left\| h_{t,i}^f \right\|^2 + 16 L_f^2 L_z^2 \beta_t^2 N \sum_{i=0}^{N-1} \left\| h_{t,i}^g \right\|^2 \Bigg]$$

$$\overset{(f)}{\leq} \mathbb{E}\left[ 2 \left\| h_{t,n}^f - \nabla f\left(\mathbf{x}_t^n, \mathbf{y}_t^n, \mathbf{z}_t\right) \right\|^2 + 4 L_f^2 \left\| \mathbf{y}_t^n - \mathbf{y}^*\left(\mathbf{x}_t^n\right) \right\|^2 + 8 L_f^2 \left\| \mathbf{z}_t - \mathbf{z}_t^* \right\|^2 \right.$$

$$+ 16 L_f^2 L_z^2 \alpha_t^2 N \sum_{i=0}^{N-1} \left\| h_{t,i}^f \right\|^2 + 32 L_f^2 L_z^2 \beta_t^2 N \sum_{i=0}^{N-1} \left\| h_{t,i}^g - \nabla_{\mathbf{y}} g\left(\mathbf{x}_t^i, \mathbf{y}_t^i\right) \right\|^2$$

$$+ 32 L_f^2 L_z^2 \beta_t^2 N \sum_{i=0}^{N-1} \left\| \nabla_{\mathbf{y}} g\left(\mathbf{x}_t^i, \mathbf{y}_t^i\right) \right\|^2 \Bigg]$$

$$\overset{(g)}{\leq} \mathbb{E}\left[ 2 \left\| h_{t,n}^f - \nabla f\left(\mathbf{x}_t^n, \mathbf{y}_t^n, \mathbf{z}_t\right) \right\|^2 + 4 L_f^2 \left\| \mathbf{y}_t^n - \mathbf{y}^*\left(\mathbf{x}_t^n\right) \right\|^2 + 8 L_f^2 \left\| \mathbf{z}_t - \mathbf{z}_t^* \right\|^2 \right.$$

$$+ 16 L_f^2 L_z^2 \alpha_t^2 N \sum_{i=0}^{N-1} \left\| h_{t,i}^f \right\|^2 + 32 L_f^2 L_z^2 \beta_t^2 N \sum_{i=0}^{N-1} \left\| \nabla_{\mathbf{y}} g\left(\mathbf{x}_t^i, \mathbf{y}_t^i\right) \right\|^2 \Bigg] + 32 L_f^2 L_z^2 \beta_t^2 N^2 \sigma_{g_y}^2, \quad (11)$$

where $\mathbf{z}_t = \mathbf{z}\left(\mathbf{x}_t^0, \mathbf{y}_t^0\right)$ and $\mathbf{z}_t^* = \mathbf{z}^*\left(\mathbf{x}_t^0, \mathbf{y}_t^0\right)$. (a), (c) and (f) follow from $\|x + y\|^2 \leq 2\|x\|^2 + 2\|y\|^2$ and $\left\| \nabla_{\mathbf{xy}}^2 g\left(\mathbf{x}, \mathbf{y}\right) \right\| \leq B_{g_{xy}}$. (b) utilizes the Lipschitzness of $\nabla f\left(\mathbf{x}, \mathbf{y}, \mathbf{z}\right)$ (see Lemma 5.1), and (d) is due to the Lipschitzness of $\mathbf{z}^*\left(\mathbf{x}, \mathbf{y}\right)$ (see Lemma 5.2). (e) uses equations (9) and (10). (g) is because of the bounded variance in Assumption 3.4.

Then, we bound the term $\mathbb{E}\left[ \left\| h_{t,n}^f - \nabla f\left(\mathbf{x}_t^n, \mathbf{y}_t^n, \mathbf{z}_t\right) \right\|^2 \right]$.

$$\mathbb{E}\left[ \left\| h_{t,n}^f - \nabla f\left(\mathbf{x}_t^n, \mathbf{y}_t^n, \mathbf{z}_t\right) \right\|^2 \right]$$

$$\overset{(a)}{=} \mathbb{E}\left[ \left\| \nabla_{\mathbf{x}} f\left(\mathbf{x}_t^n, \mathbf{y}_t^n, \mathcal{D}_{t,n}^{f_x}\right) + \nabla_{\mathbf{xy}}^2 g\left(\mathbf{x}_t^n, \mathbf{y}_t^n, \mathcal{D}_{t,n}^{g_{xy}}\right) \mathbf{z}_t - \nabla_{\mathbf{x}} f\left(\mathbf{x}_t^n, \mathbf{y}_t^n\right) - \nabla_{\mathbf{xy}}^2 g\left(\mathbf{x}_t^n, \mathbf{y}_t^n\right) \mathbf{z}_t \right\|^2 \right]$$

$$\leq 2\mathbb{E}\left[ \left\| \nabla_{\mathbf{x}} f\left(\mathbf{x}_t^n, \mathbf{y}_t^n, \mathcal{D}_{t,n}^{f_x}\right) - \nabla_{\mathbf{x}} f\left(\mathbf{x}_t^n, \mathbf{y}_t^n\right) \right\|^2 + \|\mathbf{z}_t\|^2 \left\| \nabla_{\mathbf{xy}}^2 g\left(\mathbf{x}_t^n, \mathbf{y}_t^n, \mathcal{D}_{t,n}^{g_{xy}}\right) - \nabla_{\mathbf{xy}}^2 g\left(\mathbf{x}_t^n, \mathbf{y}_t^n\right) \right\|^2 \right]$$

$$\overset{(b)}{\leq} \mathbb{E}\left[ 2\sigma_{g_{xy}}^2 \|\mathbf{z}_t\|^2 + 2\sigma_{f_x}^2 \right]$$

$$\overset{(c)}{\leq} \mathbb{E}\left[ 4\sigma_{g_{xy}}^2 \|\mathbf{z}_t - \mathbf{z}_t^*\|^2 + 4\sigma_{g_{xy}}^2 \|\mathbf{z}_t^*\|^2 + 2\sigma_{f_x}^2 \right]$$

$$\overset{(d)}{\leq} \mathbb{E}\left[ 4\sigma_{g_{xy}}^2 \|\mathbf{z}_t - \mathbf{z}_t^*\|^2 + 4\sigma_{g_{xy}}^2 \frac{B_{f_y}^2}{\mu_g^2} + 2\sigma_{f_x}^2 \right], \quad (12)$$

where (a) uses the definitions of $h_{t,n}^f$ and $\nabla f\left(\mathbf{x}_t^n, \mathbf{y}_t^n, \mathbf{z}_t\right)$. (b) utilizes the bounded variance in Assumption 3.4. (c) uses $\|x + y\|^2 \leq 2\|x\|^2 + 2\|y\|^2$, and (d) is due to the bound of $\mathbf{z}^*\left(\mathbf{x}, \mathbf{y}\right)$ in Lemma 5.2.

Combining (8), (11) and (12) completes the proof of the lemma. $\qquad\square$

### C.2.2  DESCENT IN THE ERROR OF $\mathbf{y}^*\left(\mathbf{x}\right)$

**Lemma C.6.** *Under Assumptions 3.2–3.4, the approximation error of $\mathbf{y}^*\left(\mathbf{x}\right)$ of Algorithm 1 satisfies the following inequality:*

$$\mathbb{E}\left[ \left\| \mathbf{y}_t^{n+1} - \mathbf{y}^*\left(\mathbf{x}_t^{n+1}\right) \right\|^2 \right]$$

$$\leq (1 + c_1)(1 + c_2)\left( 1 - 2\beta_t \frac{\mu_g L_g}{\mu_g + L_g} \right) \mathbb{E}\left[ \left\| \mathbf{y}_t^n - \mathbf{y}^*\left(\mathbf{x}_t^n\right) \right\|^2 \right] + (1 + c_1)\left( 1 + \frac{1}{c_2} \right) \beta_t^2 \sigma_{g_y}^2$$

$$+ (1 + c_1)(1 + c_2)\left( \beta_t^2 - \frac{2\beta_t}{\mu_g + L_g} \right) \mathbb{E}\left[ \left\| \nabla_{\mathbf{y}} g\left(\mathbf{x}_t^n, \mathbf{y}_t^n\right) \right\|^2 \right] + \left( 1 + \frac{1}{c_1} \right) L_y^2 \alpha_t^2 \mathbb{E}\left[ \left\| h_{t,n}^f \right\|^2 \right],$$

*for all $t \in \{0, 1, \ldots, T-1\}$ and $n \in \{0, 1, \ldots, N-1\}$ with some constants $c_1, c_2 > 0$, where the expectation is taken over the stochasticity of the algorithm.*

*Proof.* We have

$$
\mathbb{E}\left[\left\|\mathbf{y}_t^{n+1} - \mathbf{y}^*\left(\mathbf{x}_t^{n+1}\right)\right\|^2\right]
$$

$$
\overset{(a)}{\leq} \mathbb{E}\left[(1+c_1)\left\|\mathbf{y}_t^{n+1} - \mathbf{y}^*\left(\mathbf{x}_t^n\right)\right\|^2 + \left(1 + \frac{1}{c_1}\right)\left\|\mathbf{y}^*\left(\mathbf{x}_t^n\right) - \mathbf{y}^*\left(\mathbf{x}_t^{n+1}\right)\right\|^2\right]
$$

$$
\overset{(b)}{\leq} \mathbb{E}\left[(1+c_1)\left\|\mathbf{y}_t^n - \beta_t h_{t,n}^g - \mathbf{y}^*\left(\mathbf{x}_t^n\right)\right\|^2 + \left(1 + \frac{1}{c_1}\right)L_y^2\left\|\mathbf{x}_t^{n+1} - \mathbf{x}_t^n\right\|^2\right]
$$

$$
\overset{(c)}{\leq} (1+c_1)(1+c_2)\mathbb{E}\left[\left\|\mathbf{y}_t^n - \beta_t\nabla_{\mathbf{y}}g\left(\mathbf{x}_t^n, \mathbf{y}_t^n\right) - \mathbf{y}^*\left(\mathbf{x}_t^n\right)\right\|^2\right]
$$

$$
+ (1+c_1)\left(1 + \frac{1}{c_2}\right)\beta_t^2\mathbb{E}\left[\left\|h_{t,n}^g - \nabla_{\mathbf{y}}g\left(\mathbf{x}_t^n, \mathbf{y}_t^n\right)\right\|^2\right] + \left(1 + \frac{1}{c_1}\right)L_y^2\alpha_t^2\mathbb{E}\left[\left\|h_{t,n}^f\right\|^2\right]
$$

$$
\overset{(d)}{\leq} (1+c_1)(1+c_2)\mathbb{E}\left[\left\|\mathbf{y}_t^n - \beta_t\nabla_{\mathbf{y}}g\left(\mathbf{x}_t^n, \mathbf{y}_t^n\right) - \mathbf{y}^*\left(\mathbf{x}_t^n\right)\right\|^2\right] + \left(1 + \frac{1}{c_1}\right)L_y^2\alpha_t^2\mathbb{E}\left[\left\|h_{t,n}^f\right\|^2\right]
$$

$$
+ (1+c_1)\left(1 + \frac{1}{c_2}\right)\beta_t^2\sigma_{g_y}^2, \tag{13}
$$

where (a) results from Young's inequality. (b) is because of the update rule of Algorithm 1 and the Lipschitzness of $\mathbf{y}^*(\cdot)$ (see Lemma 5.1). (c) follows from Young's inequality and the update rule of Algorithm 1. (d) uses the bounded variance in Assumption 3.4.

To bound the first term on the right, we have

$$
\left\|\mathbf{y}_t^n - \beta_t\nabla_{\mathbf{y}}g\left(\mathbf{x}_t^n, \mathbf{y}_t^n\right) - \mathbf{y}^*\left(\mathbf{x}_t^n\right)\right\|^2
$$

$$
= \left\|\mathbf{y}_t^n - \mathbf{y}^*\left(\mathbf{x}_t^n\right)\right\|^2 + \beta_t^2\left\|\nabla_{\mathbf{y}}g\left(\mathbf{x}_t^n, \mathbf{y}_t^n\right)\right\|^2 - 2\beta_t\left\langle\nabla_{\mathbf{y}}g\left(\mathbf{x}_t^n, \mathbf{y}_t^n\right), \mathbf{y}_t^n - \mathbf{y}^*\left(\mathbf{x}_t^n\right)\right\rangle
$$

$$
\overset{(a)}{\leq} \left(1 - 2\beta_t\frac{\mu_g L_g}{\mu_g + L_g}\right)\left\|\mathbf{y}_t^n - \mathbf{y}^*\left(\mathbf{x}_t^n\right)\right\|^2 + \left(\beta_t^2 - \frac{2\beta_t}{\mu_g + L_g}\right)\left\|\nabla_{\mathbf{y}}g\left(\mathbf{x}_t^n, \mathbf{y}_t^n\right)\right\|^2, \tag{14}
$$

where (a) is due to $\mu_g$-strongly convexity and $L_g$-smoothness of the lower-level function $g(\mathbf{x}, \mathbf{y})$ (see Assumption 3.2), which implies

$$
\left\langle\nabla_{\mathbf{y}}g\left(\mathbf{x}_t^n, \mathbf{y}_t^n\right), \mathbf{y}_t^n - \mathbf{y}^*\left(\mathbf{x}_t^n\right)\right\rangle \geq \frac{\mu_g L_g}{\mu_g + L_g}\left\|\mathbf{y}_t^n - \mathbf{y}^*\left(\mathbf{x}_t^n\right)\right\|^2 + \frac{1}{\mu_g + L_g}\left\|\nabla_{\mathbf{y}}g\left(\mathbf{x}_t^n, \mathbf{y}_t^n\right)\right\|^2.
$$

The Lemma is proved by substituting (14) in (13). $\qquad\square$

### C.2.3 DESCENT IN THE ERROR OF $\mathbf{z}^*(\mathbf{x}, \mathbf{y})$

**Lemma C.7.** *Under Assumptions 3.1–3.4, the following inequality of the approximation error of $\mathbf{z}^*(\mathbf{x}, \mathbf{y})$ holds for Algorithm 1:*

$$
\mathbb{E}\left[\left\|\mathbf{z}_{t+1} - \mathbf{z}_{t+1}^*\right\|^2\right]
$$

$$
\leq \left((1+c_3)(1+c_4)\left(1 - \frac{2\gamma_t\mu_g L_q}{\mu_g + L_q}\right) + 4\sigma_{g_{yy}}^2\gamma_t^2(1+c_3)\left(1 + \frac{1}{c_4}\right)\right)\mathbb{E}\left[\left\|\mathbf{z}_t - \mathbf{z}_t^*\right\|^2\right]
$$

$$
+ (1+c_3)(1+c_4)\left(\gamma_t^2 - \frac{2\gamma_t}{\mu_g + L_q}\right)\mathbb{E}\left[\left\|\nabla_{\mathbf{z}}q\left(\mathbf{x}_t^N, \mathbf{y}_t^N, \mathbf{z}_t\right)\right\|^2\right] + 4\left(1 + \frac{1}{c_3}\right)L_z^2\beta_t^2 N^2\sigma_{g_y}^2
$$

$$
+ 2\left(1 + \frac{1}{c_3}\right)L_z^2\alpha_t^2 N\sum_{n=0}^{N-1}\mathbb{E}\left[\left\|h_{t,n}^f\right\|^2\right] + 4\left(1 + \frac{1}{c_3}\right)L_z^2\beta_t^2 N\sum_{n=0}^{N-1}\mathbb{E}\left[\left\|\nabla_{\mathbf{y}}g\left(\mathbf{x}_t^n, \mathbf{y}_t^n\right)\right\|^2\right]
$$

$$
+ 4\sigma_{g_{yy}}^2\frac{B_{f_y}^2}{\mu_g^2}(1+c_3)\left(1 + \frac{1}{c_4}\right)\gamma_t^2 + 2\sigma_{f_y}^2(1+c_3)\left(1 + \frac{1}{c_4}\right)\gamma_t^2,
$$

*for all $t \in \{0, 1, \ldots, T-1\}$ and $n \in \{0, 1, \ldots, N-1\}$ with some constants $c_3, c_4 > 0$, where $\mathbf{z}_t = \mathbf{z}\left(\mathbf{x}_t^0, \mathbf{y}_t^0\right)$ and $\mathbf{z}_t^* = \mathbf{z}^*\left(\mathbf{x}_t^0, \mathbf{y}_t^0\right)$. The expectation is taken over the stochasticity of the algorithm.*

*Proof.* We have

$$\mathbb{E}\left[\left\|\mathbf{z}_{t+1} - \mathbf{z}_{t+1}^*\right\|^2\right]$$

$$\overset{(a)}{\leq} \mathbb{E}\left[(1+c_3)\left\|\mathbf{z}_{t+1} - \mathbf{z}_t^*\right\|^2 + \left(1 + \frac{1}{c_3}\right)\left\|\mathbf{z}^*\left(\mathbf{x}_{t+1}^0, \mathbf{y}_{t+1}^0\right) - \mathbf{z}^*\left(\mathbf{x}_t^0, \mathbf{y}_t^0\right)\right\|^2\right]$$

$$\overset{(b)}{\leq} \mathbb{E}\left[(1+c_3)\left\|\mathbf{z}_{t+1} - \mathbf{z}_t^*\right\|^2 + \left(1 + \frac{1}{c_3}\right)L_z^2\left(\left\|\mathbf{x}_{t+1}^0 - \mathbf{x}_t^0\right\| + \left\|\mathbf{y}_{t+1}^0 - \mathbf{y}_t^0\right\|\right)^2\right]$$

$$\overset{(c)}{\leq} \mathbb{E}\left[(1+c_3)\left\|\mathbf{z}_{t+1} - \mathbf{z}_t^*\right\|^2 + 2\left(1 + \frac{1}{c_3}\right)L_z^2\left\|\mathbf{x}_t^N - \mathbf{x}_t^0\right\|^2 + 2\left(1 + \frac{1}{c_3}\right)L_z^2\left\|\mathbf{y}_t^N - \mathbf{y}_t^0\right\|^2\right]$$

$$\overset{(d)}{\leq} (1+c_3)\mathbb{E}\left[\left\|\mathbf{z}_{t+1} - \mathbf{z}_t^*\right\|^2\right] + 2\left(1 + \frac{1}{c_3}\right)L_z^2\alpha_t^2 N \sum_{n=0}^{N-1}\mathbb{E}\left[\left\|h_{t,n}^f\right\|^2\right]$$

$$+ 2\left(1 + \frac{1}{c_3}\right)L_z^2\beta_t^2 N \sum_{n=0}^{N-1}\mathbb{E}\left[\left\|h_{t,n}^g\right\|^2\right]$$

$$\overset{(e)}{\leq} (1+c_3)\mathbb{E}\left[\left\|\mathbf{z}_{t+1} - \mathbf{z}_t^*\right\|^2\right] + 4\left(1 + \frac{1}{c_3}\right)L_z^2\beta_t^2 N \sum_{n=0}^{N-1}\mathbb{E}\left[\left\|\nabla_{\mathbf{y}}g\left(\mathbf{x}_t^n, \mathbf{y}_t^n\right)\right\|^2\right]$$

$$+ 4\left(1 + \frac{1}{c_3}\right)L_z^2\beta_t^2 N \sum_{n=0}^{N-1}\mathbb{E}\left[\left\|h_{t,n}^g - \nabla_{\mathbf{y}}g\left(\mathbf{x}_t^n, \mathbf{y}_t^n\right)\right\|^2\right] + 2\left(1 + \frac{1}{c_3}\right)L_z^2\alpha_t^2 N \sum_{n=0}^{N-1}\mathbb{E}\left[\left\|h_{t,n}^f\right\|^2\right]$$

$$\overset{(f)}{\leq} (1+c_3)\mathbb{E}\left[\left\|\mathbf{z}_{t+1} - \mathbf{z}_t^*\right\|^2\right] + 2\left(1 + \frac{1}{c_3}\right)L_z^2\alpha_t^2 N \sum_{n=0}^{N-1}\mathbb{E}\left[\left\|h_{t,n}^f\right\|^2\right]$$

$$+ 4\left(1 + \frac{1}{c_3}\right)L_z^2\beta_t^2 N \sum_{n=0}^{N-1}\mathbb{E}\left[\left\|\nabla_{\mathbf{y}}g\left(\mathbf{x}_t^n, \mathbf{y}_t^n\right)\right\|^2\right] + 4\left(1 + \frac{1}{c_3}\right)L_z^2\beta_t^2 N^2\sigma_{g_y}^2, \tag{15}$$

where (a) follows from Young's inequality. (b) is due to the Lipschitzness of $\mathbf{z}^*\left(\cdot, \cdot\right)$ (see Lemma 5.2). (c) and (e) result from $\|x + y\|^2 \leq 2\|x\|^2 + 2\|y\|^2$. (d) is because of equations (9) and (10). (f) uses the bounded variance in Assumption 3.4.

Next, we bound the first term on the right:

$$\mathbb{E}\left[\left\|\mathbf{z}_{t+1} - \mathbf{z}_t^*\right\|^2\right] \overset{(a)}{=} \mathbb{E}\left[\left\|\mathbf{z}_t - \gamma_t h_t^q - \mathbf{z}_t^*\right\|^2\right]$$

$$\overset{(b)}{\leq} \mathbb{E}\left[(1+c_4)\left\|\mathbf{z}_t - \gamma_t \nabla_{\mathbf{z}}q\left(\mathbf{x}_t^N, \mathbf{y}_t^N, \mathbf{z}_t\right) - \mathbf{z}_t^*\right\|^2 + \left(1 + \frac{1}{c_4}\right)\gamma_t^2\left\|\nabla_{\mathbf{z}}q\left(\mathbf{x}_t^N, \mathbf{y}_t^N, \mathbf{z}_t\right) - h_t^q\right\|^2\right]$$

$$= \mathbb{E}\left[(1+c_4)\left(\left\|\mathbf{z}_t - \mathbf{z}_t^*\right\|^2 + \gamma_t^2\left\|\nabla_{\mathbf{z}}q\left(\mathbf{x}_t^N, \mathbf{y}_t^N, \mathbf{z}_t\right)\right\|^2 - 2\gamma_t\left\langle\nabla_{\mathbf{z}}q\left(\mathbf{x}_t^N, \mathbf{y}_t^N, \mathbf{z}_t\right), \mathbf{z}_t - \mathbf{z}_t^*\right\rangle\right)\right]$$

$$+ \left(1 + \frac{1}{c_4}\right)\gamma_t^2\mathbb{E}\left[\left\|\nabla_{\mathbf{z}}q\left(\mathbf{x}_t^N, \mathbf{y}_t^N, \mathbf{z}_t\right) - h_t^q\right\|^2\right]$$

$$\overset{(c)}{\leq} (1+c_4)\left(1 - 2\gamma_t\frac{\mu_g L_q}{\mu_g + L_q}\right)\mathbb{E}\left[\left\|\mathbf{z}_t - \mathbf{z}_t^*\right\|^2\right] + \left(1 + \frac{1}{c_4}\right)\gamma_t^2\mathbb{E}\left[\left\|\nabla_{\mathbf{z}}q\left(\mathbf{x}_t^N, \mathbf{y}_t^N, \mathbf{z}_t\right) - h_t^q\right\|^2\right]$$

$$+ (1+c_4)\left(\gamma_t^2 - \frac{2\gamma_t}{\mu_g + L_q}\right)\mathbb{E}\left[\left\|\nabla_{\mathbf{z}}q\left(\mathbf{x}_t^N, \mathbf{y}_t^N, \mathbf{z}_t\right)\right\|^2\right], \tag{16}$$

where (a) results from the update rule of Algorithm 1, and (b) uses Young's inequality. (c) follows from $\mu_g$-strongly convexity and $L_q$-smoothness of $q\left(\mathbf{x}, \mathbf{y}, \mathbf{z}\right)$, which implies

$$\left\langle\nabla_{\mathbf{z}}q\left(\mathbf{x}_t^N, \mathbf{y}_t^N, \mathbf{z}_t\right), \mathbf{z}_t - \mathbf{z}_t^*\right\rangle \geq \frac{\mu_g L_q}{\mu_g + L_q}\left\|\mathbf{z}_t - \mathbf{z}_t^*\right\|^2 + \frac{1}{\mu_g + L_q}\left\|\nabla_{\mathbf{z}}q\left(\mathbf{x}_t^N, \mathbf{y}_t^N, \mathbf{z}_t\right)\right\|^2.$$

Then, we bound the second term on the right as follows:

$$\mathbb{E}\left[\left\|\nabla_{\mathbf{z}}q\left(\mathbf{x}_t^N,\mathbf{y}_t^N,\mathbf{z}_t\right)-h_t^q\right\|^2\right]$$

$$\stackrel{(a)}{=}\mathbb{E}\left[\left\|\nabla_{\mathbf{yy}}^2g\left(\mathbf{x}_t^N,\mathbf{y}_t^N\right)\mathbf{z}_t+\nabla_{\mathbf{y}}f\left(\mathbf{x}_t^N,\mathbf{y}_t^N\right)-\left(\nabla_{\mathbf{yy}}^2g\left(\mathbf{x}_t^N,\mathbf{y}_t^N;\mathcal{D}_t^{g_{yy}}\right)\mathbf{z}_t+\nabla_{\mathbf{y}}f\left(\mathbf{x}_t^N,\mathbf{y}_t^N;\mathcal{D}_t^{f_y}\right)\right)\right\|^2\right]$$

$$\stackrel{(b)}{\leq}2\mathbb{E}\left[\left\|\mathbf{z}_t\right\|^2\left\|\nabla_{\mathbf{yy}}^2g\left(\mathbf{x}_t^N,\mathbf{y}_t^N\right)-\nabla_{\mathbf{yy}}^2g\left(\mathbf{x}_t^N,\mathbf{y}_t^N;\mathcal{D}_t^{g_{yy}}\right)\right\|^2+\left\|\nabla_{\mathbf{y}}f\left(\mathbf{x}_t^N,\mathbf{y}_t^N\right)-\nabla_{\mathbf{y}}f\left(\mathbf{x}_t^N,\mathbf{y}_t^N;\mathcal{D}_t^{f_y}\right)\right\|^2\right]$$

$$\stackrel{(c)}{\leq}\mathbb{E}\left[2\sigma_{g_{yy}}^2\left\|\mathbf{z}_t-\mathbf{z}_t^*+\mathbf{z}_t^*\right\|^2+2\sigma_{f_y}^2\right]$$

$$\stackrel{(d)}{\leq}\mathbb{E}\left[4\sigma_{g_{yy}}^2\left\|\mathbf{z}_t-\mathbf{z}_t^*\right\|^2+4\sigma_{g_{yy}}^2\left\|\mathbf{z}_t^*\right\|^2+2\sigma_{f_y}^2\right]$$

$$\stackrel{(e)}{\leq}4\sigma_{g_{yy}}^2\mathbb{E}\left[\left\|\mathbf{z}_t-\mathbf{z}_t^*\right\|^2\right]+4\sigma_{g_{yy}}^2\frac{B_{f_y}^2}{\mu_g^2}+2\sigma_{f_y}^2,\tag{17}$$

where (a) follows from the definitions of $h_t^q$ and $\nabla_{\mathbf{z}}q\left(\mathbf{x},\mathbf{y},\mathbf{z}\right)$. (b) and (d) are because of $\|x+y\|^2\leq 2\|x\|^2+2\|y\|^2$. (c) results from the bounded variances in Assumption 3.4. (e) utilizes the bound of $\mathbf{z}^*\left(\mathbf{x},\mathbf{y}\right)$ in Lemma 5.2.

Substituting (17) in (16) and then substituting the result in (15), the lemma is proved. $\qquad\square$

### C.2.4  Descent in the potential function

We define the potential function $W_t$ as follows:

$$W_t=\ell\left(\mathbf{x}_t^0\right)+K_y\left\|\mathbf{y}_t^0-\mathbf{y}^*\left(x_t^0\right)\right\|^2+K_z\left\|\mathbf{z}_t\left(\mathbf{x}_t^0,\mathbf{y}_t^0\right)-\mathbf{z}^*\left(\mathbf{x}_t^0,\mathbf{y}_t^0\right)\right\|^2$$

**Lemma C.8.** *Set* $c_1=\frac{\beta_t L_{\mu_g}}{2(1-\beta_t L_{\mu_g})}$, $c_2=\frac{\beta_t L_{\mu_g}}{1-2\beta_t L_{\mu_g}}$, $c_3=\frac{\gamma_t L_{\mu_q}}{2(1-\gamma_t L_{\mu_q})}$, *and* $c_4=\frac{\gamma_t L_{\mu_q}}{1-2\gamma_t L_{\mu_q}}$. *Under the same conditions as described in Theorem C.9 and using Lemmas C.1-C.3, the iterates generated by Algorithm 1 satisfies: for all* $t\in\{0,1,\dots,T-1\}$,

$$\mathbb{E}\left[W_{t+1}-W_t\right]\leq-\frac{\alpha_t}{2}\sum_{n=0}^{N-1}\mathbb{E}\left[\left\|\nabla\ell\left(\mathbf{x}_t^n\right)\right\|^2\right]+\sigma_{g_{xy}}^2\alpha_t C_{g_{xy}}N+\sigma_{f_x}^2\alpha_t C_{f_x}N+\sigma_{g_{yy}}^2\alpha_t C_{g_{yy}}N$$

$$+\sigma_{f_y}^2\alpha_t C_{f_y}N+\sigma_{g_y}^2\alpha_t\left(C_{g_1}N+C_{g_2}\frac{1}{N}\right),$$

*where the constant values* $C_{g_{xy}}$, $C_{f_x}$, $C_{g_1}$, $C_{g_2}$, $C_{g_{yy}}$ *and* $C_{f_y}$, *which are independent of N, are defined as:*

$$C_{g_{xy}}=\frac{4B_{f_y}^2}{\mu_g^2},\qquad C_{f_x}=2,\qquad C_{g_1}=\frac{2c_\beta K_y}{L_{\mu_g}}+\frac{8L_z^2c_\beta^2NK_z}{c_\gamma L_{\mu_q}},$$

$$C_{g_2}=\frac{L_l^2c_\beta^2}{16L_f^2L_z^2},\qquad C_{g_{yy}}=\frac{8B_{f_y}^2c_\gamma K_z}{\mu_g^2L_{\mu_q}N},\qquad C_{f_y}=\frac{4c_\gamma K_z}{L_{\mu_q}N},\tag{18}$$

*where* $K_y$ *and* $K_z$ *are defined in (23) of Theorem C.9.*

*Proof.* From Lemma C.1, we have

$$\sum_{n=0}^{N-1}\mathbb{E}\left[\ell\left(\mathbf{x}_t^{n+1}\right)-\ell\left(\mathbf{x}_t^n\right)\right]=\mathbb{E}\left[\ell\left(\mathbf{x}_{t+1}^0\right)-\ell\left(\mathbf{x}_t^0\right)\right]$$

$$\leq-\frac{\alpha_t}{2}\sum_{n=0}^{N-1}\mathbb{E}\left[\left\|\nabla\ell\left(\mathbf{x}_t^n\right)\right\|^2\right]+\left(-\frac{\alpha_t}{2}+\frac{\alpha_t^2L_l}{2}+8L_f^2L_z^2\alpha_t^3N^2\right)\sum_{n=0}^{N-1}\mathbb{E}\left[\left\|h_{t,n}^f\right\|^2\right]+2\sigma_{f_x}^2N\alpha_t$$

$$+\left(4\sigma_{g_{xy}}^2N\alpha_t+4L_f^2N\alpha_t\right)\mathbb{E}\left[\left\|\mathbf{z}_t-\mathbf{z}_t^*\right\|^2\right]+16L_f^2L_z^2\beta_t^2N^2\alpha_t\sum_{n=0}^{N-1}\mathbb{E}\left[\left\|\nabla_{\mathbf{y}}g\left(\mathbf{x}_t^n,\mathbf{y}_t^n\right)\right\|^2\right]$$

$$+ 2\alpha_t L_f^2 \sum_{n=0}^{N-1} \mathbb{E}\left[\|\mathbf{y}_t^n - \mathbf{y}^*(\mathbf{x}_t^n)\|^2\right] + 4\tilde{\sigma}_{g_{xy}}^2 |\mathcal{D}^{g_{xy}}|^{-1} \frac{B_{f_y}^2}{\mu_g^2} N\alpha_t + 16 L_f^2 L_z^2 \beta_t^2 N^3 \sigma_{g_y}^2 \alpha_t.$$

Choosing $\alpha_t \le \frac{L_l}{16 L_f^2 L_z^2 N^2}$ and using the definition of $\beta_t = c_\beta \alpha_t$, we get

$$\mathbb{E}\left[\ell\left(\mathbf{x}_{t+1}^0\right) - \ell\left(\mathbf{x}_t^0\right)\right] \le -\frac{\alpha_t}{2} \sum_{n=0}^{N-1} \mathbb{E}\left[\|\nabla\ell(\mathbf{x}_t^n)\|^2\right] + \left(4\sigma_{g_{xy}}^2 N\alpha_t + 4L_f^2 N\alpha_t\right) \mathbb{E}\left[\|\mathbf{z}_t - \mathbf{z}_t^*\|^2\right]$$

$$+ \left(-\frac{\alpha_t}{2} + \alpha_t^2 L_l\right) \sum_{n=0}^{N-1} \mathbb{E}\left[\left\|h_{t,n}^f\right\|^2\right] + 2\alpha_t L_f^2 \sum_{n=0}^{N-1} \mathbb{E}\left[\|\mathbf{y}_t^n - \mathbf{y}^*(\mathbf{x}_t^n)\|^2\right] + 2\sigma_{f_x}^2 N\alpha_t$$

$$+ 16 L_f^2 L_z^2 c_\beta^2 N^2 \alpha_t^3 \sum_{n=0}^{N-1} \mathbb{E}\left[\|\nabla_{\mathbf{y}} g(\mathbf{x}_t^n, \mathbf{y}_t^n)\|^2\right] + \frac{L_l^2 c_\beta^2}{16 L_f^2 L_z^2 N} \sigma_{g_y}^2 \alpha_t + 4\sigma_{g_{xy}}^2 \frac{B_{f_y}^2}{\mu_g^2} N\alpha_t. \tag{19}$$

With the result from Lemma C.2, we have

$$\sum_{n=0}^{N-1} \mathbb{E}\left[\left\|\mathbf{y}_t^{n+1} - \mathbf{y}^*(\mathbf{x}_t^{n+1})\right\|^2 - \|\mathbf{y}_t^n - \mathbf{y}^*(\mathbf{x}_t^n)\|^2\right] = \mathbb{E}\left[\left\|\mathbf{y}_t^0 - \mathbf{y}^*(\mathbf{x}_t^0)\right\|^2 - \left\|\mathbf{y}_t^0 - \mathbf{y}^*(\mathbf{x}_t^0)\right\|^2\right]$$

$$\le \left((1+c_1)(1+c_2)\left(1 - 2\beta_t \frac{\mu_g L_g}{\mu_g + L_g}\right) - 1\right) \sum_{n=0}^{N-1} \mathbb{E}\left[\|\mathbf{y}_t^n - \mathbf{y}^*(\mathbf{x}_t^n)\|^2\right]$$

$$+ \left(1 + \frac{1}{c_1}\right) L_y^2 \alpha_t^2 \sum_{n=0}^{N-1} \mathbb{E}\left[\left\|h_{t,n}^f\right\|^2\right] + (1+c_1)\left(1 + \frac{1}{c_2}\right) \beta_t^2 \tilde{\sigma}_g^2 |\mathcal{D}^{g_y}|^{-1} N$$

$$+ (1+c_1)(1+c_2)\left(\beta_t^2 - \frac{2\beta_t}{\mu_g + L_g}\right) \sum_{n=0}^{N-1} \mathbb{E}\left[\|\nabla_{\mathbf{y}} g(\mathbf{x}_t^n, \mathbf{y}_t^n)\|^2\right].$$

Denote $L_{\mu_g} = \frac{\mu_g L_g}{\mu_g + L_g}$. Choose $c_1$ and $c_2$ such that

$$(1+c_1)(1+c_2)\left(1 - 2\beta_t L_{\mu_g}\right) = 1 - \frac{\beta_t L_{\mu_g}}{2}.$$

Let
$$(1+c_2)\left(1 - 2\beta_t L_{\mu_g}\right) = 1 - \beta_t L_{\mu_g} \implies c_2 = \frac{\beta_t L_{\mu_g}}{1 - 2\beta_t L_{\mu_g}} \ \& \ \beta_t \le \frac{1}{2L_{\mu_g}}.$$

Thus,
$$c_1 = \frac{\beta_t L_{\mu_g}}{2\left(1 - \beta_t L_{\mu_g}\right)}.$$

Moreover, this implies that

$$1 + \frac{1}{c_2} = 1 + \frac{1 - 2\beta_t L_{\mu_g}}{\beta_t L_{\mu_g}} \le \frac{1}{\beta_t L_{\mu_g}}, \qquad 1 + \frac{1}{c_1} = \frac{2\left(1 - \beta_t L_{\mu_g}\right)}{\beta_t L_{\mu_g}} \le \frac{2}{\beta_t L_{\mu_g}}.$$

Use the definition of $\beta_t = c_\beta \alpha_t$. Substituting $c_1$ and $c_2$ and choosing $\beta_t \le \frac{1}{\mu_g + L_g}$, we have

$$\mathbb{E}\left[\left\|\mathbf{y}_{t+1}^0 - \mathbf{y}^*(\mathbf{x}_{t+1}^0)\right\|^2 - \left\|\mathbf{y}_t^0 - \mathbf{y}^*(\mathbf{x}_t^0)\right\|^2\right] \le -\frac{c_\beta L_{\mu_g}}{2} \alpha_t \sum_{n=0}^{N-1} \mathbb{E}\left[\|\mathbf{y}_t^n - \mathbf{y}^*(\mathbf{x}_t^n)\|^2\right]$$

$$+ \frac{2L_y^2 \alpha_t}{c_\beta L_{\mu_g}} \sum_{n=0}^{N-1} \mathbb{E}\left[\left\|h_{t,n}^f\right\|^2\right] - \frac{c_\beta \alpha_t}{\mu_g + L_g} \sum_{n=0}^{N-1} \mathbb{E}\left[\|\nabla_{\mathbf{y}} g(\mathbf{x}_t^n, \mathbf{y}_t^n)\|^2\right] + \frac{2}{L_{\mu_g}} c_\beta \alpha_t \sigma_{g_y}^2 N. \tag{20}$$

According to Lemma C.3, we have

$$\mathbb{E}\left[\left\|\mathbf{z}_{t+1} - \mathbf{z}_{t+1}^*\right\|^2 - \left\|\mathbf{z}_t - \mathbf{z}_t^*\right\|^2\right]$$

$$= \mathbb{E}\left[\left\|\mathbf{z}\left(\mathbf{x}_{t+1}^0, \mathbf{y}_{t+1}^0\right) - \mathbf{z}^*\left(\mathbf{x}_{t+1}^0, \mathbf{y}_{t+1}^0\right)\right\|^2 - \left\|\mathbf{z}\left(\mathbf{x}_t^0, \mathbf{y}_t^0\right) - \mathbf{z}^*\left(\mathbf{x}_t^0, \mathbf{y}_t^0\right)\right\|^2\right]$$

$$\leq \left((1+c_3)(1+c_4)\left(1 - \frac{2\gamma_t \mu_g L_q}{\mu_g + L_q}\right) + 4\sigma_{g_{yy}}^2 \gamma_t^2 (1+c_3)\left(1 + \frac{1}{c_4}\right) - 1\right)\mathbb{E}\left[\left\|\mathbf{z}_t - \mathbf{z}_t^*\right\|^2\right]$$

$$+ (1+c_3)(1+c_4)\left(\gamma_t^2 - \frac{2\gamma_t}{\mu_g + L_q}\right)\mathbb{E}\left[\left\|\nabla_{\mathbf{z}} q\left(\mathbf{x}_t^N, \mathbf{y}_t^N, \mathbf{z}_t\right)\right\|^2\right] + 4\left(1 + \frac{1}{c_3}\right)L_z^2 \beta_t^2 N^2 \sigma_{g_y}^2$$

$$+ 2\left(1 + \frac{1}{c_3}\right)L_z^2 \alpha_t^2 N \sum_{n=0}^{N-1}\mathbb{E}\left[\left\|h_{t,n}^f\right\|^2\right] + 4\left(1 + \frac{1}{c_3}\right)L_z^2 \beta_t^2 N \sum_{n=0}^{N-1}\mathbb{E}\left[\left\|\nabla_{\mathbf{y}} g\left(\mathbf{x}_t^n, \mathbf{y}_t^n\right)\right\|^2\right]$$

$$+ 2\sigma_{f_y}^2 (1+c_3)\left(1 + \frac{1}{c_4}\right)\gamma_t^2 + 4\sigma_{g_{yy}}^2 \frac{B_{f_y}^2}{\mu_g^2}(1+c_3)\left(1 + \frac{1}{c_4}\right)\gamma_t^2.$$

Similar as $c_1$ and $c_2$, we choose

$$c_3 = \frac{\gamma_t L_{\mu_q}}{2(1 - \gamma_t L_{\mu_q})}, \qquad c_4 = \frac{\gamma_t L_{\mu_q}}{1 - 2\gamma_t L_{\mu_q}},$$

where $\gamma_t \leq \frac{1}{2L_{\mu_q}}$ and we denote $L_{\mu_q} = \frac{\mu_g L_q}{\mu_g + L_q}$. This implies that

$$1 + \frac{1}{c_4} \leq \frac{1}{\gamma_t L_{\mu_q}}, \qquad 1 + \frac{1}{c_3} \leq \frac{2}{\gamma_t L_{\mu_q}}.$$

According to the definitions of $\beta_t = c_\beta \alpha_t$ and $\gamma_t = c_\gamma \alpha_t$, substituting $c_3$ and $c_4$ and choosing $\gamma_t \leq \frac{1}{\mu_g + L_q}$, we get

$$\mathbb{E}\left[\left\|\mathbf{z}\left(\mathbf{x}_{t+1}^0, \mathbf{y}_{t+1}^0\right) - \mathbf{z}^*\left(\mathbf{x}_{t+1}^0, \mathbf{y}_{t+1}^0\right)\right\|^2 - \left\|\mathbf{z}\left(\mathbf{x}_t^0, \mathbf{y}_t^0\right) - \mathbf{z}^*\left(\mathbf{x}_t^0, \mathbf{y}_t^0\right)\right\|^2\right]$$

$$\leq \left(-\frac{c_\gamma L_{\mu_q}}{2}\alpha_t + \frac{8}{L_{\mu_q}}\sigma_{g_{yy}}^2 c_\gamma \alpha_t\right)\mathbb{E}\left[\left\|\mathbf{z}_t - \mathbf{z}_t^*\right\|^2\right] - \frac{c_\gamma \alpha_t}{\mu_g + L_q}\mathbb{E}\left[\left\|\nabla_{\mathbf{z}} q\left(\mathbf{x}_t^N, \mathbf{y}_t^N, \mathbf{z}_t\right)\right\|^2\right]$$

$$+ \frac{4L_z^2 \alpha_t N}{c_\gamma L_{\mu_q}}\sum_{n=0}^{N-1}\mathbb{E}\left[\left\|h_{t,n}^f\right\|^2\right] + \frac{8L_z^2 c_\beta^2 \alpha_t N}{c_\gamma L_{\mu_q}}\sum_{n=0}^{N-1}\mathbb{E}\left[\left\|\nabla_{\mathbf{y}} g\left(\mathbf{x}_t^n, \mathbf{y}_t^n\right)\right\|^2\right] + \frac{8}{c_\gamma L_{\mu_q}}L_z^2 c_\beta^2 \alpha_t N^2 \sigma_{g_y}^2$$

$$+ 2\sigma_{f_y}^2 \frac{2}{L_{\mu_q}}c_\gamma \alpha_t + 4\sigma_{g_{yy}}^2 \frac{B_{f_y}^2}{\mu_g^2}\frac{2}{L_{\mu_q}}c_\gamma \alpha_t. \tag{21}$$

Adding equations (19), (20) and (21), we get

$$\mathbb{E}\left[W_{t+1} - W_t\right]$$

$$\leq -\frac{\alpha_t}{2}\sum_{n=0}^{N-1}\mathbb{E}\left[\left\|\nabla l\left(\mathbf{x}_t^n\right)\right\|^2\right] + \bar{C}_y \sum_{n=0}^{N-1}\mathbb{E}\left[\left\|\mathbf{y}_t^n - \mathbf{y}^*\left(\mathbf{x}_t^n\right)\right\|^2\right] + \bar{C}_z \mathbb{E}\left[\left\|\mathbf{z}_t - \mathbf{z}_t^*\right\|^2\right]$$

$$+ \bar{C}_g \sum_{n=0}^{N-1}\mathbb{E}\left[\left\|\nabla_{\mathbf{y}} g\left(\mathbf{x}_t^n, \mathbf{y}_t^n\right)\right\|^2\right] + \bar{C}_h \sum_{n=0}^{N-1}\mathbb{E}\left[\left\|h_{t,n}^f\right\|^2\right] + \bar{C}_q \mathbb{E}\left[\left\|\nabla_{\mathbf{z}} q\left(\mathbf{x}_t^N, \mathbf{y}_t^N, \mathbf{z}_t\right)\right\|^2\right]$$

$$+ \frac{L_l^2 c_\beta^2}{16 L_f^2 L_z^2 N}\sigma_{g_y}^2 \alpha_t + 4\sigma_{g_{xy}}^2 \frac{B_{f_y}^2}{\mu_g^2}N\alpha_t + 2\sigma_{f_x}^2 N\alpha_t + K_y \frac{2c_\beta}{L_{\mu_g}}\alpha_t \sigma_{g_y}^2 N$$

$$+ K_z \left(\frac{8B_{f_y}^2}{\mu_g^2 L_{\mu_q}}\sigma_{g_{yy}}^2 c_\gamma \alpha_t + \frac{4c_\gamma}{L_{\mu_q}}\sigma_{f_y}^2 \alpha_t + \frac{8}{c_\gamma L_{\mu_q}}L_z^2 c_\beta^2 \alpha_t N^2 \sigma_{g_y}^2\right),$$

where

$$\bar{C}_y = 2\alpha_t L_f^2 - \frac{c_\beta L_{\mu_g}}{2}\alpha_t K_y$$

$$\bar{C}_z = 4\alpha_t \sigma_{g_{xy}}^2 N + 4L_f^2 N\alpha_t - \frac{c_\gamma L_{\mu_q}}{2}\alpha_t K_z + \frac{8}{L_{\mu_q}}\sigma_{g_{yy}}^2 c_\gamma \alpha_t K_z$$

$$\bar{C}_g = 16L_f^2 L_z^2 c_\beta^2 N^2 \alpha_t^3 - \frac{c_\beta \alpha_t}{\mu_g + L_g}K_y + \frac{8}{c_\gamma L_{\mu_q}}L_z^2 c_\beta^2 \alpha_t N K_z$$

$$\bar{C}_h = \alpha_t^2 L_l - \frac{\alpha_t}{2} + \frac{2}{c_\beta L_{\mu_g}}L_y^2 \alpha_t K_y + \frac{4}{c_\gamma L_{\mu_q}}L_z^2 \alpha_t N K_z$$

$$\bar{C}_q = -\frac{c_\gamma \alpha_t}{\mu_g + L_q}K_z \leq 0.$$

To ensure $\bar{C}_y \leq 0$, we choose $K_y \geq \frac{4L_f^2}{c_\beta L_{\mu_g}}$.

To ensure $\bar{C}_z \leq 0$, we have

$$\bar{C}_z = 4\alpha_t \sigma_{g_{xy}}^2 N + 4L_f^2 N\alpha_t - \frac{c_\gamma L_{\mu_q}}{2}\alpha_t K_z + \frac{8}{L_{\mu_q}}\sigma_{g_{yy}}^2 c_\gamma \alpha_t K_z$$

$$= 4\alpha_t \sigma_{g_{xy}}^2 N + 4L_f^2 N\alpha_t - \frac{c_\gamma L_{\mu_q}}{2}\alpha_t K_z + \frac{8}{L_{\mu_q}}\tilde{\sigma}_{g_{yy}}^2 |\mathcal{D}^{g_{yy}}|^{-1} c_\gamma \alpha_t K_z$$

$$\overset{(a)}{\leq} \frac{c_\gamma L_{\mu_q}}{6}\alpha_t K_z + \frac{c_\gamma L_{\mu_q}}{6}\alpha_t K_z - \frac{c_\gamma L_{\mu_q}}{2}\alpha_t K_z + \frac{c_\gamma L_{\mu_q}}{6}\alpha_t K_z = 0,$$

where (a) utilizes $K_z \geq \max\left\{\frac{24\sigma_{g_{xy}}^2 N}{c_\gamma L_{\mu_q}}, \frac{24L_f^2 N}{c_\gamma L_{\mu_q}}\right\}$ and $|\mathcal{D}^{g_{yy}}| \geq \frac{48\tilde{\sigma}_{g_{yy}}^2}{L_{\mu_q}^2}$, which is the data batch.

To ensure $\bar{C}_g \leq 0$, we have

$$\bar{C}_g = 16L_f^2 L_z^2 c_\beta^2 N^2 \alpha_t^3 - \frac{c_\beta \alpha_t}{\mu_g + L_g}K_y + \frac{8}{c_\gamma L_{\mu_q}}L_z^2 c_\beta^2 \alpha_t N K_z$$

$$\overset{(a)}{\leq} \frac{1}{2}\frac{c_\beta \alpha_t}{\mu_g + L_g}K_y - \frac{c_\beta \alpha_t}{\mu_g + L_g}K_y + \frac{1}{2}\frac{c_\beta \alpha_t}{\mu_g + L_g}K_y = 0,$$

where (a) results from $\alpha_t \leq \sqrt{\frac{K_y}{32(\mu_g + L_g)L_f^2 L_z^2 c_\beta N^2}}$ and $c_\gamma \geq \frac{16(\mu_g + L_g)L_z^2 c_\beta N K_z}{K_y L_{\mu_q}}$.

To ensure $\bar{C}_h \leq 0$, we have

$$\bar{C}_h = \alpha_t^2 L_l - \frac{\alpha_t}{2} + \frac{2}{c_\beta L_{\mu_g}}L_y^2 \alpha_t K_y + \frac{4}{c_\gamma L_{\mu_q}}L_z^2 \alpha_t N K_z$$

$$\overset{(a)}{\leq} \frac{\alpha_t}{6} - \frac{\alpha_t}{2} + \frac{\alpha_t}{6} + \frac{\alpha_t}{6} = 0,$$

where (a) is due to $\alpha_t \leq \frac{1}{6L_l}$, $c_\beta \geq \frac{12L_y^2 K_y}{L_{\mu_g}}$ and $c_\gamma \geq \frac{24L_z^2 N K_z}{L_{\mu_q}}$.

As a summary, to ensure the descent of the potential function, we choose

$$\alpha_t \leq \min\left\{\frac{1}{6L_l}, \frac{1}{c_\beta(\mu_g + L_g)}, \frac{1}{c_\gamma(\mu_g + L_q)}, \frac{1}{2L_{\mu_g}c_\beta}, \frac{1}{2L_{\mu_q}c_\gamma}, \frac{L_l}{16L_f^2 L_z^2 N^2},\right.$$

$$\left.\sqrt{\frac{K_y}{32(\mu_g + L_g)L_f^2 L_z^2 N^2 c_\beta}}\right\},$$

$$c_\beta = \frac{12L_y^2 K_y}{L_{\mu_g}}, \quad c_\gamma = \max\left\{\frac{24L_z^2 N K_z}{L_{\mu_q}}, \frac{192(\mu_g + L_g)L_z^2 N K_z L_y^2}{L_{\mu_g}L_{\mu_q}}\right\}, \quad K_y = \frac{L_f}{\sqrt{3}L_y},$$

$$K_z = \max\left\{\frac{24\sigma_{g_{xy}}^2 N}{c_\gamma L_{\mu_q}}, \frac{24L_f^2 N}{c_\gamma L_{\mu_q}}\right\}, \quad |\mathcal{D}^{g_{yy}}| \geq \frac{48\tilde{\sigma}_{g_{yy}}^2}{L_{\mu_q}^2}.$$

Then, we get

$$\mathbb{E}\left[W_{t+1} - W_t\right] \leq -\frac{\alpha_t}{2} \sum_{n=0}^{N-1} \mathbb{E}\left[\|\nabla l\left(\mathbf{x}_t^n\right)\|^2\right] + \frac{L_l^2 c_\beta^2}{16 L_f^2 L_z^2 N} \sigma_{g_y}^2 \alpha_t + 4\sigma_{g_{xy}}^2 \frac{B_{f_y}^2}{\mu_g^2} N\alpha_t + 2\sigma_{f_x}^2 N\alpha_t$$

$$+ K_y \frac{2c_\beta}{L_{\mu_g}} \alpha_t \sigma_{g_y}^2 N + K_z \left(\frac{8B_{f_y}^2}{\mu_g^2 L_{\mu_q}} \sigma_{g_{yy}}^2 c_\gamma \alpha_t + \frac{4c_\gamma}{L_{\mu_q}} \sigma_{f_y}^2 \alpha_t + \frac{8}{c_\gamma L_{\mu_q}} L_z^2 c_\beta^2 \alpha_t N^2 \sigma_{g_y}^2\right).$$

Therefore, the lemma is proved. $\qquad\square$

### C.2.5  Proof of Theorem 5.3

**Theorem C.9** (Non-Convex $\ell(\mathbf{x})$). *Under Assumptions 3.1–3.4, choose step-sizes $\alpha_t = \alpha$, $\beta_t \triangleq c_\beta \alpha$, and $\gamma_t \triangleq c_\gamma \alpha$ for all $t \in \{0, 1, \ldots, T\}$ with*

$$c_\beta = \frac{12 L_y^2 K_y}{L_{\mu_g}}, \quad c_\gamma = \max\left\{\frac{24 L_z^2 N K_z}{L_{\mu_q}}, \frac{192\left(\mu_g + L_g\right) L_z^2 N K_z L_y^2}{L_{\mu_g} L_{\mu_q}}\right\}, \tag{22}$$

*where*

$$K_y = \frac{L_f}{\sqrt{3} L_y}, \quad K_z = \max\left\{\frac{24 \sigma_{g_{xy}}^2 N}{c_\gamma L_{\mu_q}}, \frac{24 L_f^2 N}{c_\gamma L_{\mu_q}}\right\}, \quad L_{\mu_g} = \frac{\mu_g L_g}{\mu_g + L_g}, \quad L_{\mu_q} = \frac{\mu_g L_q}{\mu_g + L_q}. \tag{23}$$

*Moreover, choose $\alpha$ such that*

$$\alpha \leq \min\Big\{\frac{1}{6 L_l}, \frac{1}{c_\beta\left(\mu_g + L_g\right)}, \frac{1}{c_\gamma\left(\mu_g + L_q\right)}, \frac{1}{2 L_{\mu_g} c_\beta}, \frac{1}{2 L_{\mu_q} c_\gamma}, \frac{L_l}{16 L_f^2 L_z^2 N^2},$$

$$\sqrt{\frac{K_y}{32\left(\mu_g + L_g\right) L_f^2 L_z^2 N^2 c_\beta}}\Big\}.$$

*Then, the iterates generated by* LazyBLO *satisfy:*

$$\frac{1}{TN} \sum_{t=0}^{T-1} \sum_{n=0}^{N-1} \mathbb{E}\left[\|\nabla\ell\left(\mathbf{x}_t^n\right)\|^2\right] = \mathcal{O}\left(\frac{\Delta_0}{NT\alpha}\right) + \mathcal{O}\left(\sigma_{g_y}^2 + \sigma_{g_{xy}}^2 + \sigma_{f_x}^2 + \sigma_{g_{yy}}^2 + \sigma_{f_y}^2\right),$$

*where $\Delta_0 = \left(\ell(\mathbf{x}_0^0) - \ell^*\right) + \|\mathbf{y}_0^0 - \mathbf{y}^*(\mathbf{x}_0^0)\|^2 + \|\mathbf{z}_0 - \mathbf{z}^*(\mathbf{x}_0^0, \mathbf{y}_0^0)\|^2$.*

*Proof.* Choose $\alpha_t$ as a constant stepsize $\alpha_t = \alpha$. Summing the result in Lemma C.4 from $t = 0$ to $T - 1$, and then dividing by $NT$ on both sides, we get

$$\frac{\mathbb{E}\left[W_T - W_0\right]}{NT} \leq -\frac{\alpha}{2TN} \sum_{t=0}^{T-1} \sum_{n=0}^{N-1} \mathbb{E}\left[\|\nabla\ell\left(\mathbf{x}_t^n\right)\|^2\right] + \frac{\alpha}{N}\left(\sigma_{f_x}^2 C_{f_x} N + \sigma_{f_y}^2 C_{f_y} N + \sigma_{g_{yy}}^2 C_{g_{yy}} N\right.$$

$$\left. + \sigma_{g_y}^2 \left(C_{g_1} N + C_{g_2} \frac{1}{N}\right) + \sigma_{g_{xy}}^2 C_{g_{xy}} N\right).$$

Rearranging the terms and multiplying by $2/\alpha$ on both sides, we have

$$\frac{1}{TN} \sum_{t=0}^{T-1} \sum_{n=0}^{N-1} \mathbb{E}\left[\|\nabla\ell\left(\mathbf{x}_t^n\right)\|^2\right]$$

$$\leq \frac{2\mathbb{E}\left[W_0 - \ell^*\right]}{\alpha NT} + 2\left(\sigma_{f_x}^2 C_{f_x} + \sigma_{f_y}^2 C_{f_y} + \sigma_{g_y}^2 \left(C_{g_1} + C_{g_2} \frac{1}{N^2}\right) + \sigma_{g_{yy}}^2 C_{g_{yy}} + \sigma_{g_{xy}}^2 C_{g_{xy}}\right)$$

$$\leq \frac{2\left(W_0 - \ell^*\right)}{\alpha NT} + 2\left(\sigma_{f_x}^2 C_{f_x} + \sigma_{f_y}^2 C_{f_y} + \sigma_{g_y}^2 \left(C_{g_1} + C_{g_2} \frac{1}{N^2}\right) + \sigma_{g_{yy}}^2 C_{g_{yy}} + \sigma_{g_{xy}}^2 C_{g_{xy}}\right),$$

where $W_0 = \ell\left(\mathbf{x}_0^0\right) + K_y \left\|\mathbf{y}_0^0 - \mathbf{y}^*\left(x_0^0\right)\right\|^2 + K_z \left\|\mathbf{z}_0 - \mathbf{z}^*\left(\mathbf{x}_0^0, \mathbf{y}_0^0\right)\right\|^2$.

Therefore,

$$\frac{1}{TN} \sum_{t=0}^{T-1} \sum_{n=0}^{N-1} \mathbb{E}\left[\|\nabla\ell\left(\mathbf{x}_t^n\right)\|^2\right]$$

$$= \mathcal{O}\left(\frac{\ell\left(\mathbf{x}_0^0\right) - \ell^*}{NT\alpha}\right) + \mathcal{O}\left(\frac{\left\|\mathbf{y}_0^0 - \mathbf{y}^*\left(\mathbf{x}_0^0\right)\right\|^2}{NT\alpha}\right) + \mathcal{O}\left(\frac{\left\|\mathbf{z}_0 - \mathbf{z}^*\left(\mathbf{x}_0^0, \mathbf{y}_0^0\right)\right\|^2}{NT\alpha}\right)$$

$$+ \mathcal{O}\left(\sigma_{g_{xy}}^2 + \sigma_{f_x}^2 + \sigma_{g_{yy}}^2 + \sigma_{f_y}^2 + \sigma_{g_y}^2\right).$$

The proof of the theorem is completed.

$\square$

## D   PROOF OF THEOREM 5.6: STRONGLY-CONVEX $\ell\left(\mathbf{x}\right)$

### D.1   DESCENT IN THE UPPER-LEVEL OBJECTIVE FUNCTION

**Lemma D.1.** *Under Assumptions 3.1–3.4. For strongly-convex and smooth $\ell\left(\mathbf{x}\right)$, the following inequality holds for successive iterations of Algorithm 1:*

$$\mathbb{E}\left[\ell\left(\mathbf{x}_t^{n+1}\right) - \ell^*\right]$$

$$\leq \left(1 - \mu_f \alpha_t\right) \mathbb{E}\left[\ell\left(\mathbf{x}_t^n\right) - \ell^*\right] - \left(\frac{\alpha_t}{2} - \frac{\alpha_t^2 L_l}{2}\right) \mathbb{E}\left[\left\|h_{t,n}^f\right\|^2\right] + 8L_f^2 L_z^2 \alpha_t^3 N \sum_{n=0}^{N-1} \mathbb{E}\left[\left\|h_{t,n}^f\right\|^2\right]$$

$$+ 2\alpha_t L_f^2 \mathbb{E}\left[\|\mathbf{y}_t^n - \mathbf{y}^*\left(\mathbf{x}_t^n\right)\|^2\right] + \left(4\sigma_{g_{xy}}^2 \alpha_t + 4L_f^2 \alpha_t\right) \mathbb{E}\left[\|\mathbf{z}_t - \mathbf{z}_t^*\|^2\right] + 16L_f^2 L_z^2 \beta_t^2 N^2 \sigma_{g_y}^2 \alpha_t$$

$$+ 16L_f^2 L_z^2 \beta_t^2 N \alpha_t \sum_{n=0}^{N-1} \mathbb{E}\left[\|\nabla_{\mathbf{y}} g\left(\mathbf{x}_t^n, \mathbf{y}_t^n\right)\|^2\right] + 2\sigma_{f_x}^2 \alpha_t + 4\sigma_{g_{xy}}^2 \frac{B_{f_y}^2}{\mu_g^2} \alpha_t,$$

*for all $t \in \{0, 1, \ldots, T-1\}$ and $n \in \{0, 1, \ldots, N-1\}$, where the expectation is taken over the stochasticity of the algorithm.*

*Proof.* From Lemma C.1, we have

$$\mathbb{E}\left[\ell\left(\mathbf{x}_t^{n+1}\right) - \ell\left(\mathbf{x}_t^n\right)\right]$$

$$\leq -\frac{\alpha_t}{2} \mathbb{E}\left[\|\nabla\ell\left(\mathbf{x}_t^n\right)\|^2\right] - \left(\frac{\alpha_t}{2} - \frac{\alpha_t^2 L_l}{2}\right) \mathbb{E}\left[\left\|h_{t,n}^f\right\|^2\right] + 8L_f^2 L_z^2 \alpha_t^3 N \sum_{n=0}^{N-1} \mathbb{E}\left[\left\|h_{t,n}^f\right\|^2\right]$$

$$+ 2\alpha_t L_f^2 \mathbb{E}\left[\|\mathbf{y}_t^n - \mathbf{y}^*\left(\mathbf{x}_t^n\right)\|^2\right] + \left(4\sigma_{g_{xy}}^2 \alpha_t + 4L_f^2 \alpha_t\right) \mathbb{E}\left[\|\mathbf{z}_t - \mathbf{z}_t^*\|^2\right] + 2\sigma_{f_x}^2 \alpha_t$$

$$+ 16L_f^2 L_z^2 \beta_t^2 N \alpha_t \sum_{n=0}^{N-1} \mathbb{E}\left[\|\nabla_{\mathbf{y}} g\left(\mathbf{x}_t^n, \mathbf{y}_t^n\right)\|^2\right] + 16L_f^2 L_z^2 \beta_t^2 N^2 \sigma_{g_y}^2 \alpha_t + 4\sigma_{g_{xy}}^2 \frac{B_{f_y}^2}{\mu_g^2} \alpha_t. \quad (24)$$

For a strongly convex function $\ell\left(\mathbf{x}\right)$, we have the fact that for all $\mathbf{x} \in \mathbb{R}^u$,

$$\|\nabla\ell(\mathbf{x})\|^2 \geq 2\mu_f\left(\ell(\mathbf{x}) - \ell^*\right). \quad (25)$$

Substitute (25) in (24) and subtract $\ell^*$ from both sides. After rearranging the terms, the lemma is proved. $\square$

### D.2   DESCENT IN THE ERROR OF $\mathbf{y}^*\left(\mathbf{x}\right)$

**Lemma D.2.** *Under Assumptions 3.2–3.4, the approximation error of $\mathbf{y}^*\left(\mathbf{x}\right)$ of Algorithm 1 satisfies the following inequality:*

$$\mathbb{E}\left[\left\|\mathbf{y}_t^{n+1} - \mathbf{y}^*\left(\mathbf{x}_t^{n+1}\right)\right\|^2\right]$$

$$\leq (1 + c_1) (1 - 2\beta_t \mu_g) \mathbb{E}\left[ \left\| \mathbf{y}_t^n - \mathbf{y}^* (\mathbf{x}_t^n) \right\|^2 \right] + 2\beta_t^2 (1 + c_1) \mathbb{E}\left[ \left\| \nabla_{\mathbf{y}} g (\mathbf{x}_t^n, \mathbf{y}_t^n) \right\|^2 \right]$$

$$+ \left( 1 + \frac{1}{c_1} \right) L_y^2 \alpha_t^2 \mathbb{E}\left[ \left\| h_{t,n}^f \right\|^2 \right] + 2 (1 + c_1) \beta_t^2 \sigma_{g_y}^2,$$

*for all $t \in \{0, 1, \ldots, T - 1\}$ and $n \in \{0, 1, \ldots, N - 1\}$ with a constant $c_1 > 0$, where the expectation is taken over the stochasticity of the algorithm.*

*Proof.*

$$\mathbb{E}\left[ \left\| \mathbf{y}_t^{n+1} - \mathbf{y}^* (\mathbf{x}_t^{n+1}) \right\|^2 \right]$$

$$\overset{(a)}{\leq} \mathbb{E}\left[ (1 + c_1) \left\| \mathbf{y}_t^{n+1} - \mathbf{y}^* (\mathbf{x}_t^n) \right\|^2 + \left( 1 + \frac{1}{c_1} \right) \left\| \mathbf{y}^* (\mathbf{x}_t^n) - \mathbf{y}^* (\mathbf{x}_t^{n+1}) \right\|^2 \right]$$

$$\overset{(b)}{\leq} \mathbb{E}\left[ (1 + c_1) \left\| \mathbf{y}_t^n - \beta_t h_{t,n}^g - \mathbf{y}^* (\mathbf{x}_t^n) \right\|^2 + \left( 1 + \frac{1}{c_1} \right) L_y^2 \left\| \mathbf{x}_t^{n+1} - \mathbf{x}_t^n \right\|^2 \right]$$

$$\overset{(c)}{=} \mathbb{E}\left[ (1 + c_1) \left\| \mathbf{y}_t^n - \beta_t h_{t,n}^g - \mathbf{y}^* (\mathbf{x}_t^n) \right\|^2 + \left( 1 + \frac{1}{c_1} \right) L_y^2 \alpha_t^2 \left\| h_{t,n}^f \right\|^2 \right], \tag{26}$$

where (a) results from Young's inequality. (b) is because of the update rule of Algorithm 1 and the Lipschitzness of $\mathbf{y}^* (\cdot)$ (see Lemma 5.1). (c) follows from the update rule of Algorithm 1.

Next, we bound the first term of the above inequality.

$$\mathbb{E}\left[ \left\| \mathbf{y}_t^n - \beta_t h_{t,n}^g - \mathbf{y}^* (x_t^n) \right\|^2 \right]$$

$$= \mathbb{E}\left[ \left\| \mathbf{y}_t^n - \mathbf{y}^* (\mathbf{x}_t^n) \right\|^2 \right] + \beta_t^2 \mathbb{E}\left[ \left\| h_{t,n}^g \right\|^2 \right] - 2\beta_t \mathbb{E}\left[ \langle h_{t,n}^g, \mathbf{y}_t^n - \mathbf{y}^* (\mathbf{x}_t^n) \rangle \right]$$

$$\leq \mathbb{E}\left[ \left\| \mathbf{y}_t^n - \mathbf{y}^* (\mathbf{x}_t^n) \right\|^2 \right] + 2\beta_t^2 \mathbb{E}\left[ \left\| h_{t,n}^g - \nabla_{\mathbf{y}} g (\mathbf{x}_t^n, \mathbf{y}_t^n) \right\|^2 \right] + 2\beta_t^2 \mathbb{E}\left[ \left\| \nabla_{\mathbf{y}} g (\mathbf{x}_t^n, \mathbf{y}_t^n) \right\|^2 \right]$$

$$- 2\beta_t \mathbb{E}\left[ \langle h_{t,n}^g, \mathbf{y}_t^n - \mathbf{y}^* (\mathbf{x}_t^n) \rangle \right]$$

$$\overset{(a)}{\leq} \mathbb{E}\left[ \left\| \mathbf{y}_t^n - \mathbf{y}^* (\mathbf{x}_t^n) \right\|^2 \right] + 2\beta_t^2 \mathbb{E}\left[ \left\| h_{t,n}^g - \nabla_{\mathbf{y}} g (\mathbf{x}_t^n, \mathbf{y}_t^n) \right\|^2 \right] + 2\beta_t^2 \mathbb{E}\left[ \left\| \nabla_{\mathbf{y}} g (\mathbf{x}_t^n, \mathbf{y}_t^n) \right\|^2 \right]$$

$$- 2\beta_t \mathbb{E}\left[ \langle \nabla_{\mathbf{y}} g (\mathbf{x}_t^n, \mathbf{y}_t^n), \mathbf{y}_t^n - \mathbf{y}^* (\mathbf{x}_t^n) \rangle \right]$$

$$\overset{(b)}{\leq} (1 - 2\beta_t \mu_g) \mathbb{E}\left[ \left\| \mathbf{y}_t^n - \mathbf{y}^* (\mathbf{x}_t^n) \right\|^2 \right] + 2\beta_t^2 \mathbb{E}\left[ \left\| h_{t,n}^g - \nabla_{\mathbf{y}} g (\mathbf{x}_t^n, \mathbf{y}_t^n) \right\|^2 \right] + 2\beta_t^2 \mathbb{E}\left[ \left\| \nabla_{\mathbf{y}} g (\mathbf{x}_t^n, \mathbf{y}_t^n) \right\|^2 \right]$$

$$\overset{(c)}{\leq} (1 - 2\beta_t \mu_g) \mathbb{E}\left[ \left\| \mathbf{y}_t^n - \mathbf{y}^* (\mathbf{x}_t^n) \right\|^2 \right] + 2\beta_t^2 \mathbb{E}\left[ \left\| \nabla_{\mathbf{y}} g (\mathbf{x}_t^n, \mathbf{y}_t^n) \right\|^2 \right] + 2\beta_t^2 \sigma_{g_y}^2, \tag{27}$$

where (a) uses the fact that $\mathbb{E}\left[ h_{t,n}^g | \mathcal{F}_t^n \right] = \nabla_{\mathbf{y}} g (\mathbf{x}_t^n, \mathbf{y}_t^n)$, and $\mathcal{F}_t^n \triangleq \sigma \{\mathbf{y}_0^0, \mathbf{x}_0^0, \cdots, \mathbf{y}_t^n, \mathbf{x}_t^n\}$ is defined as the sigma algebra generated by the iteration sequence of Algorithm 1. (b) utilizes the fact that for $\mu_g$-strongly convex $g (\mathbf{x}, \mathbf{y})$, we have $\langle \nabla_{\mathbf{y}} g (\mathbf{x}, \mathbf{y}_1) - \nabla_{\mathbf{y}} g (\mathbf{x}, \mathbf{y}_2), \mathbf{y}_1 - \mathbf{y}_2 \rangle \geq \mu_g \|\mathbf{y}_1 - \mathbf{y}_2\|^2$. (c) is because of the bounded variance in Assumption 3.4.

Substituting (27) in (26) yields the lemma. $\qquad \square$

### D.3 DESCENT IN THE ERROR OF $\mathbf{z}^* (\mathbf{x}, \mathbf{y})$

**Lemma D.3.** *Under Assumptions 3.1–3.4, the following inequality of the approximation error of $\mathbf{z}^* (\mathbf{x}, \mathbf{y})$ holds for Algorithm 1:*

$$\mathbb{E}\left[ \left\| \mathbf{z}_{t+1} - \mathbf{z}_{t+1}^* \right\|^2 \right]$$

$$\leq (1 + c_3) \left( 1 - 2\gamma_t \mu_g + 8\sigma_{g_{yy}}^2 \gamma_t^2 \right) \mathbb{E}\left[ \left\| \mathbf{z}_t - \mathbf{z}_t^* \right\|^2 \right] + \left( 2 + \frac{2}{c_3} \right) L_z^2 \alpha_t^2 N \sum_{n=0}^{N-1} \mathbb{E}\left[ \left\| h_{t,n}^f \right\|^2 \right]$$

$$+ 2\gamma_t^2 (1 + c_3) \mathbb{E}\left[ \left\| \nabla_{\mathbf{z}} q (\mathbf{x}_t^N, \mathbf{y}_t^N, \mathbf{z}_t) \right\|^2 \right] + 4 \left( 1 + \frac{1}{c_3} \right) L_z^2 \beta_t^2 N \sum_{n=0}^{N-1} \mathbb{E}\left[ \left\| \nabla_{\mathbf{y}} g (\mathbf{x}_t^n, \mathbf{y}_t^n) \right\|^2 \right]$$

$$+ 4 \left( 1 + \frac{1}{c_3} \right) L_z^2 \beta_t^2 N^2 \sigma_{g_y}^2 + 4 \sigma_{f_y}^2 (1 + c_3) \gamma_t^2 + 8 \sigma_{g_{yy}}^2 \frac{B_{f_y}^2}{\mu_g^2} (1 + c_3) \gamma_t^2,$$

*for all $t \in \{0, 1, \ldots, T-1\}$ and $n \in \{0, 1, \ldots, N-1\}$ with some constants $c_3, c_4 > 0$, where the expectation is taken over the stochasticity of the algorithm.*

*Proof.* With the results from the proof of Lemma C.7, we have

$$\mathbb{E} \left[ \left\| \mathbf{z}_{t+1} - \mathbf{z}_{t+1}^* \right\|^2 \right] \le (1 + c_3) \mathbb{E} \left[ \left\| \mathbf{z}_{t+1} - \mathbf{z}_t^* \right\|^2 \right] + 2 \left( 1 + \frac{1}{c_3} \right) L_z^2 \alpha_t^2 N \sum_{n=0}^{N-1} \mathbb{E} \left[ \left\| h_{t,n}^f \right\|^2 \right]$$

$$+ 4 \left( 1 + \frac{1}{c_3} \right) L_z^2 \beta_t^2 N \sum_{n=0}^{N-1} \mathbb{E} \left[ \left\| \nabla_{\mathbf{y}} g \left( \mathbf{x}_t^n, \mathbf{y}_t^n \right) \right\|^2 \right] + 4 \left( 1 + \frac{1}{c_3} \right) L_z^2 \beta_t^2 N^2 \sigma_{g_y}^2, \tag{28}$$

Then, we consider the first term on the right:

$$\mathbb{E} \left[ \left\| \mathbf{z}_{t+1} - \mathbf{z}_t^* \right\|^2 \right] \overset{(a)}{=} \mathbb{E} \left[ \left\| \mathbf{z}_t - \mathbf{z}_t^* \right\|^2 \right] + \gamma_t^2 \mathbb{E} \left[ \left\| h_t^q \right\|^2 \right] - 2 \gamma_t \mathbb{E} \left[ \langle h_t^q, \mathbf{z}_t - \mathbf{z}_t^* \rangle \right]$$

$$\le \mathbb{E} \left[ \left\| \mathbf{z}_t - \mathbf{z}_t^* \right\|^2 \right] + 2 \gamma_t^2 \mathbb{E} \left[ \left\| h_t^q - \nabla_{\mathbf{z}} q \left( \mathbf{x}_t^N, \mathbf{y}_t^N, \mathbf{z}_t \right) \right\|^2 \right] + 2 \gamma_t^2 \mathbb{E} \left[ \left\| \nabla_{\mathbf{z}} q \left( \mathbf{x}_t^N, \mathbf{y}_t^N, \mathbf{z}_t \right) \right\|^2 \right]$$

$$- 2 \gamma_t \mathbb{E} \left[ \langle h_t^q, \mathbf{z}_t - \mathbf{z}_t^* \rangle \right]$$

$$\overset{(b)}{\le} \mathbb{E} \left[ \left\| \mathbf{z}_t - \mathbf{z}_t^* \right\|^2 \right] + 2 \gamma_t^2 \mathbb{E} \left[ \left\| h_t^q - \nabla_{\mathbf{z}} q \left( \mathbf{x}_t^N, \mathbf{y}_t^N, \mathbf{z}_t \right) \right\|^2 \right] + 2 \gamma_t^2 \mathbb{E} \left[ \left\| \nabla_{\mathbf{z}} q \left( \mathbf{x}_t^N, \mathbf{y}_t^N, \mathbf{z}_t \right) \right\|^2 \right]$$

$$- 2 \gamma_t \mathbb{E} \left[ \langle \nabla_{\mathbf{z}} q \left( \mathbf{x}_t^N, \mathbf{y}_t^N, \mathbf{z}_t \right), \mathbf{z}_t - \mathbf{z}_t^* \rangle \right]$$

$$\overset{(c)}{\le} (1 - 2 \gamma_t \mu_g) \mathbb{E} \left[ \left\| \mathbf{z}_t - \mathbf{z}_t^* \right\|^2 \right] + 2 \gamma_t^2 \mathbb{E} \left[ \left\| h_t^q - \nabla_{\mathbf{z}} q \left( \mathbf{x}_t^N, \mathbf{y}_t^N, \mathbf{z}_t \right) \right\|^2 \right] + 2 \gamma_t^2 \mathbb{E} \left[ \left\| \nabla_{\mathbf{z}} q \left( \mathbf{x}_t^N, \mathbf{y}_t^N, \mathbf{z}_t \right) \right\|^2 \right], \tag{29}$$

where (a) is due to the update rule of Algorithm 1. (b) follows from the fact that $\mathbb{E} [h_t^q | \mathcal{F}_t] = \nabla_{\mathbf{z}} q \left( \mathbf{x}_t^N, \mathbf{y}_t^N, \mathbf{z}_t \right)$, and (c) utilizes the fact that for $\mu_g$-strongly convex $q(\mathbf{x}, \mathbf{y}, \mathbf{z})$, we have $\langle \nabla_{\mathbf{z}} q (\mathbf{x}, \mathbf{y}, \mathbf{z}_1) - \nabla_{\mathbf{z}} q (\mathbf{x}, \mathbf{y}, \mathbf{z}_2), \mathbf{z}_1 - \mathbf{z}_2 \rangle \ge \mu_g \left\| \mathbf{z}_1 - \mathbf{z}_2 \right\|^2$.

Next, we consider the second term $\mathbb{E} \left[ \left\| h_t^q - \nabla_{\mathbf{z}} q \left( \mathbf{x}_t^N, \mathbf{y}_t^N, \mathbf{z}_t \right) \right\|^2 \right]$ of the above inequality:

$$\mathbb{E} \left[ \left\| h_t^q - \nabla_{\mathbf{z}} q \left( \mathbf{x}_t^N, \mathbf{y}_t^N, \mathbf{z}_t \right) \right\|^2 \right]$$

$$\overset{(a)}{=} \mathbb{E} \left[ \left\| \nabla_{\mathbf{yy}}^2 g \left( \mathbf{x}_t^N, \mathbf{y}_t^N \right) \mathbf{z}_t + \nabla_{\mathbf{y}} f \left( \mathbf{x}_t^N, \mathbf{y}_t^N \right) - \left( \nabla_{\mathbf{yy}}^2 g \left( \mathbf{x}_t^N, \mathbf{y}_t^N; \mathcal{D}_t^{g_{yy}} \right) \mathbf{z}_t + \nabla_{\mathbf{y}} f \left( \mathbf{x}_t^N, \mathbf{y}_t^N; \mathcal{D}_t^{f_y} \right) \right) \right\|^2 \right]$$

$$\le 2 \mathbb{E} \left[ \left\| \mathbf{z}_t \right\|^2 \left\| \nabla_{\mathbf{yy}}^2 g \left( \mathbf{x}_t^N, \mathbf{y}_t^N \right) - \nabla_{\mathbf{yy}}^2 g \left( \mathbf{x}_t^N, \mathbf{y}_t^N; \mathcal{D}_t^{g_{yy}} \right) \right\|^2 + \left\| \nabla_{\mathbf{y}} f \left( \mathbf{x}_t^N, \mathbf{y}_t^N \right) - \nabla_{\mathbf{y}} f \left( \mathbf{x}_t^N, \mathbf{y}_t^N; \mathcal{D}_t^{f_y} \right) \right\|^2 \right]$$

$$\overset{(b)}{\le} \mathbb{E} \left[ 2 \sigma_{g_{yy}}^2 \left\| \mathbf{z}_t - \mathbf{z}_t^* + \mathbf{z}_t^* \right\|^2 \right] + 2 \sigma_{f_y}^2$$

$$\le \mathbb{E} \left[ 4 \sigma_{g_{yy}}^2 \left\| \mathbf{z}_t - \mathbf{z}_t^* \right\|^2 + 4 \sigma_{g_{yy}}^2 \left\| \mathbf{z}_t^* \right\|^2 \right] + 2 \sigma_{f_y}^2$$

$$\overset{(c)}{\le} 4 \sigma_{g_{yy}}^2 \mathbb{E} \left[ \left\| \mathbf{z}_t - \mathbf{z}_t^* \right\|^2 \right] + 4 \sigma_{g_{yy}}^2 \frac{B_{f_y}^2}{\mu_g^2} + 2 \sigma_{f_y}^2, \tag{30}$$

where (a) results from the definitions of $h_t^q$ and $\nabla_{\mathbf{z}} q \left( \mathbf{x}_t^N, \mathbf{y}_t^N, \mathbf{z}_t \right)$. (b) uses the bounded variance in Assumption 3.4. (c) is because of the bound of $\mathbf{z}^*(\mathbf{x}, \mathbf{y})$ in Lemma 5.2.

Substituting (30) into (29) and then substituting the obtained inequality into (28) proves the lemma. $\square$

## D.4 DESCENT IN THE POTENTIAL FUNCTION

We define a different potential function $\hat{W}_t$ as follows:

$$\hat{W}_t = \sum_{n=0}^{N-1} \left( \ell\left(\mathbf{x}_t^n\right) - \ell^* \right) + \sum_{n=0}^{N-1} \left\| \mathbf{y}_t^n - \mathbf{y}^*\left(\mathbf{x}_t^n\right) \right\|^2 + \left\| \mathbf{z}_t - \mathbf{z}_t^* \right\|^2.$$

**Lemma D.4.** *Choose $c_1 = \frac{\beta_t \mu_g}{2(1 - \beta_t \mu_g)}$, and $c_3 = \frac{\gamma_t \mu_g}{2(1 - \gamma_t \mu_g)}$. Under the same conditions as described in Theorem D.5 and utilizing Lemmas B.D.1-B.D.3, the iterates generated by Algorithm 1 satisfies:*

$$\mathbb{E}\left[\hat{W}_{t+1}\right] \leq (1 - \mu_f \alpha_t)\,\mathbb{E}\left[\hat{W}_t\right] + 16L_f^2 L_z^2 \hat{c}_\beta^2 \alpha_t^3 N^3 \sigma_{g_y}^2 + 4\sigma_{g_{xy}}^2 \frac{B_{f_y}^2}{\mu_g^2} \alpha_t N + 2\sigma_{f_x}^2 \alpha_t N + 4\hat{c}_\beta^2 \sigma_{g_y}^2 N \alpha_t^2$$

$$+ 8\sigma_{f_y}^2 \hat{c}_\gamma^2 \alpha_t^2 + 16\sigma_{g_{yy}}^2 \frac{B_{f_y}^2}{\mu_g^2} \hat{c}_\gamma^2 \alpha_t^2 + \frac{8}{\mu_g \hat{c}_\gamma} L_z^2 \hat{c}_\beta^2 N^2 \sigma_{g_y}^2 \alpha_t,$$

*for all $t \in \{0, 1, \ldots, T - 1\}$.*

*Proof.* With the results from Lemma D.1, we have

$$\sum_{n=0}^{N-1} \mathbb{E}\left[\ell\left(\mathbf{x}_t^{n+1}\right) - \ell^*\right]$$

$$\leq \mathbb{E}\left[ (1 - \mu_f \alpha_t) \sum_{n=0}^{N-1} \left(\ell\left(\mathbf{x}_t^n\right) - \ell^*\right) + \left( \frac{\alpha_t^2 L_l}{2} - \frac{\alpha_t}{2} + 8L_f^2 L_z^2 \alpha_t^3 N^2 \right) \sum_{n=0}^{N-1} \left\| h_{t,n}^f \right\|^2 \right.$$

$$+ \left( 4\sigma_{g_{xy}}^2 \alpha_t N + 4L_f^2 \alpha_t N \right) \left\| \mathbf{z}_t - \mathbf{z}_t^* \right\|^2 + 16L_f^2 L_z^2 \beta_t^2 \alpha_t N^2 \sum_{n=0}^{N-1} \left\| \nabla_{\mathbf{y}} g\left(\mathbf{x}_t^n, \mathbf{y}_t^n\right) \right\|^2$$

$$\left. + 2L_f^2 \alpha_t \sum_{n=0}^{N-1} \left\| \mathbf{y}_t^n - \mathbf{y}^*\left(\mathbf{x}_t^n\right) \right\|^2 \right] + 16L_f^2 L_z^2 \beta_t^2 \alpha_t N^3 \sigma_{g_y}^2 + 2\sigma_{f_x}^2 \alpha_t N + 4\sigma_{g_{xy}}^2 \frac{B_{f_y}^2}{\mu_g^2} \alpha_t N$$

$$\overset{(a)}{\leq} \mathbb{E}\left[ (1 - \mu_f \alpha_t) \sum_{n=0}^{N-1} \left(\ell\left(\mathbf{x}_t^n\right) - \ell^*\right) + \left( \frac{\alpha_t^2 L_l}{2} - \frac{\alpha_t}{2} + 8L_f^2 L_z^2 \alpha_t^3 N^2 \right) \sum_{n=0}^{N-1} \left\| h_{t,n}^f \right\|^2 \right.$$

$$\left. + \left( 4\sigma_{g_{xy}}^2 \alpha_t N + 4L_f^2 \alpha_t N \right) \left\| \mathbf{z}_t - \mathbf{z}_t^* \right\|^2 + \left( 2L_f^2 \alpha_t + 16L_f^2 L_z^2 \beta_t^2 \alpha_t N^2 L_g^2 \right) \sum_{n=0}^{N-1} \left\| \mathbf{y}_t^n - \mathbf{y}^*\left(\mathbf{x}_t^n\right) \right\|^2 \right]$$

$$+ 16L_f^2 L_z^2 \beta_t^2 \alpha_t N^3 \sigma_{g_y}^2 + 4\sigma_{g_{xy}}^2 \frac{B_{f_y}^2}{\mu_g^2} \alpha_t N + 2\sigma_{f_x}^2 \alpha_t N$$

$$\overset{(b)}{\leq} \mathbb{E}\left[ (1 - \mu_f \alpha_t) \sum_{n=0}^{N-1} \left(\ell\left(\mathbf{x}_t^n\right) - \ell^*\right) + \left( \frac{\alpha_t^2 L_l}{2} - \frac{\alpha_t}{2} + 8L_f^2 L_z^2 \alpha_t^3 N^2 \right) \sum_{n=0}^{N-1} \left\| h_{t,n}^f \right\|^2 \right.$$

$$\left. + \left( 4\sigma_{g_{xy}}^2 \alpha_t N + 4L_f^2 \alpha_t N \right) \left\| \mathbf{z}_t - \mathbf{z}_t^* \right\|^2 + \left( 2L_f^2 \alpha_t + 16L_f^2 L_z^2 \hat{c}_\beta^2 \alpha_t^3 N^2 L_g^2 \right) \sum_{n=0}^{N-1} \left\| \mathbf{y}_t^n - \mathbf{y}^*\left(\mathbf{x}_t^n\right) \right\|^2 \right]$$

$$+ 16L_f^2 L_z^2 \hat{c}_\beta^2 \alpha_t^3 N^3 \sigma_{g_y}^2 + 4\sigma_{g_{xy}}^2 \frac{B_{f_y}^2}{\mu_g^2} \alpha_t N + 2\sigma_{f_x}^2 \alpha_t N, \tag{31}$$

where (a) uses the fact that $\nabla_{\mathbf{y}} g\left(\mathbf{x}, \mathbf{y}^*\left(\mathbf{x}\right)\right) = 0$ and utilizes the the Lipschitzness of $\nabla_{\mathbf{y}} g\left(\mathbf{x}, \mathbf{y}\right)$ (see Assumption 3.2). (b) follows from the definition of $\beta_t = \hat{c}_\beta \alpha_t$.

From Lemma D.2, we have

$$\sum_{n=0}^{N-1} \mathbb{E}\left[ \left\| \mathbf{y}_t^{n+1} - \mathbf{y}^*\left(\mathbf{x}_t^{n+1}\right) \right\|^2 \right]$$

$$\overset{(a)}{\leq} \mathbb{E}\left[(1+c_1)\left(1-2\beta_t\mu_g+2\beta_t^2L_g^2\right)\sum_{n=0}^{N-1}\|\mathbf{y}_t^n-\mathbf{y}^*\left(\mathbf{x}_t^n\right)\|^2 + \left(1+\frac{1}{c_1}\right)L_y^2\alpha_t^2\sum_{n=0}^{N-1}\left\|h_{t,n}^f\right\|^2\right]$$

$$+\, 2\left(1+c_1\right)\beta_t^2\sigma_{g_y}^2 N,$$

where (a) is because of the fact that $\nabla_\mathbf{y} g\left(\mathbf{x},\mathbf{y}^*\left(\mathbf{x}\right)\right)=0$ and follows from the the Lipschitzness of $\nabla_\mathbf{y} g\left(\mathbf{x},\mathbf{y}\right)$ (see Assumption 3.2).

From the choice of $c_1 = \frac{\beta_t\mu_g}{2(1-\beta_t\mu_g)}$, we have $1+\frac{1}{c_1} \leq \frac{2}{\mu_g\beta_t}$. Choosing $\beta_t \leq \frac{\mu_g}{2L_g^2}$ and using the definition of $\beta_t = \hat{c}_\beta\alpha_t$, we get

$$\sum_{n=0}^{N-1}\mathbb{E}\left[\left\|\mathbf{y}_t^{n+1}-\mathbf{y}^*\left(x_t^{n+1}\right)\right\|^2\right]$$

$$\leq \mathbb{E}\left[\left(1-\frac{\mu_g\hat{c}_\beta\alpha_t}{2}\right)\sum_{n=0}^{N-1}\|\mathbf{y}_t^n-\mathbf{y}^*\left(\mathbf{x}_t^n\right)\|^2 + \frac{2}{\mu_g\hat{c}_\beta}L_y^2\alpha_t\sum_{n=0}^{N-1}\left\|h_{t,n}^f\right\|^2\right] + 4\hat{c}_\beta^2\sigma_{g_y}^2 N\alpha_t^2. \quad (32)$$

Following from Lemma D.3, we have

$$\mathbb{E}\left[\left\|\mathbf{z}_{t+1}-\mathbf{z}_{t+1}^*\right\|^2\right]$$

$$\overset{(a)}{\leq}(1+c_3)\left(1-2\gamma_t\mu_g+8\sigma_{g_{yy}}^2\gamma_t^2+2\gamma_t^2L_q^2\right)\mathbb{E}\left[\left\|\mathbf{z}_t-\mathbf{z}_t^*\right\|^2\right] + 2\left(1+\frac{1}{c_3}\right)L_z^2\alpha_t^2 N\sum_{n=0}^{N-1}\mathbb{E}\left[\left\|h_{t,n}^f\right\|^2\right]$$

$$+\, 4\left(1+\frac{1}{c_3}\right)L_z^2\beta_t^2 NL_g^2\sum_{n=0}^{N-1}\mathbb{E}\left[\|\mathbf{y}_t^n-\mathbf{y}^*\left(\mathbf{x}_t^n\right)\|^2\right] + 4\left(1+\frac{1}{c_3}\right)L_z^2\beta_t^2 N^2\sigma_{g_y}^2 + 4\sigma_{f_y}^2\left(1+c_3\right)\gamma_t^2$$

$$+\, 8\sigma_{g_{yy}}^2\frac{B_{f_y}^2}{\mu_g^2}\left(1+c_3\right)\gamma_t^2,$$

where (a) utilizes fact that $\nabla_\mathbf{y} g\left(\mathbf{x},\mathbf{y}^*\left(\mathbf{x}\right)\right)=0$ and $\nabla_\mathbf{z} q\left(\mathbf{x},\mathbf{y},\mathbf{z}^*\right)=0$. In addition, it uses the the Lipschitzness of $\nabla_\mathbf{y} g\left(\mathbf{x},\mathbf{y}\right)$ in Assumption 3.2 and $\nabla_\mathbf{z} q\left(\mathbf{x},\mathbf{y},\mathbf{z}\right)$ proved as follows.

$$\|\nabla_\mathbf{z} q\left(\mathbf{x},\mathbf{y},\mathbf{z}_1\right)-\nabla_\mathbf{z} q\left(\mathbf{x},\mathbf{y},\mathbf{z}_2\right)\| \overset{(a)}{=} \left\|\nabla_{\mathbf{yy}}^2 g(\mathbf{x},\mathbf{y})\mathbf{z}_1+\nabla_\mathbf{y} f(\mathbf{x},\mathbf{y})-\nabla_{\mathbf{yy}}^2 g(\mathbf{x},\mathbf{y})\mathbf{z}_2-\nabla_\mathbf{y} f(\mathbf{x},\mathbf{y})\right\|$$

$$= \left\|\nabla_{\mathbf{yy}}^2 g(\mathbf{x},\mathbf{y})\right\|\|\mathbf{z}_1-\mathbf{z}_2\| \overset{(b)}{\leq} B_{g_{yy}}\|\mathbf{z}_1-\mathbf{z}_2\| \overset{(c)}{=} L_q\|\mathbf{z}_1-\mathbf{z}_2\|,$$

where (a) follows from the definition of $\nabla_\mathbf{z} q\left(\mathbf{x},\mathbf{y},\mathbf{z}\right)$. (b) assumes $\left\|\nabla_{yy}^2 g(x,y)\right\| \leq B_{g_{yy}}$, and (c) defines $L_q = B_{g_{yy}}$.

From the choice of $c_3 = \frac{\gamma_t\mu_g}{2(1-\gamma_t\mu_g)}$, we get $1+\frac{1}{c_3} \leq \frac{2}{\mu_g\gamma_t}$. Selecting $\gamma_t \leq \frac{\mu_g}{4L_q^2}$, $\gamma_t \leq \frac{\mu_g}{16\sigma_{g_{yy}}^2}$ and using the definition of $\beta_t = \hat{c}_\beta\alpha_t$, $\gamma_t = \hat{c}_\gamma\alpha_t$, we have

$$\mathbb{E}\left[\left\|\mathbf{z}_{t+1}-\mathbf{z}_{t+1}^*\right\|^2\right] \leq \mathbb{E}\left[\left(1-\frac{\mu_g\hat{c}_\gamma\alpha_t}{2}\right)\|\mathbf{z}_t-\mathbf{z}_t^*\|^2 + \frac{4}{\mu_g\hat{c}_\gamma}L_z^2\alpha_t N\sum_{n=0}^{N-1}\left\|h_{t,n}^f\right\|^2\right.$$

$$\left.+\frac{8}{\mu_g\hat{c}_\gamma}L_z^2\hat{c}_\beta^2 NL_g^2\alpha_t\sum_{n=0}^{N-1}\|\mathbf{y}_t^n-\mathbf{y}^*\left(\mathbf{x}_t^n\right)\|^2\right] + 16\sigma_{g_{yy}}^2\frac{B_{f_y}^2}{\mu_g^2}\hat{c}_\gamma^2\alpha_t^2 + 8\sigma_{f_y}^2\hat{c}_\gamma^2\alpha_t^2 + \frac{8}{\mu_g\hat{c}_\gamma}L_z^2\hat{c}_\beta^2 N^2\sigma_{g_y}^2\alpha_t.$$
$$(33)$$

Combining equations (31), (32) and (33), we get

$$\mathbb{E}\left[\hat{W}_{t+1}\right] \leq (1-\mu_f\alpha_t)\mathbb{E}\left[\hat{W}_t\right] + \hat{C}_h\mathbb{E}\left[\sum_{n=0}^{N-1}\left\|h_{t,n}^f\right\|^2\right] + \hat{C}_y\mathbb{E}\left[\sum_{n=0}^{N-1}\|\mathbf{y}_t^n-\mathbf{y}^*\left(\mathbf{x}_t^n\right)\|^2\right]$$

$$+\, \hat{C}_z\mathbb{E}\left[\|\mathbf{z}_t-\mathbf{z}_t^*\|^2\right] + 16L_f^2L_z^2\hat{c}_\beta^2\alpha_t^3 N^3\sigma_{g_y}^2 + 4\sigma_{g_{xy}}^2\frac{B_{f_y}^2}{\mu_g^2}\alpha_t N + 2\sigma_{f_x}^2\alpha_t N + 4\hat{c}_\beta^2\sigma_{g_y}^2 N\alpha_t^2$$

$$+ 16\sigma_{g_{xy}}^2 \frac{B_{f_y}^2}{\mu_g^2} \hat{c}_\gamma^2 \alpha_t^2 + 8\sigma_{f_y}^2 \hat{c}_\gamma^2 \alpha_t^2 + \frac{8}{\mu_g \hat{c}_\gamma} L_z^2 \hat{c}_\beta^2 N^2 \sigma_{g_y}^2 \alpha_t,$$

where

$$\hat{C}_h = \frac{\alpha_t^2 L_l}{2} - \frac{\alpha_t}{2} + 8L_f^2 L_z^2 \alpha_t^3 N^2 + \frac{2}{\mu_g \hat{c}_\beta} L_y^2 \alpha_t + \frac{4}{\mu_g \hat{c}_\gamma} L_z^2 \alpha_t N,$$

$$\hat{C}_y = 2L_f^2 \alpha_t + 16L_f^2 L_z^2 \hat{c}_\beta^2 \alpha_t^3 N^2 L_g^2 + \mu_f \alpha_t - \frac{\mu_g \hat{c}_\beta \alpha_t}{2} + \frac{8}{\mu_g \hat{c}_\gamma} L_z^2 \hat{c}_\beta^2 N L_g^2 \alpha_t,$$

$$\hat{C}_z = 4\sigma_{g_{xy}}^2 \alpha_t N + 4L_f^2 \alpha_t N + \mu_f \alpha_t - \frac{\mu_g \hat{c}_\gamma \alpha_t}{2}.$$

To ensure $\hat{C}_h \leq 0$, we have

$$\hat{C}_h = \frac{\alpha_t^2 L_l}{2} - \frac{\alpha_t}{2} + 8L_f^2 L_z^2 \alpha_t^3 N^2 + \frac{2}{\mu_g \hat{c}_\beta} L_y^2 \alpha_t + \frac{4}{\mu_g \hat{c}_\gamma} L_z^2 \alpha_t N$$

$$\overset{(a)}{\leq} \frac{\alpha_t}{8} - \frac{\alpha_t}{2} + \frac{\alpha_t}{8} + \frac{\alpha_t}{8} + \frac{\alpha_t}{8} = 0,$$

where (a) follows from $\alpha_t \leq \min\left\{\frac{1}{4L_l}, \frac{1}{8L_f L_z N}\right\}$, $\hat{c}_\beta \geq \frac{16L_y^2}{\mu_g}$, and $\hat{c}_\gamma \geq \frac{32L_z^2}{\mu_g}$.

To ensure $\hat{C}_y \leq 0$, we have

$$\hat{C}_y = 2L_f^2 \alpha_t + 16L_f^2 L_z^2 \hat{c}_\beta^2 \alpha_t^3 N^2 L_g^2 + \mu_f \alpha_t - \frac{\mu_g \hat{c}_\beta \alpha_t}{2} + \frac{8}{\mu_g \hat{c}_\gamma} L_z^2 \hat{c}_\beta^2 N L_g^2 \alpha_t$$

$$= \left(2L_f^2 + \mu_f\right) \alpha_t + 16L_f^2 L_z^2 \hat{c}_\beta^2 \alpha_t^3 N^2 L_g^2 - \frac{\mu_g \hat{c}_\beta \alpha_t}{2} + \frac{8}{\mu_g \hat{c}_\gamma} L_z^2 \hat{c}_\beta^2 N L_g^2 \alpha_t$$

$$\overset{(a)}{\leq} \frac{\mu_g \hat{c}_\beta \alpha_t}{6} + \frac{\mu_g \hat{c}_\beta \alpha_t}{6} - \frac{\mu_g \hat{c}_\beta \alpha_t}{2} + \frac{\mu_g \hat{c}_\beta \alpha_t}{6} = 0,$$

where (a) is because of $\alpha_t \leq \sqrt{\frac{\mu_g}{96L_f^2 L_z^2 L_g^2 N^2 \hat{c}_\beta}}$, $\hat{c}_\beta \geq \frac{12L_f^2 + 6\mu_f}{\mu_g}$, and $\hat{c}_\gamma \geq \frac{\mu_g^2}{48L_z^2 L_g^2 N \hat{c}_\beta}$.

To ensure $\hat{C}_z \leq 0$, we utilize that $\hat{c}_\gamma \geq \frac{8\sigma_{g_{xy}}^2 N + 8L_f^2 N + 2\mu_f}{\mu_g}$.

As a summary, to ensure the descent of the potential function, we select

$$\alpha_t \leq \min\left\{\frac{1}{4L_l}, \frac{1}{8L_f L_z N}, \sqrt{\frac{\mu_g}{96L_f^2 L_z^2 L_g^2 N^2 \hat{c}_\beta}}, \frac{\mu_g}{2L_g^2 \hat{c}_\beta}, \frac{2}{3\mu_g \hat{c}_\beta}, \frac{\mu_g}{16\sigma_{g_{yy}}^2 \hat{c}_\gamma}, \frac{\mu_g}{4L_q^2 \hat{c}_\gamma}, \frac{2}{3\mu_g \hat{c}_\gamma}\right\},$$

$$\hat{c}_\beta = \max\left\{\frac{16L_y^2}{\mu_g}, \frac{12L_f^2 + 6\mu_f}{\mu_g}\right\}, \quad \hat{c}_\gamma = \max\left\{\frac{32L_z^2}{\mu_g}, \frac{\mu_g^2}{48L_z^2 L_g^2 N \hat{c}_\beta}, \frac{8\sigma_{g_{xy}}^2 N + 8L_f^2 N + 2\mu_f}{\mu_g}\right\}.$$

Then, we get

$$\mathbb{E}\left[\hat{W}_{t+1}\right] \leq (1 - \mu_f \alpha_t) \mathbb{E}\left[\hat{W}_t\right] + 16L_f^2 L_z^2 \hat{c}_\beta^2 \alpha_t^3 N^3 \sigma_{g_y}^2 + 4\sigma_{g_{xy}}^2 \frac{B_{f_y}^2}{\mu_g^2} \alpha_t N + 2\sigma_{f_x}^2 \alpha_t N$$

$$+ 4\hat{c}_\beta^2 \sigma_{g_y}^2 N \alpha_t^2 + 16\sigma_{g_{xy}}^2 \frac{B_{f_y}^2}{\mu_g^2} \hat{c}_\gamma^2 \alpha_t^2 + 8\sigma_{f_y}^2 \hat{c}_\gamma^2 \alpha_t^2 + \frac{8}{\mu_g \hat{c}_\gamma} L_z^2 \hat{c}_\beta^2 N^2 \sigma_{g_y}^2 \alpha_t,$$

Therefore, the lemma is proved. $\qquad\square$

### D.5 PROOF OF THEOREM 5.6

**Theorem D.5** (Strongly Convex $\ell(\mathbf{x})$). *Suppose the upper-level function $\ell(\mathbf{x})$ is $\mu_f$-strongly-convex. Under Assumptions 3.1–3.4, choose the step-sizes $\alpha_t = \alpha$, $\beta_t \triangleq \hat{c}_\beta \alpha$ and $\gamma_t \triangleq \hat{c}_\gamma \alpha$ for all*

$t \in \{0, 1, \ldots, T-1\}$, *where*

$$\hat{c}_\beta = \max \left\{ \frac{16L_y^2}{\mu_g}, \frac{6\left(2L_f^2 + \mu_f\right)}{\mu_g} \right\}, \ \hat{c}_\gamma = \max \left\{ \frac{32L_z^2}{\mu_g}, \frac{\mu_g^2}{48L_z^2 L_g^2 N \hat{c}_\beta}, \frac{8\sigma_{g_{xy}}^2 N + 8L_f^2 N + 2\mu_f}{\mu_g} \right\}.$$

$$(34)$$

*Moreover, choose $\alpha$ such that*

$$\alpha \leq \min \left\{ \frac{1}{4L_l}, \frac{1}{8L_f L_z N}, \sqrt{\frac{\mu_g}{96L_f^2 L_z^2 L_g^2 N^2 \hat{c}_\beta}}, \frac{\mu_g}{2L_g^2 \hat{c}_\beta}, \frac{2}{3\mu_g \hat{c}_\beta}, \frac{\mu_g}{16\sigma_{g_{yy}}^2 \hat{c}_\gamma}, \frac{\mu_g}{4L_q^2 \hat{c}_\gamma}, \frac{2}{3\mu_g \hat{c}_\gamma} \right\}.$$

*Then, the iterates generated by* LazyBLO *satisfy:*

$$\sum_{n=0}^{N-1} \mathbb{E}\left[ \ell\left(\mathbf{x}_t^n\right) - \ell^* \right] \leq (1 - \mu_f \alpha)^t \, \hat{\Delta}_0 + \frac{1}{\mu_f} \left( 4\sigma_{g_{xy}}^2 \frac{B_{f_y}^2}{\mu_g^2} N + 2\sigma_{f_x}^2 N + \frac{8}{\mu_g \hat{c}_\gamma} L_z^2 \hat{c}_\beta^2 N^2 \sigma_{g_y}^2 \right)$$

$$+ \frac{\alpha}{\mu_f} \left( 16\sigma_{g_{yy}}^2 \frac{B_{f_y}^2}{\mu_g^2} \hat{c}_\gamma^2 + 8\sigma_{f_y}^2 \hat{c}_\gamma^2 + 4\hat{c}_\beta^2 \sigma_{g_y}^2 N \right) + \frac{16\alpha^2}{\mu_f} L_f^2 L_z^2 \hat{c}_\beta^2 N^3 \sigma_{g_y}^2,$$

*for any $t \geq 1$, where $\hat{\Delta}_0 = \sum_{n=0}^{N-1} \left(\ell\left(\mathbf{x}_0^n\right) - \ell^*\right) + \sum_{n=0}^{N-1} \|\mathbf{y}_0^n - \mathbf{y}^*\left(\mathbf{x}_0^n\right)\|^2 + \|\mathbf{z}_0 - \mathbf{z}_0^*\|^2$.*

*Proof.* Selecting a constant step-size $\alpha_t = \alpha$ for all $t \in \{0, 1, \cdots, T-1\}$ and from Lemma D.4, we have

$$\mathbb{E}\left[\hat{W}_{t+1}\right] \leq (1 - \mu_f \alpha) \, \mathbb{E}\left[\hat{W}_t\right] + 16L_f^2 L_z^2 \hat{c}_\beta^2 \alpha^3 N^3 \sigma_{g_y}^2 + 4\sigma_{g_{xy}}^2 \frac{B_{f_y}^2}{\mu_g^2} \alpha N + 2\sigma_{f_x}^2 \alpha N + 4\hat{c}_\beta^2 \sigma_{g_y}^2 N \alpha^2$$

$$+ 16\sigma_{g_{yy}}^2 \frac{B_{f_y}^2}{\mu_g^2} \hat{c}_\gamma^2 \alpha^2 + 8\sigma_{f_y}^2 \hat{c}_\gamma^2 \alpha^2 + \frac{8}{\mu_g \hat{c}_\gamma} L_z^2 \hat{c}_\beta^2 N^2 \sigma_{g_y}^2 \alpha.$$

Applying the above inequality recursively yields

$$\mathbb{E}\left[\hat{W}_t\right] \leq (1 - \mu_f \alpha)^t \, \mathbb{E}\left[\hat{W}_0\right] + \sum_{k=0}^{t-1} (1 - \mu_f \alpha)^k \left( +16L_f^2 L_z^2 \hat{c}_\beta^2 \alpha^3 N^3 \sigma_{g_y}^2 + 2\sigma_{f_x}^2 \alpha N + 4\hat{c}_\beta^2 \sigma_{g_y}^2 N \alpha^2 \right.$$

$$\left. +8\sigma_{f_y}^2 \hat{c}_\gamma^2 \alpha^2 + 16\sigma_{g_{yy}}^2 \frac{B_{f_y}^2}{\mu_g^2} \hat{c}_\gamma^2 \alpha^2 + \frac{8}{\mu_g \hat{c}_\gamma} L_z^2 \hat{c}_\beta^2 N^2 \sigma_{g_y}^2 \alpha + 4\sigma_{g_{xy}}^2 \frac{B_{f_y}^2}{\mu_g^2} \alpha N \right)$$

$$\overset{(a)}{\leq} (1 - \mu_f \alpha)^t \, \mathbb{E}\left[\hat{W}_0\right] + \frac{1}{\mu_f} \left( 16L_f^2 L_z^2 \hat{c}_\beta^2 \alpha^2 N^3 \sigma_{g_y}^2 + 4\sigma_{g_{xy}}^2 \frac{B_{f_y}^2}{\mu_g^2} N + 2\sigma_{f_x}^2 N + 4\hat{c}_\beta^2 \sigma_{g_y}^2 N \alpha \right.$$

$$\left. +16\sigma_{g_{yy}}^2 \frac{B_{f_y}^2}{\mu_g^2} \hat{c}_\gamma^2 \alpha + 8\sigma_{f_y}^2 \hat{c}_\gamma^2 \alpha + \frac{8}{\mu_g \hat{c}_\gamma} L_z^2 \hat{c}_\beta^2 N^2 \sigma_{g_y}^2 \right),$$

where (a) follows from the summation of a geometric progression.

Utilizing the definition of the potential function $\hat{W}_t$ and Jenson's inequality finishes the proof of the theorem.

$\square$

