# OpenReview forum: "A Lazy Hessian Evaluation Framework for Accelerating Stochastic Bilevel Optimization"
_ICLR.cc/2025/Conference — Submitted to ICLR 2025_

### Official Review · Reviewer_pCd9 · 2024-10-28

**Soundness:** 2
**Presentation:** 3
**Contribution:** 2
**Rating:** 5
**Confidence:** 4

**Summary:**

This paper provides a more efficient method, LazyBLO, for bilevel optimization that reduces the frequency of computing the hessian vectors. Provably, it converges faster and presents better experimental performance than other second-order, and first-order methods.

**Strengths:**

1. The presentation is very smooth and makes it easy to read. Notations are declared well.
2. I went through the proof details and it seems rigorous.
3. Codes are provided.

**Weaknesses:**

1. The paper does not seem very novel to me since the "lazy" strategies have already been tried though they may not be the lazy hessian in Bilevel optimization. I understood that no former works provided theoretical guarantees of Lazy hessian methods in BO. After I went through the proof details, it looked straightforward. However, I am open to changing my mind if I missed some important parts.
2. The order selections in the corollaries are a little confusing to me. In Corollary 5.4, I assume that N is chosen as a constant, T is at the order of $\epsilon^{-1}$, and batch sizes are all $\epsilon^{-1}$. Then LazyBLO requires $\epsilon^{-2}$ partial gradients but why $\epsilon^{-2}$ for hessian evaluations? Is it $\mathcal O(T)=\mathcal O(\epsilon^{-1})$. The selection in Corollary 5.7 is more unclear to me. With the choice of a constant level of batch sizes of $D^{f_y}$ and $D^{g_{yy}}$, N cannot be constant and I did not get your results. Correct me if I misunderstand it.
3. Normally, strong convexity will have a better convergence rate or better sample complexity. In the current version, both non-convex and strongly convex settings have the same complexity order in the partial gradients in your corollaries. Why is that?
4. The explanation in the experimental part is not clear. For example, table 2 shows the time to "converge" but there is no explanation of what the standard is (such as loss change is smaller than a threshold). Similarly, table 3 presents the number of hessian computing when reaching the same "training loss". Then how much? I suggest announcing those parts more clearly.
5. In Figure 3, can the authors provide a plot with #hessian computing as the x-axis?

**Questions:**

Please check the weakness.

---

> ### Author Response · Authors · 2024-11-22
>
> ### Weaknesses:
>
> > **Comment 1:** The paper does not seem very novel to me since the "lazy" strategies have already been tried though they may not be the lazy hessian in Bilevel optimization. I understood that no former works provided theoretical guarantees of Lazy hessian methods in BO. After I went through the proof details, it looked straightforward. However, I am open to changing my mind if I missed some important parts.
>
>
> **Our Response:** Thank you for your comment. We would like to emphasize that the design of our LazyBLO algorithm is far from a straightforward adaptation of single-level optimization to the bilevel counterpart. Compared to single-level optimization, bilevel optimization is *fundamentally different* and has a *far more complex problem structure*. Specifically, in bilevel optimization, Hessian information is **essential** for hypergradient computation, rather than serving as an optional enhancement as in second-order methods for single-level optimization.
>
> Moreover, due to the far more complex problem structure, analyzing the use of lazy Hessian in bilevel optimization is far more challenging than that in single-level optimization settings. There are two major challenges in our theoretical analysis:
>
> 1. **Increased Stochastic Hypergradient Error:** The use of lazy Hessian evaluation introduces additional errors in the stochastic hypergradient estimator for the upper-level function.
> 2. **Error Propagation in the Coupled Structure:** Due to the hierarchical and coupled structure of bilevel optimization problems, the error resulting from the stochastic hypergradient estimator with stale Hessian information further propagates to and increases the approximation error of $\mathrm{y}^*(\mathrm{x})$, which is the optimal solution of the lower-level problem, and the approximation error of $\mathrm{z}^*(\mathrm{x},\mathrm{y})$, which is the optimal Hessian-inverse estimation.
>
> Note that $\mathrm{z}^*(\mathrm{x},\mathrm{y})$ is inherently linked to $\mathrm{y}^*(\mathrm{x})$, creating a tight coupling between these errors that significantly complicates the convergence analysis of LazyBLO. These complications in error propagation and approximation are *unseen* in single-level optimization convergence analysis and even in conventional bilevel optimization algorithm analysis, highlighting the novelty of our theoretical contributions.
>
>
> > **Comment 2:** The order selections in the corollaries are a little confusing to me. In Corollary 5.4, I assume that N is chosen as a constant, T is at the order of $\epsilon^{-1}$, and batch sizes are all $\epsilon^{-1}$. Then LazyBLO requires $\epsilon^{-2}$ partial gradients but why $\epsilon^{-2}$ for hessian evaluations? Is it $\mathcal{O}(T)=\mathcal{O}(\epsilon^{-1})$. The selection in Corollary 5.7 is more unclear to me. With the choice of a constant level of batch sizes of $D^{f_{y}}$ and $D^{g_{yy}}$, N cannot be constant and I did not get your results. Correct me if I misunderstand it.
>
>
> **Our Response:** Thanks for your comment. In Corollary 5.4, the number of hessian evaluations is calculated as the product of the number of iterations $T$ and the batch size $D^{g_{yy}}$. Specifically:
> * $\mathcal{O}(T)=\mathcal{O}(\epsilon^{-1})$, and
> * $\left| \mathcal{D}^{g_{yy}} \right|=\Theta(\epsilon^{-1})$.
>
> This results in $\mathcal{O}(\epsilon^{-2})$ Hessian evaluations.
>
> In Corollary 5.7, we acknowledge a typo in the original paper. Specifically, we missed a $1/N$ term in the theorem's expression. Based on the corrected inequality:
> $$\frac{1}{N}\sum_{n=0}^{N-1}\mathbb{E}\bigg[\ell\left(\mathrm{x}_t^n\right)-\ell^*\bigg] \leq \left(1-\mu_f \alpha\right)^t\hat{\Delta}_0$$
>
> $$  \qquad \qquad \qquad \qquad \qquad \quad +\mathcal{O}\bigg(\sigma_{g_{xy}}^2 + \sigma_{f_{x}}^2  + \frac{1}{N^{4}}\sigma_{g_{yy}}^2 +\frac{1}{N^{4}} \sigma_{f_{y}}^2 + \frac{1}{N}\sigma_{g_{y}}^2 \bigg),$$ where
> $$\hat{\Delta}_0=\frac{1}{N} \sum _{n=0}^{N-1}\left(\ell\left(\mathrm{x}_0^n\right)-\ell^*\right)+\frac{1}{N} \sum _{n=0}^{N-1}\left\|\mathrm{y}_0^n-\mathrm{y}^*\left(\mathrm{x}_0^n\right)\right\|^2+\frac{1}{N}\left\|\mathrm{z}_0-\mathrm{z}_0^*\right\|^2,$$
>
> we have $\mathcal{O}(T)=\mathcal{O}(\log\epsilon^{-1})$.
>
> By choosing
> * $\left| \mathcal{D}^{f_{x}} \right|,\left| \mathcal{D}^{g_{xy}} \right|=\Theta\left( \epsilon^{-1} \right)$,
> * $\left| \mathcal{D}^{f_{y}} \right|,\left| \mathcal{D}^{g_{yy}} \right|=\Theta\left( N^{-4}\epsilon^{-1} \right)$, and
> * $\left| \mathcal{D}^{g_{y}} \right|=\Theta\left( N^{-1}\epsilon^{-1} \right)$,
>
> LazyBLO requires $\mathcal{O}(\epsilon^{-1}\log\epsilon^{-1})$ partial gradient evaluations and  $\mathcal{O}(N^{-4}\epsilon^{-1}\log\epsilon^{-1})$ HVP evaluations to reach an $\epsilon$-optimal point which significantly improves the performance compared to the non-convex setting. We have corrected the typo in the revised version of the paper.

---

> ### Author Response · Authors · 2024-11-22
>
> > **Comment 3:** Normally, strong convexity will have a better convergence rate or better sample complexity. In the current version, both non-convex and strongly convex settings have the same complexity order in the partial gradients in your corollaries. Why is that?
>
>
> **Our Response:** Thank you for your question. After addressing the typo mentioned earlier in the results for the strongly convex setting, we find that the strongly convex case requires $\mathcal{O}(\epsilon^{-1}\log\epsilon^{-1})$ partial gradient evaluations, which is fewer than the $\mathcal{O}(\epsilon^{-2})$ partial gradient evaluations required in the non-convex setting. In addition, the batch sizes $\left| \mathcal{D}^{g_{y}} \right|$, $\left| \mathcal{D}^{f_{y}} \right|$, $\left| \mathcal{D}^{g_{yy}} \right|$ required in the strongly convex setting are smaller than those required in the non-convex setting.
>
>
> > **Comment 4:** The explanation in the experimental part is not clear. For example, table 2 shows the time to "converge" but there is no explanation of what the standard is (such as loss change is smaller than a threshold). Similarly, table 3 presents the number of hessian computing when reaching the same "training loss". Then how much? I suggest announcing those parts more clearly.
>
>
> **Our Response:** Thank you for your suggestion. In Table 2, convergence is defined as the point where the training loss reaches its minimum and does not decrease further for 5 consecutive iterations. For Table 3, the training loss criteria for Task 1 is 2.35 and for Task 2 is 0.71. We will clarify these in the revised version of our paper.
>
>
> > **Comment 5:** In Figure 3, can the authors provide a plot with #hessian computing as the x-axis?
>
>
> **Our Response:** Thank you for the suggestion. As per your suggestion we have added a figure using the number of Hessian computations as the x-axis in the revised version of our paper (see Figure 3(b)).

---

> > ### Comment · Reviewer_pCd9 · 2024-11-25
> >
> > Thanks for the detailed response.
> > For my comment on the novelty, I am familiar with Bilevel optimization and aware of the importance of hessian-inverse-vector approximation in the second-order method. The challenges you mentioned are natural when applying lazy hessian tricks. I do not see enough novel techniques in both algorithms and analysis. Meanwhile, the rest questions are solved. I would like to keep my score.

---

> > > ### Author Response · Authors · 2024-11-27
> > >
> > > > **Comment:** Thanks for the detailed response. For my comment on the novelty, I am familiar with Bilevel optimization and aware of the importance of hessian-inverse-vector approximation in the second-order method. The challenges you mentioned are natural when applying lazy hessian tricks. I do not see enough novel techniques in both algorithms and analysis. Meanwhile, the rest questions are solved. I would like to keep my score.
> > >
> > >
> > >
> > > **Our Response:** Thank you for your follow-up comment and for taking the time to carefully evaluate our work. We respectfully disagree with the assessment that our work lacks novelty. While the idea of using lazy Hessian updates might seem natural, rigorously analyzing and proving its **identical convergence rate compared to its non-lazy counterpart** and showing its **effectiveness in training time saving** in bilevel optimization is far from straightforward. Below, we provide additional clarifications on the novelty of our contributions:
> > >
> > > 1. **Publication Worthiness of "Lazy Update" Techniques in Machine Learning:** We also respectfully disagree that lazy Hessian is just a trick and hence is not worth publishing. Quite the contrary, the idea of *"lazy update exploiting small-timescale positive correlations"* has played a **foundational role** in a large numbmer of subfields in optimization algorithm design for machine learning and led to some of the most well-known and significant algorithms in various learning paradigms, including but not limited to "FedAvg" [1] in federated learning (laziness in communications), "SVRG" [2] in variance-reduced first-order optimizatio approahces (laziness in full gradient evaluations), and the actor-critic framework [3] in reinforcement learning (laziness in value function updates). Due to the errros induced by laziness, judicious algorithmic designs and highly non-trivial analysis are needed in these "lazy update" approaches. As further evidence to show the significance of the "lazy update" idea, in what follows, we have collected some related works on "lazy updates" in the areas of single-level optimization, stochastic approximation, zeroth-order optimization, and graph neural networks. For example, lazy updates have been successfully applied in [4]-[10], where signficant challenges have been overcome to establish their theoretical performance guarantees. **Compared to Refs. [1]-[10], our analysis of lazy Hessian techniques in bilevel optimization is even more complex due to the coupled bilevel structure, which justifies its worthiness of publication to disseminate the new knowledge we discovered with the machine learning community**.
> > >
> > >
> > > 2. **Non-Trivial Challenges in Analysis:** The theoretical understanding of lazy Hessian techniques in bilevel optimization involves addressing non-trivial challenges, including:
> > >    * **Stochastic Hypergradient Error:** Lazy Hessian updates introduce additional errors in the hypergradient estimation, which must be carefully bounded and analyzed.
> > >    * **Error Propagation due to Laziness in the Coupled Structure in Bilevel Optimization:** In bilevel optimization, the hierarchical and coupled nature of the problem means that errors from stale Hessians propagate and compound, affecting both the lower-level solution and the Hessian-inverse-vector approximation. Managing this error propagation while ensuring **theoretical convergence guarantee** is a significant theoretical challenge that has not been addressed in prior work of bilevel optimization.
> > >
> > > 3. **New Knowledge for the Machine Learning Community:** To the best of our knowledge, our work is the first to rigorously establish theoretical convergence guarantees for lazy Hessian techniques in bilevel optimization. This contribution represents new knowledge for the field, providing a solid foundation for future research on computationally efficient bilevel optimization methods. The theoretical guarantees we provide ensure that the lazy Hessian approach works reliably, making it a valuable contribution to the community.
> > >
> > >
> > > In conclusion, while the "lazy update" idea may appear to be intuitive in bilevel optimization, proving its identical theoretical  convergence rate compared to its non-lazy counterpart is **fundamentally hard** and necessitates novel proof techniques (cf. our responses in the previous round). Since the reviewer stated that our techniques in algorithms and analysis are not novel, **we would highly appreciate if the reviewer can pinpoint exactly which part(s) of our work are not novel and provide corresponding evidence and reference source. If our work is indeed not novel, this will help us improve in the future.**

---

> > > ### Author Response · Authors · 2024-11-27
> > >
> > > **References:**
> > >
> > > [1] McMahan, Brendan, Eider Moore, Daniel Ramage, Seth Hampson, and Blaise Aguera y Arcas. "Communication-efficient learning of deep networks from decentralized data." In Artificial intelligence and statistics, pp. 1273-1282. PMLR, 2017.
> > >
> > > [2] Johnson, Rie, and Tong Zhang. "Accelerating stochastic gradient descent using predictive variance reduction." Advances in neural information processing systems 26 (2013).
> > >
> > > [3] Konda, Vijay, and John Tsitsiklis. "Actor-critic algorithms." Advances in neural information processing systems 12 (1999).
> > >
> > > [4] Doikov, Nikita, and Martin Jaggi. "Second-order optimization with lazy hessians." In International Conference on Machine Learning, pp. 8138-8161. PMLR, 2023.
> > >
> > > [5] Lampariello, Francesco, and Marco Sciandrone. "Global convergence technique for the Newton method with periodic Hessian evaluation." Journal of optimization theory and applications 111 (2001): 341-358.
> > >
> > > [6] Wang, Chang-yu, Yuan-yuan Chen, and Shou-qiang Du. "Further insight into the Shamanskii modification of Newton method." Applied mathematics and computation 180, no. 1 (2006): 46-52.
> > >
> > > [7] Adler, Ilan, Zhiyue T. Hu, and Tianyi Lin. "New proximal Newton-type methods for convex optimization." In 2020 59th IEEE Conference on Decision and Control (CDC), pp. 4828-4835. IEEE, 2020.
> > >
> > > [8] Fan, Jinyan. "A Shamanskii-like Levenberg-Marquardt method for nonlinear equations." Computational Optimization and Applications 56, no. 1 (2013): 63-80.
> > >
> > > [9] Xiao, Quan, Qing Ling, and Tianyi Chen. "Lazy queries can reduce variance in zeroth-order optimization." IEEE Transactions on Signal Processing (2023).
> > >
> > > [10] Xue, Rui, Haoyu Han, MohamadAli Torkamani, Jian Pei, and Xiaorui Liu. "Lazygnn: Large-scale graph neural networks via lazy propagation." In International Conference on Machine Learning, pp. 38926-38937. PMLR, 2023.

---

### Official Review · Reviewer_z4EQ · 2024-11-03

**Soundness:** 2
**Presentation:** 2
**Contribution:** 3
**Rating:** 5
**Confidence:** 2

**Summary:**

The paper introduces a bilevel optimization algorithm that reduces computational costs by infrequently computing Hessian.

**Strengths:**

The algorithm effectively addresses a major difficulty in bilevel optimization by reducing the need for frequent, costly Hessian computations.

**Weaknesses:**

See the questions.

**Questions:**

1. While the problem of reducing Hessian computations is valid, the main technical novelty of LazyBLO is unclear. The approach reduces Hessian computation, which may seem like an incremental adaptation of existing methods rather than a novel algorithm. It would be helpful if you can specify them.
2. Could you provide more insights into how LazyBLO performs on problems where Hessian is unstable?

---

> ### Author Response · Authors · 2024-11-22
>
> ### Questions:
>
> > **Comment 1:** While the problem of reducing Hessian computations is valid, the main technical novelty of LazyBLO is unclear. The approach reduces Hessian computation, which may seem like an incremental adaptation of existing methods rather than a novel algorithm. It would be helpful if you can specify them.
>
>
> **Our Response:** Thank you for your comment. While reducing Hessian computations has been explored in single-level optimization to accelerate second-order methods, this is the first time such an approach has been applied to bilevel optimization. Before our work, it was unclear whether reducing Hessian computations was feasible or not in bilevel optimization due to the complexity of the bilevel problems.
>
> Importantly, our lazy Hessian algorithm design is far from being a simple adaptation from single-level optimization to the bilevel counterpart. Compared to single-level optimization, bilevel optimization is *fundamentally different* and has a *far more complex problem structure*. Specifically, in bilevel optimization, Hessian information is **essential** for hypergradient computation, rather than being an optional enhancement as in second-order methods for single-level optimization.
>
> Furthermore, due to the far more complex problem structure, analyzing the use of lazy Hessian in bilevel optimization introduces substantial challenges that are not present in single-level optimization settings. These challenges include:
>
> 1. **Increased Stochastic Hypergradient Error:** The use of lazy Hessian evaluation introduces additional errors into the stochastic hypergradient estimator for the upper-level function.
> 2. **Error Propagation in the Coupled Structure:** Due to the hierarchical and coupled structure of bilevel optimization problems, the error resulting from the stochastic hypergradient estimator with stale Hessian information further propagates to and increases the approximation error of $\mathrm{y}^*(\mathrm{x})$, which is the optimal solution of the lower-level problem, and the approximation error of $\mathrm{z}^*(\mathrm{x},\mathrm{y})$, which is the optimal Hessian-inverse estimation.
>
> Note that $\mathrm{z}^*(\mathrm{x},\mathrm{y})$ is inherently linked to $\mathrm{y}^*(\mathrm{x})$, creating a tight coupling between these errors that significantly complicates the convergence analysis of LazyBLO. These complexities in error propagation and approximation are *unseen* in single-level optimization convergence analysis, even in the conventional bilevel optimization algorithm analysis.
>
>
>
>
> > **Comment 2:** Could you provide more insights into how LazyBLO performs on problems where Hessian is unstable?
>
>
> **Our Response:** Thanks for your question. We assume your comment that "Hessian is unstable" refers to the situations where the Hessian matrix varies rapidly over time. If this is the case (e.g., in highly non-linear models), LazyBLO can be adapted by increasing the frequency of Hessian-vector product evaluations. This flexibility allows the algorithm to remain robust and perform effectively even in these challenging scenarios.

---

### Official Review · Reviewer_vwLm · 2024-11-03

**Soundness:** 2
**Presentation:** 2
**Contribution:** 2
**Rating:** 5
**Confidence:** 4

**Summary:**

This paper introduces LazyBLO, an algorithm designed to reduce the frequency of Hessian-vector product (HVP) evaluations in bilevel optimization, particularly for stochastic problems with nonconvex upper-level objectives and strongly convex lower-level objectives. By employing a lazy strategy, LazyBLO improves computational efficiency, achieving ${\cal O}(\epsilon^{-2})$ HVP complexity for stochastic cases. Theoretical convergence guarantees are provided for both nonconvex and strongly convex cases, and empirical results from data hyper-cleaning and deep hyper-representation tasks validate the effectiveness and HVP computational savings of the proposed method.

**Strengths:**

Proposed method is effective in saving HVP computation.

**Weaknesses:**

1. The convergence rate of the proposed method depends on increasing batch sizes. However, the current state-of-the-art theoretical guarantees for stochastic bilevel optimization do not require this, as demonstrated by methods such as ALSET, SOBA, FSLA, F$^2$SA, and etc.

2. Although the proposed method achieves the same convergence rate to fully single-loop bilevel algorithms with respect to $\epsilon$  for general nonconvex–strongly convex bilevel problems, the HVP computational complexity appears to be $T={\cal O}(N\epsilon^{-1})$, which is even higher than that of fully single-loop bilevel algorithms. This implies that the lazy strategy not only offers no theoretical improvement, but also is even worse than its non-lazy counterpart. Thus, the results presented here are not unexpected.

3. The motivation for reducing only HVP computations, rather than both HVP and JVP computations, is unclear. In fully single-loop bilevel methods, both HVP and JVP are computed once per iteration, unlike the Neumann series approach discussed in this paper. There is no apparent reason to prioritize reducing only HVP computations. HVP and JVP should be considered equally important, particularly in single-loop methods, which form the basis of the proposed approach. Besides, it is better to provide the computation complexity comparison of the proposed method in terms of HVP and JVP.

**Questions:**

It would be better to add the complexity of the proposed method in Table 1 for comparison as well.

---

> ### Author Response · Authors · 2024-11-22
>
> ### Weaknesses:
>
> > **Comment 1:** The convergence rate of the proposed method depends on increasing batch sizes. However, the current state-of-the-art theoretical guarantees for stochastic bilevel optimization do not require this, as demonstrated by methods such as ALSET, SOBA, FSLA, F$^2$SA, and etc.
>
>
> **Our Response:** The need for increasing batch sizes is not unique to LazyBLO. Many existing bilevel optimization algorithms, such as stocBiO, AmIGO, and iNEON [R1], also adopt increasing batch sizes in their theoretical analyses to control variance. Additionally, our approach employs SGD to approximate Hessian-vector product by solving a quadratic problem. Previous bilevel optimization algorithms using the same approximation are AmIGO and SOBA. Our approach shares a similarity with AmIGO, which avoids strong third-order Lipschitz continuity assumptions at the cost of requiring increasing batch sizes. In contrast, SOBA relies on these stronger assumptions to manage variance. In addition, previous methods, such as BSA, TTSA, SUSTAIN, MSTSA, VRBO, MRBO and ALSET require all the stochastic functions to satisfy Assumptions 3.1 and 3.2 stated in the manuscript to avoid increasing batch gradient computations. Note that this is certainly a very strong assumption and is not required in our analysis of LazyBLO. This indicates that increasing batch sizes are a common trade-off in bilevel optimization methods, rather than a disadvantage specific to LazyBLO.
>
> It is also important to note that ALSET adopts a different strategy, using Neumann series to approximate the Hessian inverse, which distinguishes it from LazyBLO. While FSLA and F$^2$SA do not require increasing batch sizes, they introduce additional assumptions. Specifically, Assumptions A and B in FSLA are also hold for their stochastic version, which are not required in our analysis; and Assumptions 4 and 5 in F$^2$SA are not required for our LazyBLO.
>
>
> [R1] Huang, M., Ji, K., Ma, S., & Lai, L. (2022). Efficiently escaping saddle points in bilevel optimization. arXiv preprint arXiv:2202.03684.
>
>
> > **Comment 2:** Although the proposed method achieves the same convergence rate to fully single-loop bilevel algorithms with respect to $\epsilon$ for general nonconvex–strongly convex bilevel problems, the HVP computational complexity appears to be $T=\mathcal{O}(N\epsilon^{-1})$, which is even higher than that of fully single-loop bilevel algorithms. This implies that the lazy strategy not only offers no theoretical improvement, but also is even worse than its non-lazy counterpart. Thus, the results presented here are not unexpected.
>
>
> **Our Response:** Thank you for your comment. We would like to clarify that the order of the computational complexity of our LazyBLO is $-2$, which matches the state-of-the-art SGD-based bilevel optimization algorithms. In addition, since we use a lazy approach where infrequent Hessian evaluations incurs additional errors, our goal is *not to outperform* the non-lazy approaches. Rather, our goal is to be **as close to the non-lazy counterpart approaches as possible**. Due to the additional errors accumulated because of the use of these stale Hessians, approximate HVP evaluations, and the coupling hierarchical structure of the bilevel problems, it is unclear whether LazyBLO will converge or not. In fact, establishing convergence for the lazy approach proposed in this work is itself **surprising** and a major contribution of our work.

---

> ### Author Response · Authors · 2024-11-22
>
> > **Comment 3:** The motivation for reducing only HVP computations, rather than both HVP and JVP computations, is unclear. In fully single-loop bilevel methods, both HVP and JVP are computed once per iteration, unlike the Neumann series approach discussed in this paper. There is no apparent reason to prioritize reducing only HVP computations. HVP and JVP should be considered equally important, particularly in single-loop methods, which form the basis of the proposed approach. Besides, it is better to provide the computation complexity comparison of the proposed method in terms of HVP and JVP.
>
>
> **Our Response:** Thank you for your comment. We provide the following reasons for prioritizing the reduction of HVP computations:
>
> 1. Most bilevel optimization algorithms follow a double-loop structure and require **multiple** HVP computations per iteration.
> 2. While single-loop algorithms represent a subset of bilevel optimization methods and often compute HVPs only once per iteration, there are notable exceptions. Some single-loop algorithms, such as SUSTAIN, TTSA, BSA, and ALSET, require **multiple** HVP computations per iteration. In contrast, JVPs are computed only once in all these algorithms, regardless of the number of HVP computations per iteration.
>
> Given these points, HVP computations are **at least as frequent as JVP computations** and incur a more significant computational bottleneck, especially in double-loop and certain single-loop bilevel algorithms. As such, reducing the number of HVP computations is more crucial than reducing JVP computations.
>
>
>
> ### Questions:
>
> > **Comment 1:** It would be better to add the complexity of the proposed method in Table 1 for comparison as well.
>
>
> **Our Response:** Thank you for your suggestion. We have included the complexity of our proposed method in Table 1 in the revised version of the paper.

---

> ### Comment · Reviewer_vwLm · 2024-11-27
>
> Thank you for your response. For the first question, I have reviewed the references you provided, and it appears that methods like TTSA, SUSTAIN, and ALSET do not impose any Lipschitz or strong convexity assumptions on the stochastic function, which aligns with your Assumptions 3.1–3.2. Furthermore, these methods also do not require an increasing batch size. For the third-order Lipschitz continuity assumption in SOBA, it is because they use fully single loop so that this assumption is needed to avoid the increasing batch size. However, the method you used is double loop, removing increasing batch size does not need the third order Lipschitz continuity; See TTSA, SUSTAIN, and ALSET. Therefore, I believe that eliminating the increasing batch size assumption would be necessary to achieve state-of-the-art theoretical performance.
>
> On the second point, I still have some reservations. Fully single-loop algorithms such as SOBA and FLSA represent a recent trend in bilevel optimization as they avoid frequent HVP computations per iteration, aligning them with the direction of your work. Comparing against double-loop algorithms, which are not considered the current baselines, seems less convincing as a justification for focusing on the reduction of HVP rather than both HVP and JVP. Are there specific challenges or limitations that prevent skipping JVP entirely in your framework?
>
> Based on the current response, I would like to keep my score.

---

> > ### Author Response · Authors · 2024-11-29
> >
> > > **Comment 1:** Thank you for your response. For the first question, I have reviewed the references you provided, and it appears that methods like TTSA, SUSTAIN, and ALSET do not impose any Lipschitz or strong convexity assumptions on the stochastic function, which aligns with your Assumptions 3.1–3.2. Furthermore, these methods also do not require an increasing batch size. For the third-order Lipschitz continuity assumption in SOBA, it is because they use fully single loop so that this assumption is needed to avoid the increasing batch size. However, the method you used is double loop, removing increasing batch size does not need the third order Lipschitz continuity; See TTSA, SUSTAIN, and ALSET. Therefore, I believe that eliminating the increasing batch size assumption would be necessary to achieve state-of-the-art theoretical performance.
> >
> > **Our Response:** Thank you for your comment. We would like to clarify a few points regarding the assumptions in the referenced methods:
> >
> > 1. **Assumptions on Stochastic Functions (more precisely, sample-wise assumptions on $f(x,y;\xi)$ and $g(x,y;\zeta)$, where $\xi$ and $\zeta$ are data samples) in SUSTAIN, TTSA, and ALSET:**
> >    * For SUSTAIN [3], please refer to Assumption 3 in the SUSTAIN paper [3], which specifies the Lipschitz smoothness and strong convexity assumptions on the stochastic function.
> >    * Although TTSA [2] and ALSET [4] did not explicitly state the Lipschitz smoothness and strong convexity assumptions on the stochastic function in their papers, TTSA **implicitly** relies on Lemma 3.2 in [1] for its theoretical analysis (**see Appendix F.3 in [2]**). The proof of Lemma 5 in the ALSET paper [4] relies on the strong-convexity and the Lipschitz smoothness assumptions on the stochastic function since Lemma 11 in [2] and Lemma 3.2 in [1] are used. Notably, **Lemma 3.2 in [1] is derived based on the strong-convexity and the Lipschitz smoothness assumptions on the stochastic function**. Specifically, on **Page 3 of [1]**, the authors stated that *"Assumptions 1 and 2 will be made throughout the paper for both deterministic and stochastic functions"*.
> >
> >      Since TTSA, SUSTAIN, and ALSET all employed the **sample-wise** strong-convexity and the Lipschitz smoothness assumptions on the stochastic function, they are able to bound the variance of the stochastic gradient estimation (see Lemma 1 in TTSA, Lemma B.1 in SUSTAIN [3], and Lemma 5 in ALSET [4]) and thereby relax the requirement for increasing batch sizes. In stark contrast, to abandon these strong **sample-wise** assumptions on the stochastic objective functions, we elect to use the adaptive-batch-size approach to establish our convergence results.
> >
> > 2. **Regarding Single-Loop vs. Double-Loop Structure:**
> > While our proposed algorithm seemingly uses a double-loop structure, it can be easily re-written in a "single-loop" form, where the updates of $\mathbf{x}$ and $\mathbf{y}$ are both stated as "one-line" subroutines within the same loop, which aligns with the structure of SOBA. Here, we would like to point out that, when classifying bilevel optimization (BLO) alglorithms, the real **fundamental difference** in terms of algorithmic structure is *"single-timescale vs. two-timescale,"* rather than *"single-loop vs. double-loop."* Specifically, let $\alpha_t$ and $\beta_t$ denote the upper-level and lower-level learning rates, respectively. Then, a BLO algorithm is referred to as "single-timescale" if $\alpha_t/\beta_t = \text{Constant}$, $\forall t$. Otherwise, it is referred to as "two-timescale" if $\alpha_t/\beta_t \rightarrow 0$ or $\alpha_t/\beta_t \rightarrow \infty$ as $t\rightarrow \infty$.
> >
> >    With the above notions on timescale differences, it is clear that both SOBA [5] and our LazyBLO approach are single-timescale algorithms, even though LazyBLO is written in a "double-loop" form. In this sense, SOBA and our LazyBLO methods are on an "equal footting." However, to achieve convergence guarantee, SOBA relies on the stronger third-order Lipschitz continuity assumption, while we elect to use large batch sizes to relax this strong assumption. Consequently, given the same single-timescale algorithmic nature between LazyBLO and SOBA, removing the increasing batch size in our LazyBLO method would require the stronger third-order Lipschitz continuity assumption to compensate.
> >
> > [1] Approximation Methods for Bilevel Programming: https://arxiv.org/pdf/1802.02246
> >
> > [2] TTSA: https://arxiv.org/pdf/2007.05170
> >
> > [3] SUSTAIN: https://proceedings.neurips.cc/paper_files/paper/2021/file/fe2b421b8b5f0e7c355ace66a9fe0206-Paper.pdf
> >
> > [4] ALSET: https://arxiv.org/pdf/2106.13781
> >
> > [5] SOBA: https://proceedings.neurips.cc/paper_files/paper/2022/file/aa84ec1ac3f5fdcf77bce2c22705ab77-Paper-Conference.pdf

---

> > ### Author Response · Authors · 2024-11-29
> >
> > > **Comment 2:** On the second point, I still have some reservations. Fully single-loop algorithms such as SOBA and FLSA represent a recent trend in bilevel optimization as they avoid frequent HVP computations per iteration, aligning them with the direction of your work. Comparing against double-loop algorithms, which are not considered the current baselines, seems less convincing as a justification for focusing on the reduction of HVP rather than both HVP and JVP. Are there specific challenges or limitations that prevent skipping JVP entirely in your framework?
> >
> > **Our Response:** Thank you for your comment. We assume "FLSA" is a typo and that you are referring to FSLA. We would like to clarify that both SOBA and FSLA do not skip any Hessian computations like the proposed LazyBLO framework; instead, they avoid explicitly computing the Hessian inverse but they still compute Hessian (and Jacobian) vector products in each iteration.
> >
> > * **SOBA** requires one HVP computation per iteration (see Equation (10b) in the SOBA paper), resulting in $T$ HVP computations over $T$ iterations. In contrast, our proposed algorithm reduces the number of HVP computations to $T/N$ if $T$ is the total number of iterations. We have compared our algorithm with SOBA in our experiments, and the results in Table 2 show that SOBA requires 126 times more Hessian computations compared to the proposed LazyBLO algorithm.
> > * **FSLA** also requires one HVP computation per iteration (see Equation (10) in the FSLA paper), leading to $T$ HVP computations in total, which is more than the number required by our algorithm. In addition, FSLA employs a **momentum-based variance reduction** technique to accelerate convergence, hence not being a fair comparison to our stochastic-gradient-only approach. We note that the momentum-based variance reduction technique could also be incorporated into our proposed framework to further enhance its performance. But we chose to focus on a stochastic-gradient-only approach in order **not** to dilute the key "LazyHessian" idea.
> >
> > Regarding the reduction of both HVP and JVP, our framework can indeed be adapted to achieve this (please see Algorithm 1 below). In Algorithm 1, the computation frequency of the JVP is reduced, and JVP ($JVP_{t+1}$) would be computed in the outer loop. Moreover, since $z_t$ is the HVP estimator, reducing the computation frequency of JVP also reduces the computation frequency of HVP. Thus, the reduction in computation frequency for both JVP and HVP is coupled.
> >
> > The convergence analysis of such a modification would be significantly more challenging than that of LazyBLO due to the "multiplicative" structure of $JVP_t$, which is coupled with both JVP and HVP. This "Joint-Lazy-JVP-HVP" approach presents a non-trivial theoretical challenge and deserves an independent paper dedicated to this new algorithm. We thank the reviewer for motivating this new research direction for our future studies!
> >
> >
> > |Algorithm 1: The Algorithm for Reducing both HVP and JVP Computations.|
> > | -------- |
> > | **Input:** Initial parameters $x_{0}^{0}$, $y_{0}^{0}$, $z_{0}$, and stepsize $\\{ \alpha_{t}, \beta_{t}, \gamma_{t} \\}_{t=0}^{T-1}$     |
> > | **for** $t=0$ **to** $T-1$ **do** |
> > | $\quad$ **for** $n=0$ **to** $N-1$ **do** |
> > |$\qquad$ Initialize $x_{t}^{0}=x_{t-1}^{N}$ and $y_{t}^{0}=y_{t-1}^{N}$|
> > |$\qquad$ Sample data batches $D_{t,n}^{g}$ and $D_{t,n}^{f_{x}}$|
> > |$\qquad$ Compute the gradient estimate $h_{t,n}^{g}$ using $h_{t,n}^{g}=\nabla_{y} g\left(x_{t}^{n}, y_{t}^{n}; D_{t,n}^{g}\right)$ and update $y_{t}^{n+1}=y_{t}^{n}-\beta_{t}h_{t,n}^{g}$|
> > |$\qquad$ Compute the gradient estimate $h_{t,n}^{f}$ using $h_{t,n}^{f}  =  \nabla_{x} f  \left(  x_{t}^{n}, y_{t}^{n}; D_{t,n}^{f_{x}}  \right)  +  JVP_t$ and update $x_{t}^{n+1}=x_{t}^{n}-\alpha_{t}h_{t,n}^{f}$|
> > |$\quad$ **end for**|
> > |$\quad$ Sample data batches $D_{t}^{g_{yy}}$,  $D_{t}^{f_{y}}$, and $D_{t}^{g_{xy}}$|
> > |$\quad$ Compute the gradient estimate $h_{t}^{q}$ using ${\color{blue} h_{t}^{q}=\nabla_{y y}^{2} g(x_{t}^{N}, y_{t}^{N}; D_{t}^{g_{yy}}) z_{t}+\nabla_{y} f(x_{t}^{N}, y_{t}^{N}; D_{t}^{f_{y}} ) }$ and update $z_{t+1}=z_{t}-\gamma_{t}h_{t}^{q}$|
> > |$\quad$ Compute ${\color{blue} JVP_{t+1} = \nabla_{x y}^{2} g  \left(x_{t}^{N}, y_{t}^{N}; D_{t}^{g_{xy}}\right)z_{t+1} }$ |
> > |**end for**|

---

> > > ### Comment · Reviewer_vwLm · 2024-12-02
> > >
> > > Thank you for your detailed response. Regarding my first concern, I don’t believe there is any implicit assumption on stochastic functions for TTSA or ALSET. The original analysis for BSA is indeed not tight, which necessitates the stochastic function assumption; however, TTSA improves upon this analysis. For reference, the analysis of TTSA you are referring to (Appendix F.3 in [2]) only requires the first part of Lemma 3.2 in [1]. This is a simple fact that does not rely on any stochastic function assumption. Specifically, the unbiasedness from independent samples, combined with  $a^{-1}=\sum_{n=0}^\infty (1-a)^n$, suffices to achieve the desired results. Since ALSET builds upon TTSA, it also does not require stochastic assumptions.
> > >
> > > Additionally, I respectfully disagree with your assertion that the third-order smoothness assumption is used to eliminate the large batch size requirement. For clarity, you may refer to the proof of SOBA, which better illustrates the necessity of this assumption in ensuring the smoothness of the solution mapping for the single-loop update. Moreover, the analysis of SOBA has been tightened in a recent work (https://arxiv.org/abs/2410.13743), which removes the need for the third-order Lipschitz assumption and might help clarify the role of this assumption.
> > >
> > > As for my second concern, while I understand that your algorithm can be reduced to a fully single-loop approach, similar to FSLA and SOBA, I would still recommend exploring JVP reduction in addition to HVP. This could provide further benefits or insights.
> > >
> > > Overall, while the paper addresses interesting points, I believe there is room for theoretical improvement including but not limited to the batch size requirement, order-wise saving improvement, JVP saving, etc, and the novelty is somewhat limited. Therefore, I will keep my current score.

---

> > > > ### Author Response · Authors · 2024-12-03
> > > >
> > > > > **Comment 1:** Thank you for your detailed response. Regarding my first concern, I don’t believe there is any implicit assumption on stochastic functions for TTSA or ALSET. The original analysis for BSA is indeed not tight, which necessitates the stochastic function assumption; however, TTSA improves upon this analysis. For reference, the analysis of TTSA you are referring to (Appendix F.3 in [2]) only requires the first part of Lemma 3.2 in [1]. This is a simple fact that does not rely on any stochastic function assumption. Specifically, the unbiasedness from independent samples, combined with $a^{-1} = \sum_{n=0}^{\infty }(1-a)^n$, suffices to achieve the desired results. Since ALSET builds upon TTSA, it also does not require stochastic assumptions.
> > > >
> > > >
> > > > **Our Response:** Thank you for your comments. We respectfully disagree with the reviewer’s assertion that ALSET does not require sample-wise assumptions on the LL stochastic objective. Based on the timeline of uploads made to arXiv, the proof of ALSET relies on Lemma 11 in version 2 of the TTSA paper on arXiv. Please see the following link (https://arxiv.org/pdf/2007.05170v2). Note that in version 2 of TTSA available on arXiv, Lemma 11 is derived from Equation (106), which is a slight modification of the proof of Lemma 3.2 in BSA. Here, we provide a brief proof of the result (Equation (106) in TTSA version 2) to establish the requirement on the sample-wise assumptions required by ALSET. Note from the definition of $H_{yy}$ in https://arxiv.org/pdf/2007.05170v2 that
> > > >
> > > > $$
> > > > \left\\|\mathbb{E}\left[H_{y y}\right]\right\\|^2  \leq\left(\mathbb{E}\left[\left\\|H_{y y}\right\\|\right]\right)^2
> > > >  {\leq} \left( \frac{b}{L_g} \mathbb{E}\left[\prod_{i=1}^r\left\\|I-\frac{1}{L_g} \nabla_{y y}^2 G\left(\bar{x}, \bar{y}, \zeta_i\right)\right\\|\right] \right)^2
> > > >  \overset{(a)}{\leq} \left( \frac{b}{L_g} \mathbb{E}_r\left[1-\frac{\mu_g}{L_g}\right]^r \right)^2
> > > >  =\left( \frac{1}{L_g} \sum _{i=0}^{b-1}\left[1-\frac{\mu_g}{L_g}\right]^i \right)^2 \leq \mu_g^{-2},
> > > > $$
> > > >
> > > > where, inequality $(a)$ above holds **precisely because of the strong convexity and the Lipschitz-smoothness of the LL stochastic function**. Note that from this derivation, the final expression of the variance of the hypergradient in ALSET can be easily derived and is equal to
> > > >
> > > > $$ \mathbb{E}\left[\left\\|h_f^k-\bar{h}_f^k\right\\|^2\right] $$
> > > >
> > > > $$\leq \sigma_f^2+\frac{3}{\mu_g^2}\left[\left(\sigma_f^2+\ell_{f, 0}^2\right)\left(\sigma_{g, 2}^2+2 \ell_{g, 1}^2\right)+\sigma_f^2 \ell_{g, 1}^2\right]=: \tilde{\sigma}_f^2=\mathcal{O}\left(\kappa^2\right).$$
> > > >
> > > > Note that this expression **exactly matches** the variance expression in TTSA version 2 (see Equation (104)). Note from the derivation above it is easy to see that this expression requires sample-wise smoothness and strong convexity of the lower level objective.
> > > >
> > > > Also, note that the latest version of TTSA (version 4, https://arxiv.org/pdf/2007.05170v4) removes the stochastic function assumption, however, this comes at a cost. Specifically, the bound in Equation (112) **worsens** as it becomes dependent on the dimension of $x$ (see Equation (112) in TTSA version 4). **Furthermore, we have consulted with the authors of TTSA, and they agree with our interpretation of the result**.
> > > >
> > > > In summary, we would like to stress again that the results presented in ALSET rely on the assumtions made on the stochastic lower level objective function that are based on the results derived in TTSA version 2 available on arXiv.

---

> > > > ### Author Response · Authors · 2024-12-03
> > > >
> > > > > **Comment 2:** Additionally, I respectfully disagree with your assertion that the third-order smoothness assumption is used to eliminate the large batch size requirement. For clarity, you may refer to the proof of SOBA, which better illustrates the necessity of this assumption in ensuring the smoothness of the solution mapping for the single-loop update. Moreover, the analysis of SOBA has been tightened in a recent work (https://arxiv.org/abs/2410.13743), which removes the need for the third-order Lipschitz assumption and might help clarify the role of this assumption.
> > > >
> > > >
> > > > **Our Response:** Thanks for your comments. We would like to clarify that our argument to the necessity of the third-order Lipschitz assumption is based on the MA-SOBA paper [1]. In MA-SOBA, the authors claimed that the third-order Lipschitz assumption is required to bound the bias in the Hessian-inverse approximation. In addition, they also showed that this assumption can be removed by using momentum-based SGD updates in their MA-SOBA framework.
> > > >
> > > > We also appreciate the reviewer bringing the recent work (https://arxiv.org/abs/2410.13743) to our attention. While this work appears to provide a tightened analysis for SOBA and eliminates the third-order Lipschitz assumption, it was uploaded to arXiv after the submission deadline for ICLR 2025. It's somewhat unfair to reject our paper based on a new paper posted on arXiv after our submission. Nevertheless, we are very interested in checking this paper and see if this could potentially help address the large batch size requirement in our work.
> > > >
> > > > [1] Chen, Xuxing, Tesi Xiao, and Krishnakumar Balasubramanian. "Optimal algorithms for stochastic bilevel optimization under relaxed smoothness conditions." Journal of Machine Learning Research 25, no. 151 (2024): 1-51.

---

### Official Review · Reviewer_ktRo · 2024-11-04

**Soundness:** 2
**Presentation:** 3
**Contribution:** 2
**Rating:** 5
**Confidence:** 5

**Summary:**

This paper proposes a novel Lazy Hessian Evaluation method, which allows less Hessian vector product computation in overall iterations by using only one Hessian vector product for multiple hypergradient estimations.

**Strengths:**

1. This work aims to tackle a very important issue in bilevel optimization and such a method has never been tried in previous works.
2. The numerical experiment shows a significant improvement in the proposed method.

**Weaknesses:**

1. Theorem 5.3 looks like the algorithm has some trouble in convergence when the gradient estimation variance is not arbitrarily small. The only way to solve this problem is to take a sample size that is large enough. Previous methods allow such error terms to diminish with iteration. This is a big disadvantage of this algorithm. As a result, Corollary 5.4 shows that the proposed method has bare improvement in partial gradient and HVP evaluations.
2. It may be necessary to assume $l^*$ to be lower bounded for lines 372 and 428.

**Questions:**

1. Since the author assumes the strong convexity of the inner problem, why the challenge 2 is still challenging?
2. The method only considers the computation of HVP. Can the author further explain how to reduce the computation of JVPs as mentioned in line 308?
3. Theorem 5.3 indicates that N=1 results in the optimal convergence rate, which is SOBA. Then how can the proposed method improve the convergent in theory? Combining the weakness mentioned above, the proposed method has marginally improved.
4. Can we author explain what is the main reason for such a significant improvement in Table 2 since the theory can not show such an advantage?

---

> ### Author Response · Authors · 2024-11-22
>
> ### Weaknesses:
>
> > **Comment 1:** Theorem 5.3 looks like the algorithm has some trouble in convergence when the gradient estimation variance is not arbitrarily small. The only way to solve this problem is to take a sample size that is large enough. Previous methods allow such error terms to diminish with iteration. This is a big disadvantage of this algorithm. As a result, Corollary 5.4 shows that the proposed method has bare improvement in partial gradient and HVP evaluations.
>
>
> **Our Response:** Thanks for your comment. We would like to address your concerns and provide additional context as follows:
> 1. The large batch size is required only for the *theoretical convergence guarantees*. In our experiments, we use a small batch size and still show fast-convergence of LazyBLO compared to the state-of-the-art baselines. This practical evidence indicates that the theoretical requirement of a large batch size does not negatively impact the real-world applicability of the proposed method.
> 2. The need for large batch sizes is not unique to LazyBLO. Many existing bilevel optimization algorithms, such as stocBiO, AmIGO, and iNEON [R1], also adopt large batch sizes in their theoretical analyses to control variance. Additionally, our approach employs SGD to approximate Hessian-vector product by solving a quadratic problem. Previous bilevel optimization algorithms using the same approximation are AmIGO and SOBA. Our approach shares a similarity with AmIGO, which avoids strong third-order Lipschitz continuity assumptions at the cost of requiring large batch sizes. In contrast, SOBA relies on these stronger assumptions to manage variance. In addition, previous methods, such as BSA, TTSA, SUSTAIN, MSTSA, VRBO, MRBO and ALSET require all the stochastic functions to satisfy Assumptions 3.1 and 3.2 stated in the manuscript to avoid large batch gradient computations. Note that this is certainly a very strong assumption and is not required in our analysis of LazyBLO. This indicates that large batch sizes are a common trade-off in bilevel optimization methods, rather than a disadvantage specific to LazyBLO.
> 6. Since we use a lazy approach where we perform infrequent Hessian evaluations that incur additional errors, our aim is *not to outperform* the non-lazy approaches. Rather, our goal is to be **as close to the non-lazy counterpart approaches as possible**. Due to the additional errors accumulated because of the use of these stale Hessians, approximate HVP evaluations, and the coupling hierarchical structure of the bilevel problems, it is unclear whether LazyBLO will converge or not. In fact, establishing convergence for the lazy approach proposed in this work is itself **surprising** and a major contribution of our work.
>
>
> [R1] Huang, M., Ji, K., Ma, S., & Lai, L. (2022). Efficiently escaping saddle points in bilevel optimization. arXiv preprint arXiv:2202.03684.
>
>
>
>
>
> > **Comment 2:** It may be necessary to assume $l^*$ to be lower bounded for lines 372 and 428.
>
>
> **Our Response:** Thank you for your comment. We will include the lower-bounded assumption of $l^*$ in the revised version of our paper.

---

> ### Author Response · Authors · 2024-11-22
>
> ### Questions:
>
> > **Comment 1:** Since the author assumes the strong convexity of the inner problem, why the challenge 2 is still challenging?
>
>
> **Our Response:** Thank you for your question. While it is true that strongly convex inner problems are generally easier to solve compared to non-convex ones, the assumption of strong convexity does not imply that these problems can always be solved in closed form, except in a few very specific cases. In most scenarios, well-designed numerical procedures are still required to solve the inner problems approximately. In addition, the approximation to which the lower level problem is solved determines the **bias** in the implicit gradient computation of the objective function. As a consequence, this bias needs to be controlled carefully (both theoretically and numerically) to ensure convergence of the algorithm.
>
>
> > **Comment 2:** The method only considers the computation of HVP. Can the author further explain how to reduce the computation of JVPs as mentioned in line 308?
>
>
> **Our Response:** Thanks for your question. Rewrite Equation (5) in our paper as
> $h_{t,n}^{f}  =  \nabla_{x} f  \left(  x_{t}^{n}, y_{t}^{n}; D_{t,n}^{f_{x}}  \right)  +  H_{t}^{j}z_{t}$, where $H_{t}^{j}:=\nabla_{x y}^{2} g  \left(x_{t}^{N}, y_{t}^{N}; D_{t}^{g_{xy}}\right)$. Then, it can be seen that JVP (i.e., $H_{t}^{j}z_{t}$) would be updated **less frequently** compared to HVP, since it is in the outer loop of the algorithm. If one wants to reduce the computation JVP, a similar lazy approach can also be applied on the JVP update in LazyBLO.
>
>
> > **Comment 3:** Theorem 5.3 indicates that N=1 results in the optimal convergence rate, which is SOBA. Then how can the proposed method improve the convergent in theory? Combining the weakness mentioned above, the proposed method has marginally improved.
>
>
> **Our Response:** Thanks for your comment. We would like to clarify that when $N=1$, our proposed method recovers the result of AmIGO, not SOBA. This is because SOBA relies on an additional assumption—third-order Lipschitz continuity—which our method does not require. Furthermore, since we use a lazy approach where infrequent Hessian evaluations incurs additional errors, our goal is *not to outperform* the non-lazy approaches. Rather, our goal is to be **as close to the non-lazy counterpart approaches as possible**. Due to the additional errors accumulated because of the use of these stale Hessians, approximate HVP evaluations, and the coupling hierarchical structure of the bilevel problems, it is unclear whether LazyBLO will converge or not. In fact, establishing convergence for the lazy approach proposed in this work is itself **surprising** and a major contribution of our work.
>
>
> > **Comment 4:** Can we author explain what is the main reason for such a significant improvement in Table 2 since the theory can not show such an advantage?
>
>
> **Our Response:** Thank you for your comment. The convergence rate alone can sometimes be misleading, as it is based on the number of iterations, without accounting for the computational cost of each iteration. To provide a more objective comparison, we measure performance using **wall-clock time**. LazyBLO achieves significant improvements in wall-clock time because it performs infrequent Hessian evaluations, which greatly reduce computational overhead. This efficiency explains the notable improvement observed in Table 2, even if the theoretical convergence rate does not directly reflect this advantage.

---

> > ### Comment · Reviewer_ktRo · 2024-11-25
> >
> > I truly appreciate the detailed feedback from the author. I do still have some questions:
> > 1. Certainly, removing the third-order Lipschitz continuity assumption is significant. Am I correct in understanding that using a large batch size serves as a trade-off for removing this assumption? Additionally, does this technique originate from AmiGO?
> > 2. To reduce JVP, can you still maintain the same convergence result? If my understanding is correct, this would require moving the update of $x_{t+1}$ to the outer loop, which indicates that $x$ is updated less frequently. I guess this may impact the convergence result. Moreover, what will happen if one wants to update both HVP and JVP? Based on the algorithm, it appears that the inner problem is solved solely through a sub-loop, with $x$ and $z$ being updated only once. While I do not expect too detailed proof given the limited time, additional explanation on this point would be very helpful.

---

> > > ### Author Response · Authors · 2024-11-27
> > >
> > > > **Comment 1:** Certainly, removing the third-order Lipschitz continuity assumption is significant. Am I correct in understanding that using a large batch size serves as a trade-off for removing this assumption? Additionally, does this technique originate from AmiGO?
> > >
> > >
> > > **Our Response:** Thank you for your follow-up questions. The answers to both the questions are yes. Using a large batch size allows us to remove the third-order Lipschitz continuity assumption, and this technique originates from the AmIGO algorithm.

---

> > > ### Author Response · Authors · 2024-11-27
> > >
> > > > **Comment 2:** To reduce JVP, can you still maintain the same convergence result? If my understanding is correct, this would require moving the update of $x_{t+1}$ to the outer loop, which indicates that $x$ is updated less frequently. I guess this may impact the convergence result. Moreover, what will happen if one wants to update both HVP and JVP? Based on the algorithm, it appears that the inner problem is solved solely through a sub-loop, with $x$ and $z$ being updated only once. While I do not expect too detailed proof given the limited time, additional explanation on this point would be very helpful.
> > >
> > >
> > > **Our Response:** Thanks for your comments and the suggestions. We would like to clarify that reducing Jacobian computations does **not** require moving the update of $x_{t+1}$ to the outer loop. Instead, the Jacobian matrix $J_t$ is moved to the outer loop, thereby reducing its update frequency compared to that of $x_t$. To see this, we provide the algorithm for reducing Jacobian computations in Algorithm 1 below.  In addition, we present the algorithm for reducing both Hessian and Jacobian computations in Algorithm 2 below. In both Algorithms 1 and 2, the inner problem is solved within the inner-loop, together with $x$.
> > >
> > > The convergence results for both Algorithms 1 and 2 will stay unchanged because the reduced Jacobian and Hessian computations primarily influence the error terms in hyper-gradient estimations, not the fundamental iteration complexity of LazyBLO. These error terms are explicitly bounded in the analysis and diminish appropriately with properly designed hyper-parameters in our algorithms. As a result, the accuracy of the hyper-gradient remains sufficient to ensure convergence, preserving the theoretical guarantees of the algorithm.
> > >
> > > |Algorithm 1: The Algorithm for Reducing Jacobian Computations Only.|
> > > | -------- |
> > > | **Input:** Initial parameters $x_{0}^{0}$, $y_{0}^{0}$, $z_{0}$, and step-sizes $\\{ \alpha_{t}, \beta_{t}, \gamma_{t} \\}_{t=0}^{T-1}$     |
> > > | **for** $t=0$ **to** $T-1$ **do** |
> > > | $\quad$ **for** $n=0$ **to** $N-1$ **do** |
> > > |$\qquad$ Initialize $x_{t}^{0}=x_{t-1}^{N}$ and $y_{t}^{0}=y_{t-1}^{N}$|
> > > |$\qquad$ Sample data batches $D_{t,n} ^{g_{yy}}$,  $D_{t,n}^{f_{y}}$, $D_{t,n}^{g}$ and $D_{t,n}^{f_{x}}$|
> > > |$\qquad$ Compute the gradient estimate $h_{t,n}^{q}$ using $h_{t,n}^{q}=\nabla_{y y}^{2} g(x_{t}^{n}, y_{t}^{n}; D_{t,n}^{g_{yy}}) z_{t}+\nabla_{y} f(x_{t}^{n}, y_{t}^{n}; D_{t,n}^{f_{y}} )$ and update $z_{t}^{n+1}=z_{t}^n-\gamma_{t}h_{t,n}^{q}$|
> > > |$\qquad$ Compute the gradient estimate $h_{t,n}^{g}$ using $h_{t,n}^{g}=\nabla_{y} g\left(x_{t}^{n}, y_{t}^{n}; D_{t,n}^{g}\right)$ and update $y_{t}^{n+1}=y_{t}^{n}-\beta_{t}h_{t,n}^{g}$|
> > > |$\qquad$ Compute the gradient estimate $h_{t,n}^{f}$ using $h_{t,n}^{f}  =  \nabla_{x} f  \left(  x_{t}^{n}, y_{t}^{n}; D_{t,n}^{f_{x}}  \right)  +  J_t  z_{t}^n$ and update $x_{t}^{n+1}=x_{t}^{n}-\alpha_{t}h_{t,n}^{f}$|
> > > |$\quad$ **end for**|
> > > |$\quad$ Sample data batches $D_{t}^{g_{xy}}$|
> > > |$\quad$  Compute ${\color{blue} J_t = \nabla_{x y}^{2} g  \left(x_{t}^{N}, y_{t}^{N}; D_{t}^{g_{xy}}\right)}$ |
> > > |**end for**|
> > >
> > >
> > > |Algorithm 2: The Algorithm for Reducing both Hessian and Jacobian Computations.|
> > > | -------- |
> > > | **Input:** Initial parameters $x_{0}^{0}$, $y_{0}^{0}$, $z_{0}$, and step-sizes $\\{ \alpha_{t}, \beta_{t}, \gamma_{t} \\}_{t=0}^{T-1}$     |
> > > | **for** $t=0$ **to** $T-1$ **do** |
> > > | $\quad$ **for** $n=0$ **to** $N-1$ **do** |
> > > |$\qquad$ Initialize $x_{t}^{0}=x_{t-1}^{N}$ and $y_{t}^{0}=y_{t-1}^{N}$|
> > > |$\qquad$ Sample data batches $D_{t,n}^{g}$ and $D_{t,n}^{f_{x}}$|
> > > |$\qquad$ Compute the gradient estimate $h_{t,n}^{g}$ using $h_{t,n}^{g}=\nabla_{y} g\left(x_{t}^{n}, y_{t}^{n}; D_{t,n}^{g}\right)$ and update $y_{t}^{n+1}=y_{t}^{n}-\beta_{t}h_{t,n}^{g}$|
> > > |$\qquad$ Compute the gradient estimate $h_{t,n}^{f}$ using $h_{t,n}^{f}  =  \nabla_{x} f  \left(  x_{t}^{n}, y_{t}^{n}; D_{t,n}^{f_{x}}  \right)  +  J_t  z_{t}$ and update $x_{t}^{n+1}=x_{t}^{n}-\alpha_{t}h_{t,n}^{f}$|
> > > |$\quad$ **end for**|
> > > |$\quad$ Sample data batches $D_{t}^{g_{yy}}$,  $D_{t}^{f_{y}}$, and $D_{t}^{g_{xy}}$|
> > > |$\quad$ Compute ${\color{blue} J_t = \nabla_{x y}^{2} g  \left(x_{t}^{N}, y_{t}^{N}; D_{t}^{g_{xy}}\right) }$ |
> > > |$\quad$ Compute the gradient estimate $h_{t}^{q}$ using $ {\color{blue} h_{t}^{q}=\nabla_{y y}^{2} g(x_{t}^{N}, y_{t}^{N}; D_{t}^{g_{yy}}) z_{t}+\nabla_{y} f(x_{t}^{N}, y_{t}^{N}; D_{t}^{f_{y}} ) }$ and update $z_{t+1}=z_{t}-\gamma_{t}h_{t}^{q}$|
> > > |**end for**|

---

> > > > ### Comment · Reviewer_ktRo · 2024-11-27
> > > >
> > > > Thank you for the author’s feedback. Could you clarify whether you mean to compute $J_t$ or $J_tz_t$ in the outer loop? Calculating a second-order matrix is significantly more computationally expensive than a matrix-vector product. However, if you intend to compute $J_tz_t$, there appears a product of lazy term. My question is: would the error introduced by lazy computation scale quadratically? Specifically, does this imply that the first term in Theorem 5 could become $\mathcal{O}(N^2/T)$?
> > > >
> > > > Moreover, I had a hard time finding the proof of Corollary 5.4. Could the author please show me the details? I am really curious about what the complexity will be if lazy computing both JVP and HVP.

---

> > > > > ### Author Response · Authors · 2024-11-29
> > > > >
> > > > > > **Comment 1:** Thank you for the author’s feedback. Could you clarify whether you mean to compute $J_t$ or $J_tz_t$ in the outer loop? Calculating a second-order matrix is significantly more computationally expensive than a matrix-vector product. However, if you intend to compute $J_tz_t$, there appears a product of lazy term. My question is: would the error introduced by lazy computation scale quadratically? Specifically, does this imply that the first term in Theorem 5 could become $\mathcal{O}(N^2/T)$?
> > > > >
> > > > > **Our Response:** Thanks for your comment. It is true that computing a second-order matrix is far more computationally expensive than computing a matrix-vector product. We would like to clarify that Algorithm 1 is designed to **only** reduce the computation frequency of the Jacobian matrix (see the title of Algorithm 1). The reason we presented Algorithm 1 is just to illustrate the "Lazy-Jacobian-Only" method, which is the counterpart of the "Lazy-Hessian-Only" method developed in this paper. We note that a key limitation of Algorithm 1 is that, to reduce the frequency of Jacobian-related computation *alone*, it has to compute Jacobian matrix separately and loses the opportunity of computing Jacobian-vector product (JVP). **This limitation is exactly one of the reasons that we are not interested in developing the "Lazy-Jacobian-Only" approach in this paper**.
> > > > >
> > > > > On the other hand, to answer your previous question on how to reduce **both** Jacobian and Hessian computations, we proposed Algorithm 2 (cf. Algorithm 2's title) in our previous response that moves the Jacobian computation to the outer loop and performs Hessian-vector product (HVP) in the outer loop. Also, it is possible to further modify Algorithm 2 to **only evaluate JVP** rather than to directly compute the Jacobian. This modified version of Algorithm 2 is illustrated as follows:
> > > > >
> > > > > |Algorithm 2a: The Algorithm for Reducing both HVP and JVP Computations.|
> > > > > | -------- |
> > > > > | **Input:** Initial parameters $x_{0}^{0}$, $y_{0}^{0}$, $z_{0}$, and stepsize $\\{ \alpha_{t}, \beta_{t}, \gamma_{t} \\}_{t=0}^{T-1}$     |
> > > > > | **for** $t=0$ **to** $T-1$ **do** |
> > > > > | $\quad$ **for** $n=0$ **to** $N-1$ **do** |
> > > > > |$\qquad$ Initialize $x_{t}^{0}=x_{t-1}^{N}$ and $y_{t}^{0}=y_{t-1}^{N}$|
> > > > > |$\qquad$ Sample data batches $D_{t,n}^{g}$ and $D_{t,n}^{f_{x}}$|
> > > > > |$\qquad$ Compute the gradient estimate $h_{t,n}^{g}$ using $h_{t,n}^{g}=\nabla_{y} g\left(x_{t}^{n}, y_{t}^{n}; D_{t,n}^{g}\right)$ and update $y_{t}^{n+1}=y_{t}^{n}-\beta_{t}h_{t,n}^{g}$|
> > > > > |$\qquad$ Compute the gradient estimate $h_{t,n}^{f}$ using $h_{t,n}^{f}  =  \nabla_{x} f  \left(  x_{t}^{n}, y_{t}^{n}; D_{t,n}^{f_{x}}  \right) +{\color{blue}JVP_t}$ and update $x_{t}^{n+1}=x_{t}^{n}-\alpha_{t}h_{t,n}^{f}$|
> > > > > |$\quad$ **end for**|
> > > > > |$\quad$ Sample data batches $D_{t}^{g_{yy}}$,  $D_{t}^{f_{y}}$, and $D_{t}^{g_{xy}}$|
> > > > > |$\quad$ Compute the gradient estimate $h_{t}^{q}$ using $h_{t}^{q}=\nabla_{y y}^{2} g(x_{t}^{N}, y_{t}^{N}; D_{t}^{g_{yy}}) z_{t}+\nabla_{y} f(x_{t}^{N}, y_{t}^{N}; D_{t}^{f_{y}} )$ and update $z_{t+1}=z_{t}-\gamma_{t}h_{t}^{q}$|
> > > > > |$\quad$ Compute $ {\color{blue} JVP_{t+1} = \nabla_{x y}^{2} g  \left(x_{t}^{N}, y_{t}^{N}; D_{t}^{g_{xy}}\right)z_{t+1}}$ |
> > > > > |**end for**|
> > > > >
> > > > >
> > > > > In Algorithm 2a, the computation frequency of the Jacobian-vector product (JVP) is reduced, and JVP ($JVP_{t+1}$) would be computed in the outer loop. Moreover, since $z_t$ is the HVP estimator, reducing the computation frequency of JVP also reduces the computation frequency of HVP. Thus, the reduction in computation frequency for both JVP and HVP is coupled.
> > > > >
> > > > > Regarding the last question about whether the first term in Theorem 5 could become $\mathcal{O}(N^2/T)$ or not, we acknowledge that this could happen due to the "multiplicative" structure of $JVP_t$, which is coupled with both JVP and HVP. However, without a careful and rigorous theoretical analysis, we cannot definitively confirm or refute this conjecture right now due to the limited time in this rebbutal period. Also, this "Joint-Lazy-JVP-HVP" question is somewhat beyond the scope of our current work and has a far more complicated computational structure, which deserves an independent paper dedicated to resolving this conjecture. We thank the reviewer for bringing up this interesting direction in our future studies.

---

> > > > > ### Author Response · Authors · 2024-11-29
> > > > >
> > > > > > **Comment 2:** Moreover, I had a hard time finding the proof of Corollary 5.4. Could the author please show me the details? I am really curious about what the complexity will be if lazy computing both JVP and HVP.
> > > > >
> > > > > **Our Response:** Thanks for your question. We omitted the proof of Corollary 5.4 since it's relatively straightforward. The computation complexity of HVP evaluations in Corollary 5.4 can be computed as the **product** of the number of iterations $T$ and the batch size $\left| \mathcal{D}^{z} \right|$. Specifically, note that:
> > > > > * $\mathcal{O}(T)=\mathcal{O}(\epsilon^{-1})$, and
> > > > > * $\left| \mathcal{D}^{z} \right| = \max \\{ \left| \mathcal{D}^{f_{y}} \right|,\left| \mathcal{D}^{g_{yy}} \right| \\}=\Theta\left( \epsilon^{-1} \right)$.
> > > > >
> > > > > Hence, it immediately follows that the computation complexity of HVP is $\mathcal{O}(\epsilon^{-2})$.
> > > > >
> > > > > Similarly, the computation complexity of partial gradient can be computed as the **product** of the number of iterations $T$ and the batch size $\left| \mathcal{D}^{f} \right|$. Specifically, note that:
> > > > > * $\mathcal{O}(T)=\mathcal{O}(\epsilon^{-1})$, and
> > > > > * $\left| \mathcal{D}^{f} \right| = \max \\{ \left| \mathcal{D}^{f_{x}} \right|, \left| \mathcal{D}^{g_{xy}} \right| \\}=\Theta\left( \epsilon^{-1} \right)$.
> > > > >
> > > > > Hence, it again immediately follows that the computation complexity of partial gradient is $\mathcal{O}(\epsilon^{-2})$.

---

> > > > > > ### Comment · Reviewer_ktRo · 2024-12-02
> > > > > >
> > > > > > Thank you for the author's feedback. I will raise my score to 5.
> > > > > >
> > > > > > However, there are still some limitations in this work. First, it focuses solely on the Hessian-efficient version, leaving some unexplored issues in the Jacobian-efficient version, such as the convergence rate in that scenario. Additionally, I am unclear why $N$ is omitted in Corollary 5.4; I believe the author should address this point more thoroughly in the revision. Overall, there is significant room for improvement to make this work more comprehensive.

---

### Meta-Review · Area_Chair_C9aT · 2024-12-20

**Metareview:**

This paper proposes a new algorithm for bilevel optimization aiming to reduce the number of Hessian-vector products. By viewing the Hessian-vector product as a solution to a quadratic minimization problem, it computes the solution periodically, which gives the so-called lazy Hessian evaluation framework.

The main concerns of the paper are that (1) it does not improve the complexity of previous algorithms; (2) it requires a large batch size. Hence, the algorithm is not very practical. Other concerns like why not reducing the JVP evaluations. The authors responded in the rebuttal but did not give a full answer.

Overall, the contribution of the paper is quite limited. Hence, I recommend a rejection.

**Additional Comments On Reviewer Discussion:**

The reviewers engaged with the authors during the discussions and acknowledged the authors' rebuttal and also have some remaining concerns.

---

### Decision · Program_Chairs · 2025-01-22

Reject